# Beyond Convexity: Proximal-Perturbed Lagrangian Methods for Efficient Functional Constrained Optimization

## Abstract

Non-convex functional constrained optimization problems have gained substantial attention in machine learning and data science, addressing broad requirements that typically go beyond the often performance-centric objectives. An influential class of algorithms for functional constrained problems is the class of primal-dual methods which has been extensively analyzed for convex problems. Nonetheless, the investigation of their efficacy for non-convex problems is under-explored. This paper develops a primal-dual algorithmic framework for solving such non-convex problems. This framework is built upon a novel form of the Lagrangian function, termed the *Proximal-Perturbed Augmented Lagrangian*, which enables the development of simple first-order algorithms that converge to a stationary solution under mild conditions. Notably, we study this framework under both non-smoothness and smoothness of the constraint function and provide three key contributions: (i) a single-loop algorithm that does not require the continuous adjustment of the penalty parameter to infinity; (ii) a non-asymptotic iteration complexity of $\widetilde{\mathcal{O}}(1/\epsilon^2)$; and (iii) extensive experimental results demonstrating the effectiveness of the proposed framework in terms of computational cost and performance, outperforming related approaches that use regularization (penalization) techniques and/or standard Lagrangian relaxation across diverse non-convex problems.

## 1 Introduction

We consider the following non-convex optimization problem with functional constraints:

$$\min_{\mathbf{x} \in \mathbb{R}^n} \quad f(\mathbf{x}) + r(\mathbf{x}) \quad \text{s.\,t.} \quad g(\mathbf{x}) \leq \mathbf{0}, \tag{1}$$

where $f : \mathbb{R}^n \to \mathbb{R}$ and $g : \mathbb{R}^n \to \mathbb{R}^m$ are continuous and possibly non-convex mappings; and $r : \mathbb{R}^n \to \mathbb{R} \cup \{+\infty\}$ is a proper, closed, and convex (possibly non-smooth) function.

Problems of this form equation 1 appear in a wide range of applications in machine learning, data science, and signal processing, e.g., wireless transmit/receive beamforming design Scutari et al. (2016b); Shi et al. (2020), constrained classification/detection problems Huang & Vishnoi (2019); Rigollet & Tong (2011); Zafar et al. (2019), and optimization for deep neural network Bai et al. (2023); Jiang & Chen (2023). Solving non-convex problems, even those without constraints, is generally challenging, as finding even an approximate global minimum is often computationally intractable Nemirovskij & Yudin (1983). The presence of functional constraints $g(\mathbf{x})$ in equation 1 that can potentially be non-convex is critical for many of the applications mentioned above, yet it makes the problem even more challenging. A further complication arises since in many of these applications, problem equation 1 tends to be large-scale, i.e., with large variable dimension $n$ Boyd et al. (2011). Hence, developing first-order methods that can find stationary solutions with lower complexity bounds is highly desirable.

Augmented Lagrangian (AL)-based algorithms are a prevailing class of approaches for constrained optimization problems. The foundational AL method, introduced by Hestenes (1969) and Powell (1969), has been a powerful algorithmic framework built on by many contemporary algorithms. In particular, the Alternating Direction Method of Multipliers (ADMM) scheme has been widely employed for solving constrained optimization problems based on the AL framework; see Bertsekas (2014); Birgin & Martínez (2014) and recent

works for constrained convex settings Ouyang et al. (2015); Lan & Monteiro (2016); Xu (2017); Liu et al. (2019); Xu (2021); Upadhyay et al. (2025).

However, AL-based methods remain fairly limited for problems in the general form of equation 1. Key challenges arise from the non-convexity of the objective and constraint functions, which can lead to complicated updates with no closed-form or compute-efficient updates and require carefully updating the penalty parameters to ensure the solution remains near feasible. Consequently, existing analyses of AL-based methods, with the best-known guarantees of $\mathcal{O}(1/\epsilon^3)$ for a given $\epsilon > 0$, require increasing penalty parameters to infinity to ensure feasibility, leading to demanding iteration complexity.

Motivated by the above discussion, we aim to answer the question:

> *Can we design algorithms to solve problems of the form equation 1 with an improved iteration complexity bound and efficient update rules?*

This paper answers this question in the affirmative. Notably, we develop an efficient single-loop first-order primal-dual method for solving problem equation 1 such that based on a new augmented Lagrangian, for a given accuracy $\epsilon > 0$ to compute an $\epsilon$-approximate stationary solution (see Definition 2). It achieves an iteration complexity of $\widetilde{\mathcal{O}}(1/\epsilon^2)$ in terms of the number of gradient evaluations.[1]

## 1.1 Related Work

We review the literature on iteration complexity and convergence of AL and penalty-based methods for non-convex constrained problems.

**Linearly constrained non-convex problems.** Many existing works have focused on the class of problems where $g(\mathbf{x})$ in equation 1 is linear. Hajinezhad & Hong (2019) introduced a perturbed-proximal primal-dual algorithm, with an iteration complexity of $\widetilde{\mathcal{O}}(1/\epsilon^4)$, under the assumption of a feasible initialization. Kong et al. (2023) proposed proximal AL methods that obtain the improved complexity result of $\widetilde{\mathcal{O}}(1/\epsilon^3)$ under Slater's condition. Finally, Zhang & Luo (2020; 2022) proposed a first-order single-loop proximal AL method that achieves $\mathcal{O}(1/\epsilon^2)$ iteration complexity, which relies on error bounds that depend on the Hoffman constant of the polyhedral constraints.[2] However, estimating the Hoffman constant is known to be difficult in practice.

**Non-convex functional constrained problems.** There are several recent works that focus on the iteration complexity of first-order AL-based methods or penalty methods to solve equation 1 Cartis et al. (2011); Scutari et al. (2016a); Li et al. (2021); Lin et al. (2022); Lu (2022); Kong et al. (2022); Sahin et al. (2019). Scutari et al. (2016a) proposed double-loop distributed primal-dual algorithms with asymptotic convergence guarantees, under the coercivity assumption and Mangasarian-Fromovitz constraint qualification (MFCQ). More recently, a set of methods have emerged employing the regularity condition (Assumption 4 of Lin et al. (2019), (8) of Sahin et al. (2019)) for ensuring solution feasibility. Sahin et al. (2019) proposed a double-loop inexact AL method (iALM) that achieves an $\widetilde{\mathcal{O}}(1/\epsilon^4)$ iteration complexity. Li et al. (2021) improved the iteration complexity to $\widetilde{\mathcal{O}}(1/\epsilon^3)$, which is obtained using a triple loop iALM. Kong et al. (2022) established an $\widetilde{\mathcal{O}}(1/\epsilon^3)$ complexity bound of the proximal AL method (NL-IAPIAL) for non-convex problems with non-linear convex constraints. Lu (2022) proposed the first single-loop gradient-based algorithm that achieves the best-known iteration complexity $\mathcal{O}(1/\epsilon^3)$ for equation 1. However, the regularity condition is non-standard and rather strong as it forces a relationship between feasibility of the generated iterates and first-order optimality. We are thus motivated to develop an algorithm that improves iteration complexity without requiring this assumption. While our proposed framework shares similarity with the above influential works, it further provides certain key components which we delineate in Section 1.2.

---

[1] In this paper, the notation $\widetilde{\mathcal{O}}(\cdot)$ suppresses all logarithmic factors from the big-$\mathcal{O}$ notation.
[2] The Hoffman constant $\kappa$ is the smallest number such that for any $\mathbf{x}$, $\text{dist}(\mathbf{x}, \{\mathbf{y} \mid A\mathbf{y} \leq b\}) \leq \kappa \|(A\mathbf{x}-b)_+\|$, where $(A\mathbf{x}-b)_+$ denotes the positive part of $A\mathbf{x} - b$.

### 1.2 Our Contributions

In this paper, we develop a novel algorithmic framework designed to solve challenging non-convex functional constrained optimization problems. Our paper offers the following key contributions:

- We propose a single-loop first-order algorithm that finds an $\epsilon$-solution with $\widetilde{\mathcal{O}}(1/\epsilon^2)$ iteration complexity for both non-smooth and smooth constraint functions without requiring the strong regularity condition used in other AL-based algorithms Li et al. (2021); Lin et al. (2022); Lu (2022); Sahin et al. (2019).

- To establish the above results, we conduct a comprehensive convergence analysis of our method for both non-smooth and smooth constraint functions, establishing simple and concise proofs compared to existing works. Notably, the analysis does not impose assumptions on the surjectivity of the Jacobian $J_g(\mathbf{x})$ Bolte et al. (2018); Boţ & Nguyen (2020), or boundedness of penalty parameters Grapiglia & Yuan (2021). It also does not require the feasibility of initialization as in Boob et al. (2022); Sun & Sun (2021); Xie & Wright (2021), which itself is a non-convex and challenging task.

- By using a constant penalty parameter, our algorithm achieves improved computational efficiency and ease of implementation compared to existing schemes. Specifically, we neither require linear independence constraint qualification (LICQ) to ensure boundedness of penalty parameters Solodov (2009), nor computational efforts for careful updating scheme of the penalty parameters. Our numerical results validate that compared with existing methods, using a fixed penalty parameter achieves more consistent progress toward solution stationarity and feasibility.

- The algorithmic framework is flexible, enabling it to effectively handle various non-convex, smooth, and non-smooth functional constraints. Experimental results demonstrate its high effectiveness in terms of computational cost and performance, outperforming related algorithms that use regularization techniques and/or standard Lagrangian relaxation.

### 1.3 Outline

Section 2 provides the notation, definitions, and assumptions that we use throughout the paper. In Section 3 and 4, we propose novel first-order primal-dual algorithms and establish their convergence results for non-smooth and smooth functional constraints, respectively. Section 5 presents numerical results on commonly encountered problems in signal processing and machine learning to demonstrate the effectiveness of the proposed algorithm. Detailed derivations are provided in the supplementary material due to space limitations.

## 2 Preliminaries

This section provides the notation, formal definitions, and assumptions utilized throughout this paper, forming the foundation for our proposed algorithmic approach and its convergence analysis.

We adopt the following notation: $\mathbb{R}^n$ denotes the $n$-dimensional Euclidean space, and $\mathbb{R}^n_+$ represents the non-negative orthant. We use $[m]$ to denote the set $\{1, \ldots, m\}$. The inner product between two vectors is denoted by $\langle \cdot, \cdot \rangle$, and the Euclidean norm of matrices and vectors is denoted by $\| \cdot \|$. The distance function between a vector $\mathbf{x}$ and a set $\mathcal{X} \subseteq \mathbb{R}^n$ is defined as $\text{dist}(\mathbf{x}, \mathcal{X}) := \inf_{\mathbf{y} \in \mathcal{X}} \|\mathbf{y} - \mathbf{x}\|$. For a proper extended real-valued function $r$, its *domain* is defined as $\text{dom}(r) := \{\mathbf{x} \in \mathbb{R}^n : r(\mathbf{x}) < +\infty\}$. A function $r$ is considered *proper* if $\text{dom}(r) \neq \emptyset$ and it does not take the value $-\infty$. It is *closed* if it is lower semicontinuous, meaning $\liminf_{\mathbf{x} \to \mathbf{x}^0} r(\mathbf{x}) \geq r(\mathbf{x}^0)$ for any $\mathbf{x}^0 \in \mathbb{R}^n$. For a convex function $r$ at $\mathbf{x}$, its *subgradient* is denoted by $\partial r(\mathbf{x}) := \{\mathbf{d} \in \mathbb{R}^n : r(\mathbf{y}) \geq r(\mathbf{x}) + \langle \mathbf{d}, \mathbf{y} - \mathbf{x} \rangle, \forall \mathbf{y} \in \mathbb{R}^n, \mathbf{x} \in \text{dom}(r)\}$. The *proximal map* associated with a proper, closed, and convex function $r : \mathbb{R}^n \to \mathbb{R} \cup \{+\infty\}$ at $\mathbf{x} \in \mathbb{R}^n$ with $\eta > 0$ is uniquely defined by $\text{prox}_{\eta r}(\mathbf{x}) = \text{argmin}_{\mathbf{y} \in \mathbb{R}^n} \left\{ r(\mathbf{y}) + \frac{1}{2\eta} \|\mathbf{x} - \mathbf{y}\|^2 \right\}$.

Next, we provide the formal definitions and assumptions for the class of functions, and the optimality measure under consideration.

Assuming a suitable constraint qualification (CQ) holds, the stationary solutions of problem equation 1 can be characterized by the points $(\mathbf{x}^*, \boldsymbol{\nu}^*)$ satisfying the Karush-Kuhn-Tucker (KKT) conditions Bertsekas (1999).

**Definition 1** (The KKT point). A point $\mathbf{x}^*$ is called a *KKT point* for problem equation 1 if there exists $\boldsymbol{\nu}^* \in \mathbb{R}_+^m$ such that

$$\begin{cases} \mathbf{0} \in \nabla f(\mathbf{x}^*) + \partial r(\mathbf{x}^*) + J_g(\mathbf{x}^*)\boldsymbol{\nu}^*, \\ g_j(\mathbf{x}^*) \leq 0, \quad \boldsymbol{\nu}_j g_j(\mathbf{x}^*) = 0, \quad j \in [m]. \end{cases} \tag{2}$$

A suitable CQ is necessary for the existence of multipliers that satisfy the KKT conditions (e.g., MFCQ, Constant Positive Linear Dependence (CPLD), and others; see Andreani et al. (2022); Bertsekas (1999)). In practice, it is difficult to find an exact KKT solution $(\mathbf{x}^*, \boldsymbol{\nu}^*)$ that satisfies equation 2. Thus, one typically aims to find an approximate KKT solution, defined as an $\epsilon$-KKT solution next.

**Definition 2** ($\epsilon$-KKT solution, Definition 2 of Lu (2022)). Given $\epsilon > 0$, a point $\mathbf{x}^\star$ is called an $\epsilon$-*KKT solution* for problem equation 1 if there exists $\boldsymbol{\nu}^\star \in \mathbb{R}_+^m$ such that

$$\begin{cases} \boldsymbol{v}^\star \in \nabla f(\mathbf{x}^\star) + \partial r(\mathbf{x}^\star) + J_g(\mathbf{x}^\star)\boldsymbol{\nu}^\star, \quad \|\boldsymbol{v}^\star\| \leq \epsilon, \\ \|\max\{0, g(\mathbf{x}^\star)\}\| \leq \epsilon, \quad \sum_{j=1}^m |\nu_j g_j(\mathbf{x}^\star)| \leq \epsilon, \end{cases}$$

where $\max\{\mathbf{0}, g(\mathbf{x}^\star)\}$ denotes the component-wise maximum of $g(\mathbf{x}^\star)$ and the zero vector $\mathbf{0}$ at $\mathbf{x}^\star$.

To establish the ensuing analysis, we introduce the following standard assumptions for problem equation 1:

**Assumption 3.** There exists a point $(\mathbf{x}, \boldsymbol{\nu}) \in \text{dom}(r) \times \mathbb{R}^m$ satisfying the KKT conditions equation 2.

**Assumption 4.** $\nabla f$ is $L_f$-Lipschitz continuous on $\text{dom}(r)$. That is, there exist $L_f > 0$ such that

$$\|\nabla f(\mathbf{x}) - \nabla f(\mathbf{x}')\| \leq L_f \|\mathbf{x} - \mathbf{x}'\|, \ \forall \mathbf{x}, \mathbf{x}' \in \text{dom}(r),$$

**Assumption 5.** $\nabla g$ is $L_g$-Lipschitz continuous on $\text{dom}(r)$. That is, there exist $L_g > 0$ such that

$$\|\nabla g(\mathbf{x}) - \nabla g(\mathbf{x}')\| \leq L_g \|\mathbf{x} - \mathbf{x}'\|, \ \forall \mathbf{x}, \mathbf{x}' \in \text{dom}(r).$$

**Assumption 6.** The domain of $r$ is compact, i.e., $D_{\mathbf{x}} := \max_{\mathbf{x}, \mathbf{x}' \in \text{dom}(r)} \|\mathbf{x} - \mathbf{x}'\| < +\infty$.

**Assumption 7** (Section 3 of Pillo et al. (1980); Section 6 of Di Pillo & Lucidi (2002); Section 4.3 of Bertsekas (2014)). The iterates $\{\boldsymbol{\lambda}_k\}$ generated by iterative methods for problem equation 1 estimating $\boldsymbol{\nu}^*$ satisfying Definition 1 are contained in a convex compact subset $\Lambda \subset \mathbb{R}^m$.

These assumptions are considered standard in the optimization literature Boob et al. (2022); Huang & Lin (2023) and are satisfied by a broad range of practical problems in signal processing and machine learning Bolte et al. (2018); Li & Xu (2021); Li et al. (2021); Lu (2022); Kong et al. (2022); Lu & Zhou (2023). Our work distinguishes itself by not requiring certain restrictive assumptions beyond those stated above, such as the surjectivity of $\nabla g(\mathbf{x})$ (or $\nabla g(\mathbf{x})\nabla g(\mathbf{x})^\top$ being positive definite) Bolte et al. (2018); Boţ et al. (2019); Boţ & Nguyen (2020); Li & Pong (2015), Slater's condition Boob et al. (2022); Kong et al. (2022), or more crucially feasibility of initialization Boob et al. (2022); Hajinezhad & Hong (2019); Xie & Wright (2021) as that by itself is a non-convex problem when $g$ is nonconvex. For problems with an unbounded $\text{dom}(r)$, they can be reformulated to satisfy Assumption 6; for instance, if $f$ is bounded below and $r$ is coercive, the problem can be transformed to one with $f + r$ for some $r$ (e.g., norm functions) with a compact domain Lu & Zhou (2023). Notably, this can be implemented in practice for machine learning problem using the standard practice of weight or gradient clipping. Moreover, Assumption 7 is commonly used in the convergence analysis of constrained optimization algorithms Nocedal & Wright (2006); Bertsekas (2014); Birgin & Martínez (2014); Hong et al. (2016); Pillo et al. (1980); Di Pillo & Lucidi (2002). From certain constraint qualification, such as MFCQ or CPLD, it can also be derived that the set of KKT multipliers corresponding to a local minimum is bounded.

Furthermore, under Assumption 6, there exist constants $B_g > 0$ and $M_g > 0$ such that

$$\max_{\mathbf{x} \in \text{dom}(r)} \|g(\mathbf{x})\| \leq B_g \quad \text{and} \quad \max_{\mathbf{x} \in \text{dom}(r)} \|\nabla g(\mathbf{x})\| \leq M_g, \tag{3}$$

which implies the Lipschitz continuity of $g$ (Rockafellar & Wets, 2009, Chapter 9.B): $\|g(\mathbf{x}) - g(\mathbf{x}')\| \leq M_g \|\mathbf{x} - \mathbf{x}'\|, \quad \forall \mathbf{x}, \mathbf{x}' \in \mathrm{dom}(r)$.

In the next section, we consider a non-convex optimization problem with non-smooth functional constraint. In this case, Assumption 8 implies the Lipschitz continuity of the subgradient, instead of equation 3 or Assumption 5.

**Assumption 8.** $g$ is continuous with $\partial g(\mathbf{x}) \neq \emptyset$ on $\mathrm{dom}(r)$, and there exists a constant $M_g > 0$ such that $\max_{\mathbf{x} \in \mathrm{dom}(r)} \|\partial g(\mathbf{x})\| \leq M_g$.

Finally, to establish non-asymptotic convergence rates, we need a local error bound condition as in Assumption 9, relating the distance to the solution set to the stationarity residual. It holds for broad classes of structured non-convex functions under mild assumptions and is commonly used in convergence analysis in non-convex non-smooth optimization Drusvyatskiy & Lewis (2018); Davis et al. (2018).

**Assumption 9.** There exists a constant $\kappa > 0$ and a neighborhood $U$ of the stationary set $X^*$ such that $\mathrm{dist}(x, X^*) \leq \kappa \cdot \mathrm{dist}(0, \partial f(x))$ for all $x \in U$.

## 3 Non-convex Non-smooth Constraints

In this section, we consider the non-convex optimization problem equation 1 with a non-smooth functional constraint $g(\cdot)$. We first introduce a novel Lagrangian with a structure designed for developing an efficient algorithm that solves the non-smooth constrained problem. A critical feature of the resulting algorithm is its reliance on fixed parameters $\alpha$, $\beta$ and $\rho = \alpha/(1 + \alpha\beta))$, which eliminates the need for the sensitive and iterative adjustments required by many existing schemes. This design not only simplifies implementation but also enhances computational efficiency. Empirical results demonstrate that the algorithm's performance is not sensitive to the choice of $\alpha$ and $\beta$, further highlighting its robustness.

### 3.1 A Variant of Proximal-Perturbed Lagrangian

Motivated by the reformulation techniques in Bertsekas (1999; 2014), we employ *perturbation* variables $\mathbf{z} \in \mathbb{R}^m$ and slack variables $\mathbf{u} \in \mathbb{R}^m_+$. By setting $g(\mathbf{x}) + \mathbf{u} = \mathbf{z}$ and $\mathbf{z} = \mathbf{0}$, we first transform problem equation 1 into an equivalent equality-constrained formulation:

$$\min_{\mathbf{x} \in \mathbb{R}^n, \mathbf{u} \in \mathbb{R}^m_+, \mathbf{z} \in \mathbb{R}^m} f(\mathbf{x}) + r(\mathbf{x}) \quad \text{s.t.} \quad g(\mathbf{x}) + \mathbf{u} = \mathbf{z}, \quad \mathbf{z} = \mathbf{0}. \tag{4}$$

Clearly, for $\mathbf{z}^* = \mathbf{0}$ and $\mathbf{u}^* \geq 0$, the extended formulation equation 4 is equivalent to problem equation 1.

For this formulation equation 4, we define a variant of *the Proximal-Perturbed Lagrangian* (P-Lagrangian) from Kim (2021) as follows:

$$\mathcal{L}_{\alpha\beta}(\mathbf{x}, \mathbf{u}, \mathbf{z}, \boldsymbol{\lambda}, \boldsymbol{\mu}) := f(\mathbf{x}) + \langle \boldsymbol{\lambda}, g(\mathbf{x}) + \mathbf{u} - \mathbf{z} \rangle + \langle \boldsymbol{\mu}, \mathbf{z} \rangle + \frac{\alpha}{2} \|\mathbf{z}\|^2 - \frac{\beta}{2} \|\boldsymbol{\lambda} - \boldsymbol{\mu}\|^2 + r(\mathbf{x}), \tag{5}$$

where $\boldsymbol{\lambda} \in \mathbb{R}^m$ is a multiplier (dual) for the constraint $g(\mathbf{x}) + \mathbf{u} - \mathbf{z} = 0$, $\boldsymbol{\mu} \in \mathbb{R}^m$ is an *auxiliary multiplier* for the constraint $\mathbf{z} = \mathbf{0}$, $\alpha > 0$ is a penalty parameter, and $\beta > 0$ is a proximal parameter.

Given $(\boldsymbol{\lambda}, \boldsymbol{\mu})$, $\mathcal{L}_{\alpha\beta}$ can be minimized with respect to $\mathbf{z}$ in closed form:

$$\mathbf{z}(\boldsymbol{\lambda}, \boldsymbol{\mu}) = (\boldsymbol{\lambda} - \boldsymbol{\mu})/\alpha. \tag{6}$$

Substituting $\mathbf{z}(\boldsymbol{\lambda}, \boldsymbol{\mu})$ back into $\mathcal{L}_{\alpha\beta}$ yields the reduced P-Lagrangian:

$$\mathcal{L}_{\alpha\beta}(\mathbf{x}, \mathbf{u}, \mathbf{z}(\boldsymbol{\lambda}, \boldsymbol{\mu}), \boldsymbol{\lambda}, \boldsymbol{\mu}) = f(\mathbf{x}) + \langle \boldsymbol{\lambda}, g(\mathbf{x}) + \mathbf{u} \rangle - \frac{1}{2\rho} \|\boldsymbol{\lambda} - \boldsymbol{\mu}\|^2 + r(\mathbf{x}), \tag{7}$$

where $\rho := \frac{\alpha}{1 + \alpha\beta}$. Note that equation 7 is $\frac{1}{\rho}$-strongly concave in $\boldsymbol{\lambda}$ for a fixed $\boldsymbol{\mu}$. This property guarantees a unique maximizer for $\boldsymbol{\lambda}$, which can be found in closed form:

$$\boldsymbol{\lambda}(\mathbf{x}, \boldsymbol{\mu}) = \underset{\boldsymbol{\lambda} \in \mathbb{R}^m}{\mathrm{argmax}}\ \mathcal{L}_{\alpha\beta}(\mathbf{x}, \mathbf{u}, \mathbf{z}(\boldsymbol{\lambda}, \boldsymbol{\mu}), \boldsymbol{\lambda}, \boldsymbol{\mu}) = \boldsymbol{\mu} + \rho(g(\mathbf{x}) + \mathbf{u}), \tag{8}$$

which is well-defined and is used for the update of $\boldsymbol{\lambda}_{k+1}$ in equation 13.

---

**Algorithm 1** P-Lagrangian based Alternating Direction Algorithm (PLADA)

---

1: **Input:** Initialization $(\mathbf{x}_0, \mathbf{u}_0, \mathbf{z}_0, \boldsymbol{\lambda}_0, \boldsymbol{\mu}_0)$, and parameters $\alpha > 1$, $\beta \in (0, 1)$, $\rho = \frac{\alpha}{1+\alpha\beta}$, $0 < \eta < \frac{1}{L_f + 3\rho M_g^2}$,
$0 < \tau < \frac{1}{3\rho}$, $\delta_0 \in (0, 1]$, and $K$.
2: **for** $k = 0, 1, \ldots, K$ **do**
3:     $\mathbf{x}_{k+1} = \underset{\mathbf{x} \in \mathbb{R}^n}{\operatorname{argmin}} \left\{ \langle \nabla f(\mathbf{x}_k), \mathbf{x} \rangle + \langle \boldsymbol{\lambda}_k, g(\mathbf{x}) \rangle + 1/2\eta \|\mathbf{x} - \mathbf{x}_k\|^2 + r(\mathbf{x}) \right\}$;
4:     $\mathbf{u}_{k+1} = \Pi_{\mathbb{R}_+^m}[\mathbf{u}_k - \tau\boldsymbol{\lambda}_k]$;
5:     $\boldsymbol{\mu}_{k+1} = \boldsymbol{\mu}_k + \frac{\sigma_k}{\rho}(\boldsymbol{\lambda}_k - \boldsymbol{\mu}_k)$, $\sigma_k = \min \left\{ \sigma_0, \frac{\rho\delta_k}{\|\boldsymbol{\lambda}_k - \boldsymbol{\mu}_k\|^2 + 1} \right\}$;
6:     $\boldsymbol{\lambda}_{k+1} = \boldsymbol{\mu}_{k+1} + \rho(g(\mathbf{x}_{k+1}) + \mathbf{u}_{k+1})$;
7: **end for**

---

## 3.2 Description of Algorithm

Based on the P-Lagrangian, we propose the P-Lagrangian based Alternating Direction Algorithm (PLADA) for solving problem equation 1 with non-smooth constraints. The complete procedure is detailed in Algorithm 1.

Each iteration of PLADA consists of a sequence of updates for the primal, dual, and auxiliary variables. The primal variable $\mathbf{x}$ is updated using a proximal gradient step:

$$\mathbf{x}_{k+1} = \underset{\mathbf{x} \in \mathbb{R}^n}{\operatorname{argmin}} \left\{ \langle \nabla f(\mathbf{x}_k), \mathbf{x} \rangle + \langle \boldsymbol{\lambda}_k, g(\mathbf{x}) \rangle + 1/2\eta \|\mathbf{x} - \mathbf{x}_k\|^2 + r(\mathbf{x}) \right\}. \tag{9}$$

The slack variable $\mathbf{u}$ is updated via projected gradient descent onto $\mathbb{R}_+^m$:

$$\mathbf{u}_{k+1} = \underset{\mathbf{u} \in \mathbb{R}_+^m}{\operatorname{argmin}} \left\{ \langle \nabla_{\mathbf{u}} \mathcal{L}_{\alpha\beta}(\mathbf{x}_k, \mathbf{u}_k, \mathbf{z}_k, \boldsymbol{\lambda}_k, \boldsymbol{\mu}_k), \mathbf{u} - \mathbf{u}_k \rangle + 1/2\tau \|\mathbf{u} - \mathbf{u}_k\|^2 \right\} = \Pi_{\mathbb{R}_+^m}[\mathbf{u}_k - \tau\boldsymbol{\lambda}_k], \tag{10}$$

where, without loss of generality, we can construct an upper bound $\max_{k \geq 1}\{\mathbf{u}_k\} \leq B_g$ as $\|g(\mathbf{x})\| \leq B_g$.

The auxiliary multiplier $\boldsymbol{\mu}$ is updated using a gradient ascent step on equation 7:

$$\boldsymbol{\mu}_{k+1} = \boldsymbol{\mu}_k + \sigma_k(\mathbf{z}_k + \beta(\boldsymbol{\lambda}_k - \boldsymbol{\mu}_k)) = \boldsymbol{\mu}_k + \frac{\sigma_k}{\rho}(\boldsymbol{\lambda}_k - \boldsymbol{\mu}_k), \tag{11}$$

with a diminishing step-size $\sigma_k = \min\left\{\sigma_0, \rho\delta_k/(\|\boldsymbol{\lambda}_k - \boldsymbol{\mu}_k\|^2 + 1)\right\}$, which is governed by a sequence $\delta_k > 0$ that satisfies the standard conditions:

$$\delta_0 \in (0, 1], \quad \lim_{k \to \infty} \delta_k = 0, \quad \text{and} \quad \sum_{k=0}^{\infty} \delta_k = +\infty. \tag{12}$$

In Algorithm 1, we choose $\delta_k = \kappa \cdot (k+1)^{-1}$ with $\kappa > 0$, so that these conditions hold.

Finally, the main dual variable $\boldsymbol{\lambda}$ is updated via an exact maximization on equation 7:

$$\boldsymbol{\lambda}_{k+1} = \boldsymbol{\mu}_{k+1} + \rho(g(\mathbf{x}_{k+1}) + \mathbf{u}_{k+1}). \tag{13}$$

A key advantage of this framework is that the parameters $\alpha, \beta$, and the dual step size $\rho$ are constants, independent of the number of iterations $k$. In Section I, we demonstrate the robustness of the algorithm with respect to the choices of $\alpha$ and $\beta$.

Noting that solving the subproblem in equation 9 globally is intractable, the primary purpose of Algorithm 1 is to establish theoretical baseline for the proposed P-Lagrangian framework under non-smooth constraints. Thus, the convergence rates shown in the following subsection, i.e., Lemmas 12-16 and Theorem 21, are with respect to the outer loop iterations, assuming that the subproblem is solved to global optimality. Later, we introduce Algorithm 2 in Section 4 that employs fully tractable proximal gradient update under smooth constraint settings.

### 3.3 Convergence Guarantees

This subsection establishes the convergence guarantees for Algorithm 1. Our analysis begins by defining the necessary concepts from subdifferential calculus for non-smooth functions. We then present a series of technical lemmas that establish convergence of the algorithm's iterates with respect to the Definitions 1 and 2. These results culminate in theorems proving the algorithm's asymptotic convergence as well as the non-asymptotic rate of convergence on expectation. All proofs are contained in Supplementary Material for concise main paper.

We first recall some essential definitions from variational analysis. We denote the Jacobian matrix of $g$ at $\mathbf{x}$ by $\partial g(\mathbf{x})$. For any set $\mathcal{X} \subseteq \mathbb{R}^d$, its indicator function $\mathbb{I}_\mathcal{X}$ is defined by $\mathbb{I}_\mathcal{X} = 0$ if $\mathbf{x} \in \mathcal{X}$ and $+\infty$, otherwise. Note that $\arg\min_{\mathbf{x} \in \mathcal{X}} F(\mathbf{x}) = \arg\min_{\mathbf{x} \in \mathbb{R}^d} \{\varphi(\mathbf{x}) := F(\mathbf{x}) + \mathbb{I}_\mathcal{X}(\mathbf{x})\}$.

**Definition 10** (Definition 8.3 of Rockafellar & Wets (2009))**.** Let $g_i : \mathbb{R}^d \to \mathbb{R} \cup \{+\infty\}$ be a proper and lower semicontinuous function. For each $\mathbf{x} \in \mathcal{X}$, the *Frechet subdifferential* of $g$ of $\mathbf{x}$ is given by

$$\widehat{\partial} g_i(\mathbf{x}) := \left\{ d_k \in \mathbb{R}^d : \liminf_{w \to \mathbf{x}} \frac{g_i(w) - g_i(\mathbf{x}) - \langle d, w - \mathbf{x} \rangle}{\|w - \mathbf{x}\|} \geq 0 \right\}.$$

**Definition 11.** The *limiting subdifferencial* (or simply the subdifferential) of $g_i$ at $\mathbf{x} \in \mathbb{R}^d$ is defined as

$$\partial g_i(\mathbf{x}) := \left\{ d \in \mathbb{R}^d : \exists \mathbf{x}_k \to \mathbf{x} \text{ and } d_k \in \widehat{\partial} g_i(\mathbf{x}_k) \text{ with } d_k \to d \text{ as } k \to \infty \right\}.$$

The inclusion $\widehat{\partial} g_i(\mathbf{x}) \subseteq \partial g_i(\mathbf{x})$ holds for each $\mathbf{x} \in \mathcal{X}$ and we set $\widehat{\partial} g_i(\mathbf{x}) = \partial g_i(\mathbf{x}) = \emptyset$ for $\mathbf{x} \notin \mathcal{X}$. Each $d \in \partial g_i(\mathbf{x})$ is called a subgradient of $g_i$ at $\mathbf{x}$.

We now present the main convergence theorems for Algorithm 1, establishing the asymptotic convergence to a KKT solution as defined in Definition 1.

**Lemma 12** (Primal Stationarity)**.** *Let $\{\mathbf{w}_k\}$ be the sequence generated by Algorithm 1, and let $\{\mathbf{p}_k := (\mathbf{x}_k, \mathbf{u}_k, \mathbf{z}_k)\}$ be the primal sequence. Under Assumptions 3, 4, 6 and 8, the running averaged of the squared primal stationarity residual converges to zero:*

$$\lim_{T \to \infty} \frac{1}{T} \sum_{k=0}^{T-1} \|\boldsymbol{\zeta}_\mathbf{p}^{k+1}\|^2 = 0, \quad \text{with the rate of } \mathcal{O}\left(\frac{\log(T)}{T}\right) = \tilde{\mathcal{O}}\left(\frac{1}{T}\right),$$

*where $\boldsymbol{\zeta}_\mathbf{p}^{k+1} := (\boldsymbol{\zeta}_\mathbf{x}^{k+1}, \boldsymbol{\zeta}_\mathbf{u}^{k+1}, \boldsymbol{\zeta}_\mathbf{z}^{k+1}) \in \partial_\mathbf{p} \mathcal{L}_{\alpha\beta}(\mathbf{w}_{k+1})$. Hence, any limit point $(\bar{\mathbf{x}}, \bar{\boldsymbol{\lambda}})$ of the sequence $(\mathbf{x}_k, \boldsymbol{\lambda}_k)$ satisfies the stationarity condition of the original problem: $\mathbf{0} \in \nabla f(\bar{\mathbf{x}}) + \partial r(\bar{\mathbf{x}}) + \partial g(\bar{\mathbf{x}})^\top \bar{\boldsymbol{\lambda}}$.*

Lemma 12 establishes that the primal iterates in an ergodic sense. The running-average of the squared stationarity residual (first-order optimality) converges to zero at a rate of $\tilde{\mathcal{O}}(1/T)$[3]:

$$\frac{1}{T} \sum_{k=0}^{T-1} \|\boldsymbol{\zeta}_\mathbf{p}^{k+1}\|^2 = \mathcal{O}\left(\frac{\log(T)}{T}\right) = \tilde{\mathcal{O}}\left(\frac{1}{T}\right).$$

*Remark* 13. An immediate consequence of Lemma 34 and 12 is that the squared successive difference of the primal iterates also converges at the same rate:

$$\frac{1}{T} \sum_{k=0}^{T} \left( \|\mathbf{x}_{k+1} - \mathbf{x}_k\|^2 + \|\mathbf{u}_{k+1} - \mathbf{u}_k\|^2 \right) = \tilde{\mathcal{O}}\left(\frac{1}{T}\right).$$

Note that Lemma 12 states the convergence in an ergodic sense, which involves averaging over the sequence of iterates or employing a randomized output selection from $T$ iterates. Thus, the primal iterates converge with $\tilde{\mathcal{O}}(1/T)$ in an ergodic sense.

---

[3]The notation $\tilde{\mathcal{O}}(\cdot)$ suppresses all logarithmic factors from the big-$\mathcal{O}$ notation.

Next, we establish that the iterates converge to a feasible point. This is achieved by showing that the gap between the dual variables vanishes as $k \to \infty$: $\lim_{k \to \infty} \|\boldsymbol{\lambda}_k - \boldsymbol{\mu}_k\| = 0$.

**Lemma 14** (Primal Feasibility). *Let $\{\mathbf{w}_k\}$ be the sequence generated by Algorithm 1. Under Assumptions 3, 4, 6, 7 and 8, the gap between the dual variables vanishes:*

$$\lim_{k \to \infty} \|\boldsymbol{\lambda}_k - \boldsymbol{\mu}_k\| = 0.$$

*Consequently, any limit point $\bar{\mathbf{x}}$ of the sequence $\{\mathbf{x}_k\}$ is feasible for problem equation 1, satisfying $g(\bar{\mathbf{x}}) \leq 0$.*

**Lemma 15** (Dual feasibility). *Let $\bar{\boldsymbol{\lambda}}$ be a limit point of the sequence $\{\boldsymbol{\lambda}_k\}$ generated by Algorithm 1. Then, $\bar{\boldsymbol{\lambda}}$ is feasible for the dual problem of equation 1, satisfying $\bar{\boldsymbol{\lambda}} \geq \mathbf{0}$.*

**Lemma 16** (Complementary slackness). *Let $(\bar{\mathbf{x}}, \bar{\boldsymbol{\lambda}})$ be a limit point of the sequence $\{(\mathbf{x}_k, \boldsymbol{\lambda}_k)\}$ generated by Algorithm 1. Then, $(\bar{\mathbf{x}}, \bar{\boldsymbol{\lambda}})$ satisfies the complementary slackness for problem of equation 1, i.e., $\bar{\boldsymbol{\lambda}}^\top g(\bar{\mathbf{x}}) = 0$.*

**Theorem 17** (Convergence to a KKT Point). *Let $\{\mathbf{w}_k = (\mathbf{x}_k, \mathbf{u}_k, \mathbf{z}_k, \boldsymbol{\lambda}_k, \boldsymbol{\mu}_k)\}$ be the sequence generated by Algorithm 1. Under Assumptions 3, 4, 6, 7 and 8, any limit point $\bar{\mathbf{w}}$ of the sequence $\{\mathbf{w}_k\}$ corresponds to a KKT point of the original problem equation 1 as defined in Definition 1.*

*Proof.* By Lemmas 12, 14, 15 and 16, $\bar{\mathbf{w}}$ satisfies the KKT conditions as defined in Definition 1. □

To find the non-asymptotic ergodic rate of convergence, we construct a non-negative auxiliary sequence as

$$\boldsymbol{\nu}_k := \boldsymbol{\lambda}_k + \frac{1}{\tau}(\mathbf{u}_{k+1} - \mathbf{u}_k). \tag{14}$$

Note that by the first-order optimality of $\mathbf{u}_{k+1}$ for equation 10,

$$\mathbf{u}_k - \tau\boldsymbol{\lambda}_k \leq \mathbf{u}_{k+1} \iff \boldsymbol{\lambda}_k + \frac{1}{\tau}(\mathbf{u}_{k+1} - \mathbf{u}_k) \geq \mathbf{0},$$

and $\boldsymbol{\nu}_k \geq \mathbf{0}$ for all $k \geq 0$. To show the ergodic convergence to a $\epsilon$-KKT solution, define the running average of this non-negative multiplier: $\bar{\boldsymbol{\nu}}_T := \frac{1}{T} \sum_{k=0}^{T-1} \boldsymbol{\nu}_k$.

**Lemma 18** (Non-asymptotic rate for primal stationarity). *Let $\{\mathbf{p}_k := (\mathbf{x}_k, \mathbf{u}_k, \mathbf{z}_k)\}$ be the primal sequence generated by Algorithm 1 using the non-negative multiplier $\bar{\boldsymbol{\nu}}_T$. Under Assumptions 3, 4, 6 and 8, average primal stationarity residual converges as*

$$\frac{1}{T} \sum_{k=0}^{T-1} \|\boldsymbol{\zeta}_{\mathbf{p}}^{k+1}\|^2 = \tilde{\mathcal{O}}\left(\frac{1}{T}\right),$$

*where $\boldsymbol{\zeta}_{\mathbf{p}}^{k+1} := (\boldsymbol{\zeta}_{\mathbf{x}}^{k+1}, \boldsymbol{\zeta}_{\mathbf{u}}^{k+1}, \boldsymbol{\zeta}_{\mathbf{z}}^{k+1}) \in \partial_{\mathbf{p}}\mathcal{L}_{\alpha\beta}(\mathbf{w}_{k+1})$.*

**Lemma 19** (Non-asymptotic rate for primal feasibility). *Let $\{\mathbf{x}_k\}$ be the primal sequence generated by Algorithm 1. Under Assumptions 3, 4, 6, 7 and 8, average primal feasibility violation converges as*

$$\frac{1}{T} \sum_{k=0}^{T-1} \|[g(\mathbf{x}_{k+1})]^+\|^2 = \tilde{\mathcal{O}}\left(\frac{1}{T}\right). \tag{15}$$

**Lemma 20** (Non-asymptotic rate for complementary slackness). *Let $\{\mathbf{x}_k, \boldsymbol{\nu}_k\}$ be the sequence generated by Algorithm 1. Under Assumptions 3, 4, 6, 7 and 8, average complementary slackness for Definition 2 converges as*

$$\frac{1}{T} \sum_{k=0}^{T-1} \sum_{j=1}^{m} |\nu_{j,k} g_j(\mathbf{x}_{k+1})| = \tilde{\mathcal{O}}(1/\sqrt{T}). \tag{16}$$

**Theorem 21** (Non-asymptotic Rate of Convergence). *Under Assumptions 3, 4, 6, 7 and 8, there exists a uniformly-at-random iterate $k \in \{0, \cdots, K-1\}$ from the sequence generated by Algorithm 1 that is a $\epsilon$-KKT solution to problem 1 on expectation as defined in Definition 2. The total number of iterations required to achieve this is bounded by $\tilde{\mathcal{O}}(1/\epsilon^2)$.*

*Proof.* By Lemmas 18, 19 and 20 and the construction of the non-negative multiplier sequence, we have the ergodic convergence of $\epsilon$-KKT residuals at a rate of $\tilde{\mathcal{O}}(1/\sqrt{T})$. Hence, all conditions for an $\epsilon$-KKT solution are met with an overall iteration complexity of $\tilde{\mathcal{O}}(1/\epsilon^2)$. Therefore, if one chooses one of the algorithm iterates uniformly at random, that solution will be $\epsilon$-KKT on expectation. □

# 4 Non-convex Continuously Differentiable Constraints

In this section, we present our novel form of augmented Lagrangian (Section 4.1), termed Proximal-Perturbed Augmented Lagrangian (PPAL), and propose a single-loop primal-dual algorithm based on it (Section 4.2).

## 4.1 Proximal-Perturbed Augmented Lagrangian

Our approach for the smooth case is built upon *Proximal-Perturbed Augmented Lagrangian* (PPAL). As in the non-smooth case, we work with the equivalent equality-constrained formulation of problem equation 1 and define the PPAL as:

$$\mathcal{L}_\rho(\mathbf{x}, \mathbf{u}, \mathbf{z}, \boldsymbol{\lambda}, \boldsymbol{\mu}) = \ell_\rho(\mathbf{x}, \mathbf{u}, \mathbf{z}, \boldsymbol{\lambda}, \boldsymbol{\mu}) + r(\mathbf{x}), \tag{17}$$

where

$$\ell_\rho(\cdot) := f(\mathbf{x}) + \langle \boldsymbol{\lambda}, g(\mathbf{x}) + \mathbf{u} - \mathbf{z} \rangle + \langle \boldsymbol{\mu}, \mathbf{z} \rangle + \frac{\alpha}{2}\|\mathbf{z}\|^2 - \frac{\beta}{2}\|\boldsymbol{\lambda} - \boldsymbol{\mu}\|^2 + \frac{\rho}{2}\|g(\mathbf{x}) + \mathbf{u}\|^2. \tag{18}$$

Analogous to equation 7, substituting the expression for $\mathbf{z}(\boldsymbol{\lambda}, \boldsymbol{\mu})$ back into equation 17 yields the reduced PPAL:

$$\mathcal{L}_\rho(\mathbf{x}, \mathbf{u}, \mathbf{z}(\boldsymbol{\lambda}, \boldsymbol{\mu}), \boldsymbol{\lambda}, \boldsymbol{\mu}) = f(\mathbf{x}) + \langle \boldsymbol{\lambda}, g(\mathbf{x}) + \mathbf{u} \rangle - \frac{1}{2\rho}\|\boldsymbol{\lambda} - \boldsymbol{\mu}\|^2 + \frac{\rho}{2}\|g(\mathbf{x}) + \mathbf{u}\|^2 + r(\mathbf{x}). \tag{19}$$

While sharing structural similarities, Kim (2021) addresses Generalized Nash Equilibrium Problems using computationally intensive inner loops to solve $n$ fixed-point subproblems. In contrast, our framework targets single-objective non-convex optimization, offering a single efficient step for smooth constraints. Furthermore, our method accelerates dual convergence via proximal-like dual updates and extends rigorous analysis to non-smooth constraints, whereas Kim (2021) effectively reduces to a standard AL step with slow outer-loop convergence due to the initialization $\boldsymbol{\lambda}_0 = \boldsymbol{\mu}_0$.

## 4.2 Description of Algorithm

We propose a single-loop first-order algorithm based on the properties of our PPAL, which computes a stationary solution to problem equation 1. The complete procedure is detailed in Algorithm 2, where $L_\ell := L_f + L_g B_\lambda + \rho(L_g B_u + L_g B_g + M_g^2)$.

Each iteration of PPALA involves updating the primal and dual variables. The primal variable $\mathbf{x}$ is updated inexactly by the *proximal gradient mapping* (see e.g., Bolte et al. (2014)), which can be rewritten as

$$\mathbf{x}_{k+1} = \text{prox}_{\eta r}\left[\mathbf{x}_k - \eta\nabla_{\mathbf{x}}\ell_\rho(\mathbf{x}_k, \mathbf{u}_k, \mathbf{z}_k, \boldsymbol{\lambda}_k, \boldsymbol{\mu}_k)\right]. \tag{22}$$

Next, the slack variable $\mathbf{u}$ is upated via projected gradient descent:

$$\begin{aligned} \mathbf{u}_{k+1} &= \Pi_{\mathbb{R}_+^m}[\mathbf{u}_k - \tau(\nabla_{\mathbf{u}}\mathcal{L}_\rho(\mathbf{x}_k, \mathbf{u}_k, \mathbf{z}_k, \boldsymbol{\lambda}_k, \boldsymbol{\mu}_k)] \\ &= \Pi_{\mathbb{R}_+^m}[\mathbf{u}_k - \tau(\boldsymbol{\lambda}_k + \rho(g(\mathbf{x}_{k+1}) + \mathbf{u}_k)]. \end{aligned}$$

Note that we can construct an upper bound $\max_{k\geq 1}\{\mathbf{u}_k\} \leq B_g$ from equation 3, since we have $\|g(\mathbf{x})\| \leq B_g$ for all feasible solutions $\mathbf{x}$.

The *auxiliary* multiplier $\boldsymbol{\mu}$ is updated as equation 21 with a diminishing sequence $\delta_k$ satisfying the conditions equation 12. In particular, we employ the form:

$$\delta_k = \frac{1}{p \cdot k^q + 1}, \quad \frac{2}{3} < q \leq 1, \quad p > 0. \tag{23}$$

---
**Algorithm 2** PPAL-based first-order Algorithm (PPALA)
---
1: **Input:** Initialization $(\mathbf{x}_0, \mathbf{u}_0, \mathbf{z}_0, \boldsymbol{\lambda}_0, \boldsymbol{\mu}_0)$, and parameters $\alpha > 1$, $\beta \in (0,1)$, $\rho = \frac{\alpha}{1+\alpha\beta}$, $0 < \eta < \frac{1}{L_\ell + 3\rho M_g^2}$,
   $0 < \tau < \frac{1}{2\rho}$, and $K$.

2: **for** $k = 0, 1, \ldots, K$ **do**

3:     Compute $\mathbf{x}_{k+1}$ by the proximal gradient scheme:

$$\mathbf{x}_{k+1} = \operatorname*{argmin}_{\mathbf{x} \in \mathbb{R}^n} \left\{ \langle \nabla_{\mathbf{x}} \ell_\rho(\mathbf{x}_k, \mathbf{u}_k, \mathbf{z}_k, \boldsymbol{\lambda}_k, \boldsymbol{\mu}_k), \mathbf{x} - \mathbf{x}_k \rangle + (1/2\eta)\|\mathbf{x} - \mathbf{x}_k\|^2 + r(\mathbf{x}) \right\};$$

4:     Compute $\mathbf{u}_{k+1}$ by the projected gradient descent:

$$\mathbf{u}_{k+1} = \Pi_{\mathbb{R}^m_+}[\mathbf{u}_k - \tau(\boldsymbol{\lambda}_k + \rho(g(\mathbf{x}_{k+1}) + \mathbf{u}_k))]; \tag{20}$$

5:     Update the auxiliary multiplier $\boldsymbol{\mu}_{k+1}$ by:

$$\boldsymbol{\mu}_{k+1} = \boldsymbol{\mu}_k + \sigma_k(\boldsymbol{\lambda}_k - \boldsymbol{\mu}_k), \ \sigma_k = \frac{\delta_k}{\|\boldsymbol{\lambda}_k - \boldsymbol{\mu}_k\|^2 + 1}; \tag{21}$$

6:     Update the multiplier $\boldsymbol{\lambda}_{k+1}$ by

$$\boldsymbol{\lambda}_{k+1} = \boldsymbol{\mu}_{k+1} + \rho(g(\mathbf{x}_{k+1}) + \mathbf{u}_{k+1});$$

7: **end for**
---

Note that several alternatives are available for the sequence $\{\delta_k\}$ satisfying the conditions in equation 12. Two popular alternative step sizes are: (i) $\delta_k = \frac{\delta_0}{(k+1)^q}$, where $\delta_0 > 0$ and $0 < q \leq 1$, and (ii) $\delta_k = \frac{\delta_{k-1}}{1 - b\delta_{k-1}}$, where $\delta_0 \in (0,1]$ and $b \in (0,1)$; see e.g., Bertsekas (1999); Scutari et al. (2014) for more possibilities for $\{\delta_k\}$. As we will see in Lemma 22 and Corollary 23, a benefit of equation 23 and choosing $q \in (2/3, 1]$ is that it allows our algorithm to achieve improved complexity bounds compared to $\mathcal{O}(1/\epsilon^3)$ found in existing works.

Then the multiplier $\boldsymbol{\lambda}$ is updated in the same manner as Algorithm 1.

### 4.3 Convergence Guarantees

In this section, we establish the convergence results of Algorithm 2. We prove that the sequence generated by Algorithm 2 converges to a KKT point of problem equation 1 as defined in equation 2. The analysis extends to demonstrating the algorithm's non-asymptotic rate of convergence in ergodic sense. Please find the proofs of each Lemma in Sumpplementary Materials.

**Lemma 22** (Primal Stationarity). *Let $\{\mathbf{w}_k\}$ be the sequence generated by Algorithm 2, and let $\{\mathbf{p}_k := (\mathbf{x}_k, \mathbf{u}_k, \mathbf{z}_k)\}$ be the generated primal sequence. Under Assumptions 3-7, the running average of the squared primal stationarity residual converges to zero:*

$$\lim_{T \to \infty} \frac{1}{T} \sum_{k=0}^{T-1} \|\boldsymbol{\zeta}_{\mathbf{p}}^{k+1}\|^2 = 0, \quad \text{with the rate of} \begin{cases} \mathcal{O}\left(\frac{\log(T)}{T}\right) = \widetilde{\mathcal{O}}\left(\frac{1}{T}\right) & \text{if } q = 1, \\ \mathcal{O}\left(\frac{1}{T^q}\right) & \text{if } 2/3 < q < 1, \end{cases} \tag{24}$$

*where $\boldsymbol{\zeta}_{\mathbf{p}}^{k+1} \in \partial_{\mathbf{p}} \mathcal{L}_\rho(\mathbf{w}_{k+1})$ and $\delta_k = \frac{1}{p \cdot k^q + 1}$. Hence, $\mathbf{0} \in \nabla f(\bar{\mathbf{x}}) + \partial r(\bar{\mathbf{x}}) + \partial g(\bar{\mathbf{x}})^\top \bar{\boldsymbol{\lambda}}$.*

Thus, a consequence of Lemma 22 is that $q = 1$ gives the fastest primal convergence rate of Algorithm 1.

**Corollary 23.** *Consider the sequence $\{\delta_k\}$ with the best choice of $q = 1$ in terms of the primal convergence rate of Algorithm 2, i.e., $\delta_k = \frac{1}{p \cdot k + 1}$. For a given tolerance $\epsilon > 0$, the number of iterations required to reach $\epsilon$-primal stationarity, $\frac{1}{T} \sum_{k=0}^{T-1} \|\boldsymbol{\zeta}_{\mathbf{p}}^{k+1}\| \leq \epsilon$, is upper bounded by $\widetilde{\mathcal{O}}\left(1/\epsilon^2\right)$.*

Note that even with the choice of $2/3 < q < 1$ for the sequence $\{\delta_k\}$, we can derive the complexity bound of $\mathcal{O}\left(1/\epsilon^{2/q}\right)$ through a similar analysis. This is still an improved complexity bound compared to the best-known complexity of $\mathcal{O}\left(1/\epsilon^3\right)$.

*Remark* 24. As an immediate consequence of results in Lemma 38 and Lemma 22, we also have the result: $\lim_{T\to\infty}\frac{1}{T}\sum_{k=0}^{T}\left(\|\mathbf{x}_{k+1}-\mathbf{x}_k\|^2+\|\mathbf{u}_{k+1}-\mathbf{u}_k\|^2\right)=0$. This result implies the following rates of the squared running-average successive differences of primal iterates:

$$\frac{1}{T}\sum_{k=0}^{T-1}\left(\|\mathbf{x}_{k+1}-\mathbf{x}_k\|^2+\|\mathbf{u}_{k+1}-\mathbf{u}_k\|^2\right)=\begin{cases}\mathcal{O}\left(\frac{\log(T)}{T}\right)=\widetilde{\mathcal{O}}\left(\frac{1}{T}\right) & \text{if } q=1,\\ \mathcal{O}\left(\frac{1}{T^q}\right) & \text{if } \frac{2}{3}<q<1,\end{cases}$$

It remains to prove that $\lim_{k\to\infty}\|\boldsymbol{\lambda}_k-\boldsymbol{\mu}_k\|=0$ to show the feasibility guarantees of our algorithm, which will complete our arguement of obtaining an improved iteration complexity among algorithms solving problem equation 1. This can be easily achieved by the structural properties of Algorithm 2.

**Lemma 25** (Primal Feasibility). *Let $\{\mathbf{w}_k\}$ be the sequence generated by Algorithm 2. Under Assumptions 3-7, the gap between the dual variables vanishes:*

$$\lim_{k\to\infty}\|\boldsymbol{\lambda}_k-\boldsymbol{\mu}_k\|=0.$$

*Consequently, any limit point $\bar{\mathbf{x}}$ of the sequence $\{\mathbf{x}_k\}$ is feasible for problem equation 1, satisfying $g(\bar{\mathbf{x}})\leq\mathbf{0}$. Moreover, defining $\boldsymbol{\zeta}_{\mathbf{d}}^{k+1}:=(\zeta_{\boldsymbol{\lambda}}^{k+1},\zeta_{\boldsymbol{\mu}}^{k+1})=(\mathbf{0},\frac{1}{\rho}(\boldsymbol{\lambda}_{k+1}-\boldsymbol{\mu}_{k+1}))\in\nabla_{\mathbf{d}}\mathcal{L}_\rho(\mathbf{w}_{k+1})$, we have the running-average feasibility residual:*

$$\frac{1}{T}\sum_{k=0}^{T-1}\|\boldsymbol{\zeta}_{\mathbf{d}}^{k+1}\|^2=\mathcal{O}\left(\frac{\log(T)}{T}\right)=\widetilde{\mathcal{O}}\left(\frac{1}{T}\right). \tag{25}$$

**Lemma 26** (Dual feasibility). *Let $\bar{\boldsymbol{\lambda}}$ be a limit point of the sequence $\{\boldsymbol{\lambda}_k\}$ generated by Algorithm 2. Under Assumptions 3-7, $\bar{\boldsymbol{\lambda}}$ is feasible for the dual problem of equation 1, satisfying $\bar{\boldsymbol{\lambda}}\geq\mathbf{0}$.*

**Lemma 27** (Complementary Slackness). *Let $(\bar{\mathbf{x}},\bar{\boldsymbol{\lambda}})$ be a limit point of the sequence $\{(\mathbf{x}_k,\boldsymbol{\lambda}_k)\}$ generated by Algorithm 2. Then, $(\bar{\mathbf{x}},\bar{\boldsymbol{\lambda}})$ satisfies the complementary slackness for problem of equation 1, i.e., $\bar{\boldsymbol{\lambda}}^\top g(\bar{\mathbf{x}})=0$.*

**Theorem 28** (Convergence to a KKT Point). *Let $\{\mathbf{w}_k=(\mathbf{x}_k,\mathbf{u}_k,\mathbf{z}_k,\boldsymbol{\lambda}_k,\boldsymbol{\mu}_k)\}$ be the sequence generated by Algorithm 2. Under Assumptions 3-7, any limit point $\bar{\mathbf{w}}$ of the sequence $\{\mathbf{w}_k\}$ corresponds to a KKT point of the original problem equation 1 as defined in Definition 1.*

*Proof.* By Lemmas 22, 25, 26 and 27, $\bar{\mathbf{w}}$ satisfies the KKT conditions as defined in Definition 1. $\square$

Notably, this eliminates the need for strong regularity assumptions, which is often imposed by several AL-based algorithms Li et al. (2021); Lin et al. (2022); Lu (2022); Sahin et al. (2019) to ensure feasibility. For Algorithm 2, we construct a non-negative auxiliary sequence as

$$\boldsymbol{\nu}_k:=\boldsymbol{\lambda}_k+\boldsymbol{\lambda}_{k+1}-\boldsymbol{\mu}_{k+1}+\left(\frac{1}{\tau}-\rho\right)(\mathbf{u}_{k+1}-\mathbf{u}_k). \tag{26}$$

Note that $\boldsymbol{\nu}_k\geq\mathbf{0}$ for all $k\geq0$. By the first order optimality of $\mathbf{u}_{k+1}$ for equation 20,

$$\mathbf{u}_k-\tau(\boldsymbol{\lambda}_k+\rho(g(\mathbf{x}_{k+1})+\mathbf{u}_k))\leq\mathbf{u}_{k+1}.$$

And by the lambda update equation 13,

$$\mathbf{0}\leq\mathbf{u}_{k+1}-\mathbf{u}_k+\tau(\boldsymbol{\lambda}_k+\boldsymbol{\lambda}_{k+1}-\boldsymbol{\mu}_{k+1}-\rho\mathbf{u}_{k+1}+\rho\mathbf{u}_k)$$

$$\mathbf{0}\leq(\mathbf{u}_{k+1}-\mathbf{u}_k)(1-\tau\rho)+\tau(\boldsymbol{\lambda}_k+\boldsymbol{\lambda}_{k+1}-\boldsymbol{\mu}_{k+1})$$

$$\mathbf{0}\leq\left(\frac{1}{\tau}-\rho\right)(\mathbf{u}_{k+1}-\mathbf{u}_k)+\boldsymbol{\lambda}_k+\boldsymbol{\lambda}_{k+1}-\boldsymbol{\mu}_{k+1}=\boldsymbol{\nu}_k.$$

**Lemma 29** (Non-asymptotic rate for primal stationarity). *Let $\{\mathbf{p}_k:=(\mathbf{x}_k,\mathbf{u}_k,\mathbf{z}_k)\}$ be the primal sequence generated by Algorithm 2 using the non-negative multiplier $\bar{\boldsymbol{\nu}}_T$. Under Assumptions 3-6, average primal stationarity residual converges as*

$$\frac{1}{T}\sum_{k=0}^{T-1}\|\boldsymbol{\zeta}_{\mathbf{p}}^{k+1}\|^2=\tilde{\mathcal{O}}\left(\frac{1}{T}\right),$$

*where $\boldsymbol{\zeta}_{\mathbf{p}}^{k+1}:=(\zeta_{\mathbf{x}}^{k+1},\zeta_{\mathbf{u}}^{k+1},\zeta_{\mathbf{z}}^{k+1})\in\partial_{\mathbf{p}}\mathcal{L}_{\alpha\beta}(\mathbf{w}_{k+1})$.*

**Lemma 30** (Non-asymptotic rate for primal feasibility). *Let $\{\mathbf{x}_k\}$ be the primal sequence generated by Algorithm 2. Under Assumptions 3-7, average primal feasibility violation converges as*

$$\frac{1}{T}\sum_{k=0}^{T-1}\|[g(\mathbf{x}_{k+1})]^+\|^2 = \tilde{\mathcal{O}}\left(\frac{1}{T}\right). \tag{27}$$

**Lemma 31** (Non-asymptotic rate for complementary slackness). *Let $\{\mathbf{x}_k, \boldsymbol{\nu}_k\}$ be the sequence generated by Algorithm 2. Under Assumptions 3-7, average complementary slackness for Definition 2 converges as*

$$\frac{1}{T}\sum_{k=0}^{T-1}\sum_{j=1}^{m}|\nu_{j,k}g_j(\mathbf{x}_{k+1})| = \tilde{\mathcal{O}}(1/\sqrt{T}). \tag{28}$$

**Theorem 32** (Non-asymptotic Rate of Convergence). *Under Assumptions 3-7, there exists a uniformly-at-random iterate $k \in \{0, \cdots, K-1\}$ from the sequence generated by Algorithm 2 that is a $\epsilon$-KKT solution to problem 1 on expectation as defined in Definition 2. The total number of iterations required to achieve this is bounded by $\tilde{\mathcal{O}}(1/\epsilon^2)$.*

*Proof.* The proof is analogous to that of Theorem 21 using Lemmas 29, 30 and 31 and the construction of the non-negative multiplier sequence equation 26. □

## 5  Numerical Experiments

This section presents a comprehensive set of numerical experiments designed to validate the theoretical results and demonstrate the practical advantages of our proposed algorithms. We evaluate our framework on a range of non-convex optimization problems, including those with non-smooth constraints and smooth, highly non-convex constraints. Our goals are twofold: (1) to empirically verify the convergence properties and improved efficiency of our methods, and (2) to benchmark their performance against existing state-of-the-art algorithms. The results confirm the robustness and superior performance of our approach, particularly in large-scale and complex settings. Experimental details including implementation details, datasets and setups are provided in Supplemenary Materials.

### 5.1  Classification Problems Under Non-smooth Fairness Constraints

We first evaluate Algorithm 1 on real-world datasets with non-convex non-smooth fairness constraints using tractable single gradient step approximation as in Algorithm 2. In Supplementary Materials, we also provide experiments on hyperparameter robustness, dual variables convergence and extension to highly stochastic setting.

#### 5.1.1  Demographic Parity Constraint

Our first experiment addresses the problem of minimizing the logistic empirical loss:

$$f(\mathbf{x}) = \frac{1}{N}\sum_{i=1}^{N}\log(1 + e^{-y_i\mathbf{x}^\top x_i}), \tag{29}$$

subject to a demographic parity (DP) constraint:

$$\widehat{\Delta}_D(\mathbf{x}) = \left|\frac{1}{N_p}\sum_{i\in I_p}\sigma(\mathbf{x}^\top x_i) - \frac{1}{N_u}\sum_{i\in I_u}\sigma(\mathbf{x}^\top x_i)\right|, \tag{30}$$

which measures the absolute difference in the positive prediction rates between protected ($I_p$) and unprotected ($I_u$) groups with corresponding sizes of $N_p = |I_p|$ and $N_u = |I_u|$. Equation equation 30 uses sigmoid $\sigma(\cdot)$ as a surrogate. This results in smooth and convex objective and a weakly convex and non-smooth constraint. Figure 1 depicts the performance of all algorithms across three datasets. The result show that PLADA consistently converges faster with lower loss and smaller constraint violation compared to the benchmark methods.

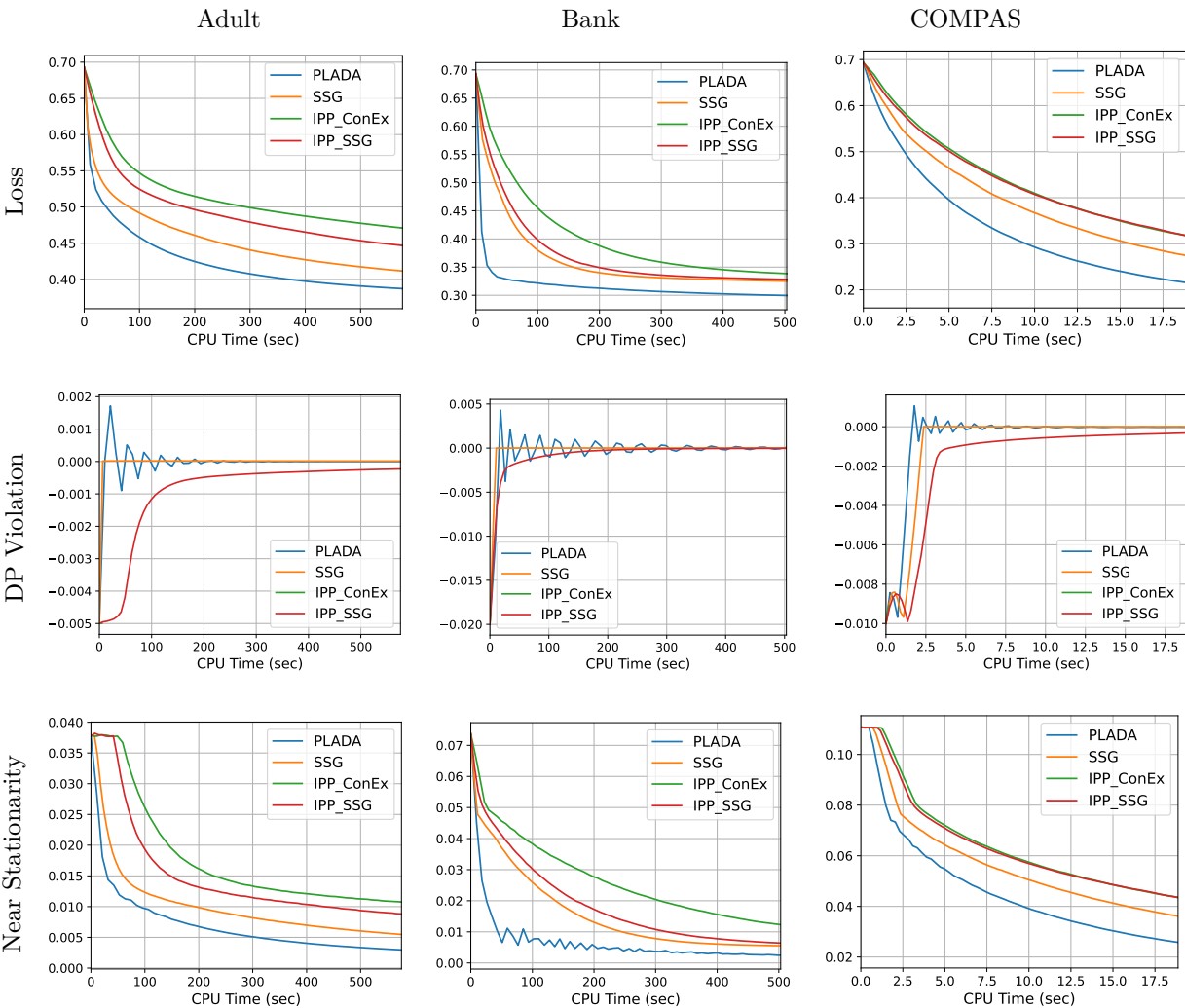

Figure 1: Comparison of the performance of PLADA, IPP-ConEx, IPP-SSG and SSG on the logistic loss equation 29 with demographic parity (DP) constraint equation 30. The results are presented in terms of their loss values, constraint violation and near stationarity (from top to bottom) on Adult, Bank and COMPAS datasets (from left to right) with respect to CPU time in seconds.

### 5.1.2 Equalized Odds Constraints

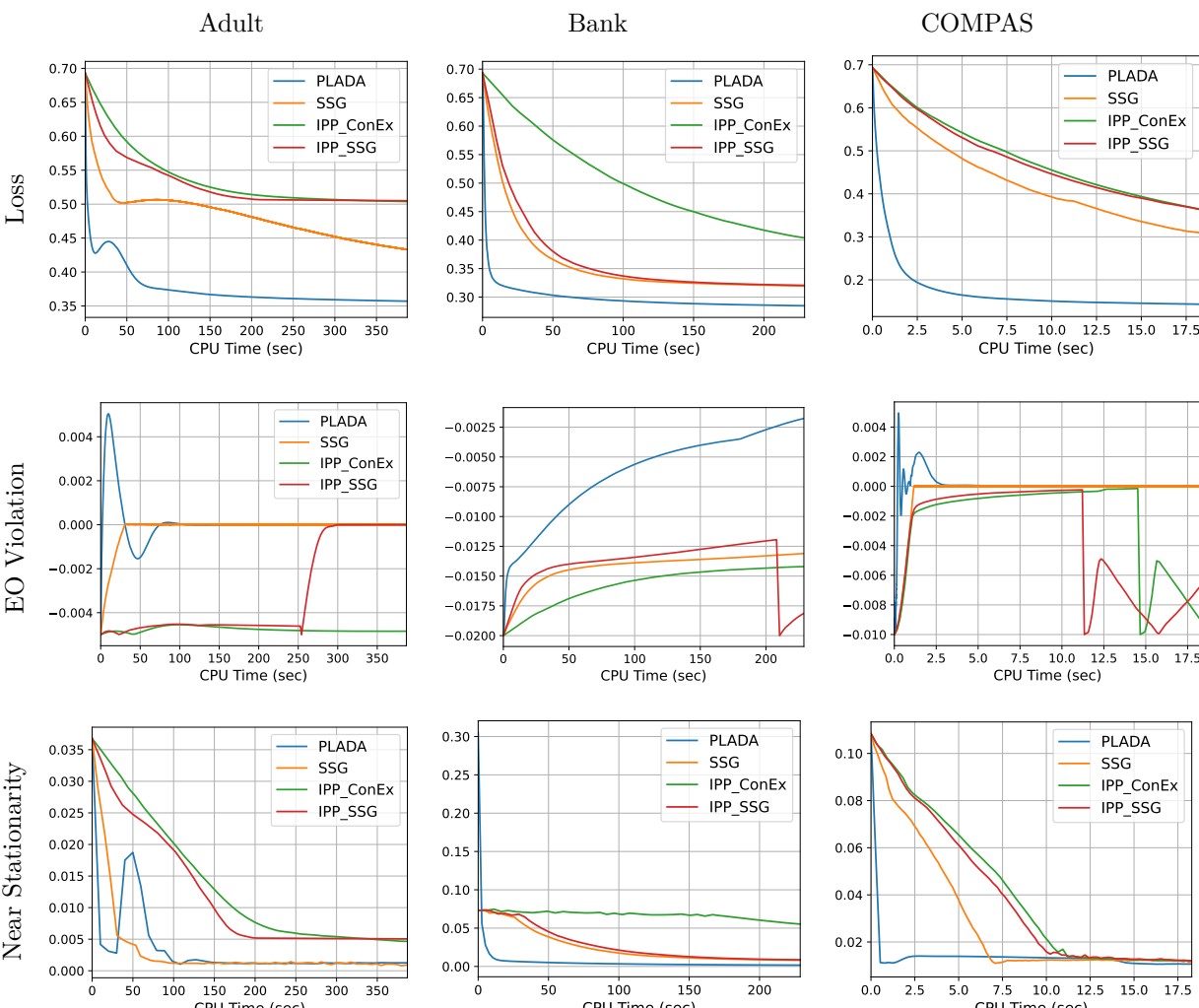

Figure 2: Comparison of the performance of PLADA, IPP-ConEx, IPP-SSG and SSG on the logistic loss objective (29) and the equalized odds (EO) constraint (31) with respect to CPU time.

Next, we consider a stricter and more challenging fairness notion: equalized odds (EO). EO constraint equation 31 requires that the true positive rates and false positive rates are equal across protected and unprotected groups. This results in two separate constraints, which we formulate using a max operator for the benchmark algorithms that only support a single constraint:

$$\widehat{\Delta}_E(\mathbf{x}) = \max\Bigg(\left| \frac{1}{N_{pq}} \sum_{i \in I_{pq}} \sigma(\mathbf{x}^\top x_i) - \frac{1}{N_{uq}} \sum_{i \in I_{uq}} \sigma(\mathbf{x}^\top x_i) \right|, $$
$$\left| \frac{1}{N_{pu}} \sum_{i \in I_{pu}} \sigma(\mathbf{x}^\top x_i) - \frac{1}{N_{uu}} \sum_{i \in I_{uu}} \sigma(\mathbf{x}^\top x_i) \right| \Bigg). \tag{31}$$

A notable advantage of PLADA is its ability to handle multiple constraints by alternatingly optimizing parameters $(\mathbf{u}, \mathbf{z}, \boldsymbol{\lambda}, \boldsymbol{\mu})$. Figure 2 shows that PLADA's advantage is even more pronounced in this more challenging setting.

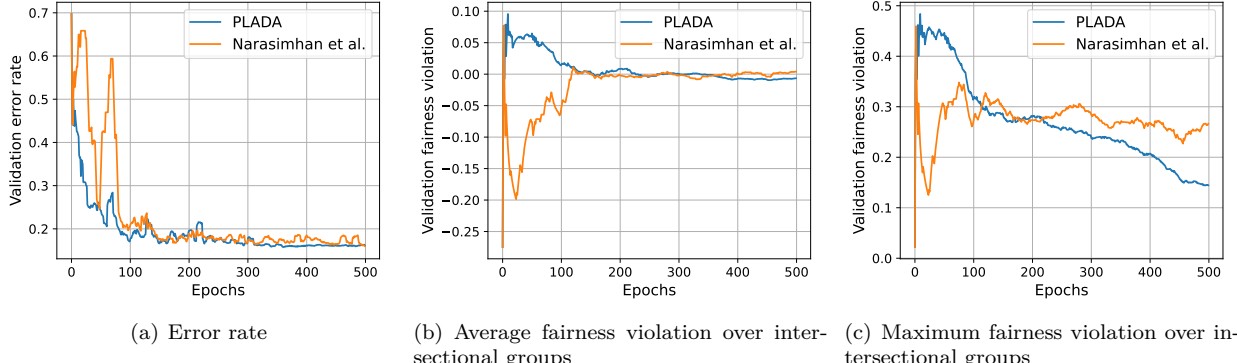

(a) Error rate    (b) Average fairness violation over intersectional groups    (c) Maximum fairness violation over intersectional groups

Figure 3: Comparison of the validation performance of PLADA and Narasimhan et al., Narasimhan et al. (2020) on the intersectional group fairness equation 32 versus Epochs.

### 5.1.3 Intersectional Group Fairness on Neural Networks

The fairness constraint equation 32 is particularly demanding, requiring parity across intersectional groups defined on the Communities and Crime dataset Redmond (2009). In particular, the groups are created with ten thresholds on three criteria: the percentages of the Black, Hispanic and Asian populations. Among $10^3$ groups, 535 groups with memberships of more than 1% of data points are selected. The constraint is formulated as an expectation over the fairness violations for each group:

$$\widehat{\Delta}_I(\mathbf{x}) = \mathbb{E}_G \left[ \frac{1}{N_G} \sum_{i \in I_G} [1 - y_i f_\mathbf{x}(x_i)]^+ - \frac{1}{N} \sum_{i=1}^N [1 - y_i f_\mathbf{x}(x_i)]^+ \right], \tag{32}$$

where $f_\mathbf{x}(\cdot)$ is the neural network classifier, $G$ is a uniformly sampled group, and $[\cdot]^+$ represents a hinge function.

We compare PLADA with the method of Narasimhan et al. (2020), which employs a separate deep neural network to update the Lagrange multipliers. In contrast, PLADA uses a simple, direct update scheme that guarantees the boundedness of the Lagrange multiplier sequence, leading to consistent fairness satisfaction. As shown in Figure 3, PLADA achieves a lower validation error rate while more effectively reducing fairness violations.

### 5.2 Non-convex Multi-class Neyman-Pearson Classification

This section evaluates the performance of PPALA on a highly non-convex multi-class Neyman-Pearson classification (mNPC) problem using neural networks.

**Task formulation.** The mNPC problem, which aims to minimize the loss for a particular class of interest while ensuring the losses for others remain below given thresholds, is formulated as:

$$\min_{\|\mathbf{x}\| \le \theta} \quad \frac{1}{|\mathcal{D}_1|} \sum_{j \ne 1} \sum_{\xi \in \mathcal{D}_1} \phi(f_1(\mathbf{x}_1; \xi) - f_j(\mathbf{x}_j; \xi))$$

$$\text{s. t.} \quad \frac{1}{|\mathcal{D}_i|} \sum_{j \ne i} \sum_{\xi \in \mathcal{D}_i} \phi(f_i(\mathbf{x}_i; \xi) - f_j(\mathbf{x}_j; \xi)) \le \kappa_i, \qquad i = 2, \dots, N,$$

where $f_i$ with weights $\mathbf{x}_i$ is a nonlinear classifier for class $i$, $\mathcal{D}_i$ is the corresponding class data, and $\phi$ is a loss function.

**Results and discussion.** Figure 4 shows that PPALA converges faster compared to GDPA. In particular, The performance advantage is particularly pronounced on the more complex CIFAR-10 dataset. PPALA's robust performance is attributable to its fixed penalty mechanism, which required minimal tuning of parameters across both datasets. In contrast, GDPA exhibited high sensitivity to its penalty parameter update

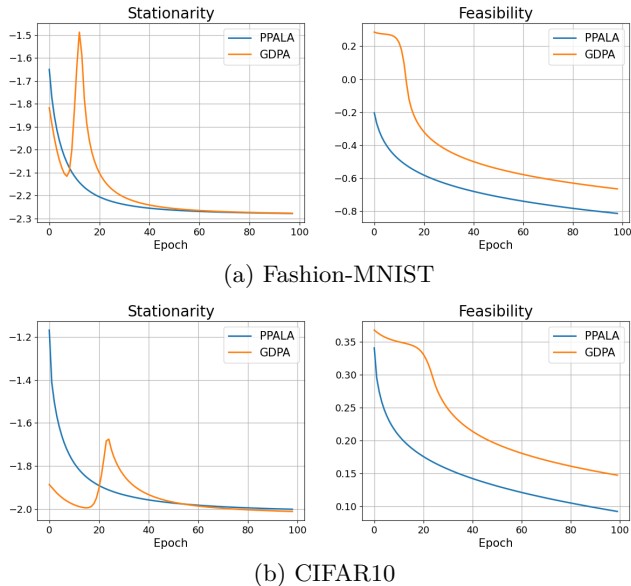

(a) Fashion-MNIST

(b) CIFAR10

Figure 4: Performance comparison of PPALA and GDPA on Fashion-MNIST and CIFAR10 datasets in terms of obtaining stationarity and feasibility. We see that PPALA provides a consistent reduction of stationarity and feasibility gaps that align with our theoretical expectations. In contrast, GDPA reduces the feasibility gap at a slower rate on Fashion-MNIST and CIFAR10 in our neural network setting.

schedule. For instance, we observed that GDPA fails to converge when using large ratio to update the penalty parameter, necessitating careful parameter selection. The gradual update of its penalty parameter hinders GDPA's ability to reduce infeasibility effifficiently, while PPALA achieves a fast and consistent reduction in infeasibility with the fixed parameter $\rho = \frac{\alpha}{1+\alpha\beta}$. The results emphasize PPALA's practical advantages in both robustness and computational efficiency when solving more complex problems with highly non-convex constraints.

## 6 Conclusions

In this paper, we have introduced a novel single-loop primal-dual algorithmic framework designed to address non-convex functional constrained optimization problems. A significant contribution of our method is its ability to achieve an improved iteration complexity of $\widetilde{\mathcal{O}}(1/\epsilon^2)$ for computing an $\epsilon$-approximate stationary solution. The proposed algorithmic framework is flexible and robust, capable of effectively handling a variety of non-convex functional constraints. This includes problems with continuously differentiable (smooth) constraints, as well as non-smooth non-convex constraints, which are made tractable through the use of suitable differentiable or sub-differentiable surrogates, particularly relevant in applications like fair classification. Our comprehensive convergence analysis demonstrates that the algorithm ensures a consistent reduction in stationarity and feasibility gaps.

Numerical experiments across diverse non-convex problems, including fairness-constrained classification and multi-class Neyman-Pearson classification (mNPC), consistently demonstrate the algorithm's effectiveness in terms of computational cost and performance. Our algorithm has been shown to outperform related approaches that use regularization techniques and/or standard Lagrangian relaxation, highlighting its superior performance and robustness, especially in large-scale and complex settings.

Future research will explore extending this simple optimization method to stochastic non-convex constrained optimization problems, possibly leveraging variance reduction strategies Cutkosky & Orabona (2019); Hashemi (2024), which would further broaden its application domain in machine learning and signal processing. Preliminary applications in such highly stochastic settings have already shown promising re-

sults in achieving better constraint satisfaction with comparable error rates. Extensions to distributed and federated learning settings is of further interests Kairouz et al. (2021); Das et al. (2022).

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

## A  Supporting Lemmas for Convergence Analysis of Algorithm 1

The core of our convergence analysis relies on the following technical lemmas, which establish bounds on the iterates and demonstrate the descent properties of the P-Lagrangian. For convenience, we let $\mathbf{w}_k := \{(\mathbf{x}_k, \mathbf{u}_k, \mathbf{z}_k, \boldsymbol{\lambda}_k, \boldsymbol{\mu}_k)\}$ denote the sequence generated by Algorithm 1.

**Lemma 33.** *Let* $\{(\mathbf{x}_k, \mathbf{u}_k, \mathbf{z}_k, \boldsymbol{\lambda}_k, \boldsymbol{\mu}_k)\}$ *be the sequence generated by Algorithm 1. Then for any* $k \geq 0$, *the following relations hold:*

$$\|\boldsymbol{\mu}_{k+1} - \boldsymbol{\mu}_k\|^2 = (\sigma_k^2/\rho^2)\|\boldsymbol{\lambda}_k - \boldsymbol{\mu}_k\|^2 \leq \delta_k^2/4; \tag{33a}$$

$$\|\boldsymbol{\mu}_{k+1} - \boldsymbol{\lambda}_k\|^2 = (1 - (\sigma_k/\rho))^2 \|\boldsymbol{\mu}_k - \boldsymbol{\lambda}_k\|^2; \tag{33b}$$

$$\|\boldsymbol{\lambda}_{k+1} - \boldsymbol{\lambda}_k\|^2 \leq 3\rho^2 M_g^2 \|\mathbf{x}_{k+1} - \mathbf{x}_k\|^2 + 3\rho^2 \|\mathbf{u}_{k+1} - \mathbf{u}_k\|^2 + 3(\sigma_k^2/\rho^2)\|\boldsymbol{\lambda}_k - \boldsymbol{\mu}_k\|^2. \tag{33c}$$

*Proof.* From the $\boldsymbol{\mu}$-update equation 11 and noting that $a + b \geq 2\sqrt{ab}$ for any $a, b \geq 0$, we immediately obtain the relations in equation 33a:

$$\|\boldsymbol{\mu}_{k+1} - \boldsymbol{\mu}_k\|^2 = \frac{\sigma_k^2}{\rho^2}\|\boldsymbol{\lambda}_k - \boldsymbol{\mu}_k\|^2 \leq \frac{\delta_k^2}{\|\boldsymbol{\lambda}_k - \boldsymbol{\mu}_k\|^2 + 2 + (1/\|\boldsymbol{\lambda}_k - \boldsymbol{\mu}_k\|^2)} \leq \frac{\delta_k^2}{4}.$$

Subtracting $\boldsymbol{\mu}_{k+1}$ from $\boldsymbol{\lambda}_k$ yields

$$\|\boldsymbol{\lambda}_k - \boldsymbol{\mu}_{k+1}\| = \left\|\boldsymbol{\lambda}_k - \boldsymbol{\mu}_k - \frac{\sigma_k}{\rho}(\boldsymbol{\lambda}_k - \boldsymbol{\mu}_k)\right\| = \left(1 - \frac{\sigma_k}{\rho}\right)\|\boldsymbol{\lambda}_k - \boldsymbol{\mu}_k\|.$$

Squaring both sides of the above inequality yields the relation equation 33b.

By the $\boldsymbol{\lambda}$-update equation 13, we have

$$\begin{aligned}\|\boldsymbol{\lambda}_{k+1} - \boldsymbol{\lambda}_k\| &\leq \|\boldsymbol{\mu}_{k+1} - \boldsymbol{\mu}_k\| + \rho\|g(\mathbf{x}_{k+1}) + \mathbf{u}_{k+1} - g(\mathbf{x}_k) - \mathbf{u}_k\| \\ &\leq \|\boldsymbol{\mu}_{k+1} - \boldsymbol{\mu}_k\| + \rho M_g\|\mathbf{x}_{k+1} - \mathbf{x}_k\| + \rho\|\mathbf{u}_{k+1} - \mathbf{u}_k\|,\end{aligned}$$

which, along with $(a + b + c)^2 \leq 3(a^2 + b^2 + c^2)$ and equation 33a, provides the relation equation 33c. $\qquad\square$

**Lemma 34** (Approximate Decrease of $\mathcal{L}_{\alpha\beta}$). *Let $\{\mathbf{w}_k\}$ be the sequence generated by Algorithm 1. Under Assumptions 4, 6 and 8, the P-Lagrangian $\mathcal{L}_{\alpha\beta}$ equation 5 satisfies:*

$$\mathcal{L}_{\alpha\beta}(\mathbf{w}_{k+1}) - \mathcal{L}_{\alpha\beta}(\mathbf{w}_k) \leq -C_1\|\mathbf{x}_{k+1} - \mathbf{x}_k\|^2 - C_2\|\mathbf{u}_{k+1} - \mathbf{u}_k\|^2 + \widehat{\delta}_k, \tag{34}$$

*where $C_1 := \frac{1}{2}\left(\frac{1}{\eta} - L_f - 3\rho M_g^2\right) > 0$, $C_2 := \frac{1}{2}\left(\frac{1}{\tau} - 3\rho\right) > 0$, and $\widehat{\delta}_k := \frac{\delta_k^2}{\rho^3} + \frac{\delta_k}{\rho^2}$.*

*Proof.* First, note that

$$\begin{aligned}\mathcal{L}_{\alpha\beta}(\mathbf{x}_k, \mathbf{u}_k, \mathbf{z}_k, \boldsymbol{\lambda}_k, \boldsymbol{\mu}_k) &= f(\mathbf{x}_k) + \langle\boldsymbol{\lambda}_k, g(\mathbf{x}_k) + \mathbf{u}_k\rangle - \langle\boldsymbol{\lambda}_k - \boldsymbol{\mu}_k, \mathbf{z}_k\rangle \\ &\quad + \frac{\alpha}{2}\|\mathbf{z}_k\|^2 - \frac{\beta}{2}\|\boldsymbol{\lambda}_k - \boldsymbol{\mu}_k\|^2 + r(\mathbf{x}_k) \\ &= f(\mathbf{x}_k) + \langle\boldsymbol{\lambda}_k, g(\mathbf{x}_k) + \mathbf{u}_k\rangle - \frac{1}{2\rho}\|\boldsymbol{\lambda}_k - \boldsymbol{\mu}_k\|^2 + r(\mathbf{x}_k) \\ &= \mathcal{L}_{\alpha\beta}(\mathbf{x}_k, \mathbf{u}_k, \widehat{\mathbf{z}}(\boldsymbol{\lambda}_k, \boldsymbol{\mu}_k), \boldsymbol{\lambda}_k, \boldsymbol{\mu}_k),\end{aligned}$$

where $\rho = \alpha/(1 + \alpha\beta)$, and thus

$$\mathcal{L}_{\alpha\beta}(\mathbf{x}_{k+1}, \mathbf{u}_{k+1}, \mathbf{z}_k, \boldsymbol{\lambda}_k, \boldsymbol{\mu}_k) = \mathcal{L}_{\alpha\beta}(\mathbf{x}_{k+1}, \mathbf{u}_{k+1}, \widehat{\mathbf{z}}(\boldsymbol{\lambda}_k, \boldsymbol{\mu}_k), \boldsymbol{\lambda}_k, \boldsymbol{\mu}_k).$$

Then the difference of two successive sequences of $\mathcal{L}_{\alpha\beta}$ can be divided into two parts:

$$\begin{aligned}&\mathcal{L}_{\alpha\beta}(\mathbf{x}_{k+1}, \mathbf{u}_{k+1}, \mathbf{z}_{k+1}, \boldsymbol{\lambda}_{k+1}, \boldsymbol{\mu}_{k+1}) - \mathcal{L}_{\alpha\beta}(\mathbf{x}_k, \mathbf{u}_k, \mathbf{z}_k, \boldsymbol{\lambda}_k, \boldsymbol{\mu}_k) \\ &= [\mathcal{L}_{\alpha\beta}(\mathbf{x}_{k+1}, \mathbf{u}_{k+1}, \mathbf{z}_k, \boldsymbol{\lambda}_k, \boldsymbol{\mu}_k) - \mathcal{L}_{\alpha\beta}(\mathbf{x}_k, \mathbf{u}_k, \mathbf{z}_k, \boldsymbol{\lambda}_k, \boldsymbol{\mu}_k)] \\ &\quad + [\mathcal{L}_{\alpha\beta}(\mathbf{x}_{k+1}, \mathbf{u}_{k+1}, \widehat{\mathbf{z}}(\boldsymbol{\lambda}_{k+1}, \boldsymbol{\mu}_{k+1}), \boldsymbol{\lambda}_{k+1}, \boldsymbol{\mu}_{k+1}) - \mathcal{L}_{\alpha\beta}(\mathbf{x}_{k+1}, \mathbf{u}_{k+1}, \widehat{\mathbf{z}}(\boldsymbol{\lambda}_k, \boldsymbol{\mu}_k), \boldsymbol{\lambda}_k, \boldsymbol{\mu}_k)].\end{aligned} \tag{35}$$

Consider the first part equation 35. Since $\mathbf{x}_{k+1}$ and $\mathbf{u}_{k+1}$ are the results of the subproblems equation 9 and equation 10, respectively, we have that for any $\mathbf{x} \in \mathcal{X}$ and for any $u \in U$,

$$\begin{aligned}&\langle\nabla f(\mathbf{x}_k), \mathbf{x}_{k+1} - \mathbf{x}\rangle + \langle\boldsymbol{\lambda}_k, g(\mathbf{x}_{k+1}) - g(\mathbf{x})\rangle \\ &\quad + \frac{1}{2\eta}\left(\|\mathbf{x}_{k+1} - \mathbf{x}_k\|^2 - \|\mathbf{x} - \mathbf{x}_k\|^2\right) + r(\mathbf{x}_{k+1}) - r(\mathbf{x}_k) \leq 0,\end{aligned}$$

and

$$\langle\nabla_u\mathcal{L}_{\alpha\beta}(\mathbf{w}_k), \mathbf{u}_{k+1} - u\rangle + \frac{1}{2\tau}(\|\mathbf{u}_{k+1} - \mathbf{u}_k\|^2 - \|\mathbf{u} - \mathbf{u}_k\|^2) \leq 0. \tag{36}$$

By taking $\mathbf{x} = \mathbf{x}_k$ in equation 36, $u = \mathbf{u}_k$ in equation 36, and using $\nabla_u \mathcal{L}_{\alpha\beta}(\mathbf{w}_k) = \boldsymbol{\lambda}_k$, we have

$$\langle \nabla f(\mathbf{x}_k), \mathbf{x}_{k+1} - \mathbf{x}_k \rangle + \langle \boldsymbol{\lambda}_k, g(\mathbf{x}_{k+1}) - g(\mathbf{x}_k) \rangle + r(\mathbf{x}_{k+1}) - r(\mathbf{x}_k) \leq -\frac{1}{2\eta} \|\mathbf{x}_{k+1} - \mathbf{x}_k\|^2,$$

and

$$\langle \boldsymbol{\lambda}_k, \mathbf{u}_{k+1} - \mathbf{u}_k \rangle \leq -\frac{1}{2\tau} \|\mathbf{u}_{k+1} - \mathbf{u}_k\|^2.$$

By adding and subtracting the term $\langle \nabla f(\mathbf{x}_k), \mathbf{x}_{k+1} - \mathbf{x}_k \rangle$, we obtain

$$
\begin{aligned}
&\mathcal{L}_{\alpha\beta}(\mathbf{x}_{k+1}, \mathbf{u}_{k+1}, \mathbf{z}_k, \boldsymbol{\lambda}_k, \boldsymbol{\mu}_k) - \mathcal{L}_{\alpha\beta}(\mathbf{x}_k, \mathbf{u}_k, \mathbf{z}_k, \boldsymbol{\lambda}_k, \boldsymbol{\mu}_k) \\
&= [f(\mathbf{x}_{k+1}) + \langle \boldsymbol{\lambda}_k, g(\mathbf{x}_{k+1}) + \mathbf{u}_{k+1} \rangle + r(\mathbf{x}_{k+1})] - [f(\mathbf{x}_k) + \langle \boldsymbol{\lambda}_k, g(\mathbf{x}_k) + \mathbf{u}_k \rangle + r(\mathbf{x}_k)] \\
&= \langle \boldsymbol{\lambda}_k, g(\mathbf{x}_{k+1}) - g(\mathbf{x}_k) \rangle + \langle \boldsymbol{\lambda}_k, \mathbf{u}_{k+1} - \mathbf{u}_k \rangle + [f(\mathbf{x}_{k+1}) - f(\mathbf{x}_k)] + [r(\mathbf{x}_{k+1}) - r(\mathbf{x}_k)] \\
&= [\langle \nabla f(\mathbf{x}_k), \mathbf{x}_{k+1} - \mathbf{x}_k \rangle + \langle \boldsymbol{\lambda}_k, g(\mathbf{x}_{k+1}) - g(\mathbf{x}_k) \rangle + r(\mathbf{x}_{k+1}) - r(\mathbf{x}_k)] \\
&\quad + [f(\mathbf{x}_{k+1}) - f(\mathbf{x}_k) - \langle \nabla f(\mathbf{x}_k), \mathbf{x}_{k+1} - \mathbf{x}_k \rangle] + \langle \boldsymbol{\lambda}_k, \mathbf{u}_{k+1} - \mathbf{u}_k \rangle \\
&\leq -\frac{1}{2} \left( \frac{1}{\eta} - L_f \right) \|\mathbf{x}_{k+1} - \mathbf{x}_k\|^2 - \frac{1}{2\tau} \|\mathbf{u}_{k+1} - \mathbf{u}_k\|^2.
\end{aligned}
\tag{37}
$$

Next, we derive an upper bound for the second part. We start by noting that

$$
\begin{aligned}
&\mathcal{L}_{\alpha\beta}(\mathbf{x}_{k+1}, \mathbf{u}_{k+1}, \widehat{\mathbf{z}}(\boldsymbol{\lambda}_{k+1}, \boldsymbol{\mu}_{k+1}), \boldsymbol{\lambda}_{k+1}, \boldsymbol{\mu}_{k+1}) - \mathcal{L}_{\alpha\beta}(\mathbf{x}_{k+1}, \mathbf{u}_{k+1}, \widehat{\mathbf{z}}(\boldsymbol{\lambda}_k, \boldsymbol{\mu}_k), \boldsymbol{\lambda}_k, \boldsymbol{\mu}_k) \\
&= \langle \boldsymbol{\lambda}_{k+1} - \boldsymbol{\lambda}_k, g(\mathbf{x}_{k+1}) + \mathbf{u}_{k+1} \rangle - \frac{1}{2\rho} \left( \|\boldsymbol{\lambda}_{k+1} - \boldsymbol{\mu}_{k+1}\|^2 - \|\boldsymbol{\lambda}_k - \boldsymbol{\mu}_k\|^2 \right).
\end{aligned}
$$

Using the facts that $g(\mathbf{x}_{k+1}) + \mathbf{u}_{k+1} = \frac{1}{\rho}(\boldsymbol{\lambda}_{k+1} - \boldsymbol{\mu}_{k+1})$ and $\langle a, b \rangle = \frac{1}{2}\|a\|^2 + \frac{1}{2}\|b\|^2 - \frac{1}{2}\|a - b\|^2$ for any $a, b \in \mathbb{R}^m$, we have

$$\frac{1}{\rho} \langle \boldsymbol{\lambda}_{k+1} - \boldsymbol{\lambda}_k, \boldsymbol{\lambda}_{k+1} - \boldsymbol{\mu}_{k+1} \rangle = \frac{1}{2\rho} \left( \|\boldsymbol{\lambda}_{k+1} - \boldsymbol{\lambda}_k\|^2 + \|\boldsymbol{\lambda}_{k+1} - \boldsymbol{\mu}_{k+1}\|^2 - \|\boldsymbol{\mu}_{k+1} - \boldsymbol{\lambda}_k\|^2 \right).$$

Hence,

$$
\begin{aligned}
&\mathcal{L}_{\alpha\beta}(\mathbf{x}_{k+1}, \mathbf{u}_{k+1}, \widehat{\mathbf{z}}(\boldsymbol{\lambda}_{k+1}, \boldsymbol{\mu}_{k+1}), \boldsymbol{\lambda}_{k+1}, \boldsymbol{\mu}_{k+1}) - \mathcal{L}_{\alpha\beta}(\mathbf{x}_{k+1}, \mathbf{u}_{k+1}, \widehat{\mathbf{z}}(\boldsymbol{\lambda}_k, \boldsymbol{\mu}_k), \boldsymbol{\lambda}_k, \boldsymbol{\mu}_k) \\
&\overset{(a)}{\leq} \frac{1}{2\rho} \left( 3\rho^2 M_g^2 \|\mathbf{x}_{k+1} - \mathbf{x}_k\|^2 + 3\rho^2 \|\mathbf{u}_{k+1} - \mathbf{u}_k\|^2 + \frac{3\sigma_k^2}{\rho^2} \|\boldsymbol{\lambda}_k - \boldsymbol{\mu}_k\|^2 \right) \\
&\quad + \frac{1}{2\rho} \left( 1 - \left( 1 - \frac{\sigma_k}{\rho} \right)^2 \right) \|\boldsymbol{\lambda}_k - \boldsymbol{\mu}_k\|^2 \\
&= \frac{1}{2} \left( 3\rho M_g^2 \|\mathbf{x}_{k+1} - \mathbf{x}_k\|^2 + 3\rho \|\mathbf{u}_{k+1} - \mathbf{u}_k\|^2 \right) + \frac{3\sigma_k^2}{2\rho^3} \|\boldsymbol{\lambda}_k - \boldsymbol{\mu}_k\|^2 \\
&\quad + \frac{1}{2\rho} \left( \frac{2\sigma_k}{\rho} - \frac{\sigma_k^2}{\rho^2} \right) \|\boldsymbol{\lambda}_k - \boldsymbol{\mu}_k\|^2 \\
&\overset{(b)}{\leq} \frac{1}{2} \left( 3\rho M_g^2 \|\mathbf{x}_{k+1} - \mathbf{x}_k\|^2 + 3\rho \|\mathbf{u}_{k+1} - \mathbf{u}_k\|^2 \right) + \frac{\delta_k^2}{\rho^3} + \frac{\delta_k}{\rho^2},
\end{aligned}
\tag{38}
$$

where $(a)$ is from equation 33b and equation 33c, and $(b)$ holds by $\sigma_k \|\boldsymbol{\lambda}_k - \boldsymbol{\mu}_k\|^2 \leq \frac{\delta_k}{1 + (1/\|\boldsymbol{\lambda}_k - \boldsymbol{\mu}_k\|^2)} \leq \delta_k$. Combining equation 37 and equation 38 yields the desired result:

$$
\begin{aligned}
&\mathcal{L}_{\alpha\beta}(\mathbf{w}_{k+1}) - \mathcal{L}_{\alpha\beta}(\mathbf{w}_k) \\
&\leq -\frac{1}{2} \left( \frac{1}{\eta} - L_f - 3\rho M_g^2 \right) \|\mathbf{x}_{k+1} - \mathbf{x}_k\|^2 - \frac{1}{2} \left( \frac{1}{\tau} - 3\rho \right) \|\mathbf{u}_{k+1} - \mathbf{u}_k\|^2 + \frac{\delta_k^2}{\rho^3} + \frac{\delta_k}{\rho^2},
\end{aligned}
$$

which completes the proof. $\qquad\square$

**Lemma 35** (Subgradient Error Bound). *Let $\{\mathbf{w}_k\}$ be the sequence generated by Algorithm 1, and let $\{\mathbf{p}_k :=$ $(\mathbf{x}_k, \mathbf{u}_k, \mathbf{z}_k)\}$ be the primal sequence. Under Assumptions 4 and 6, there exists a constant $d_1 > 0$ such that for the primal subgradient $\boldsymbol{\zeta}_{\mathbf{p}}^{k+1} := (\zeta_{\mathbf{x}}^{k+1}, \zeta_{\mathbf{u}}^{k+1}, \mathbf{0}) \in \partial_{\mathbf{p}} \mathcal{L}_{\alpha\beta}(\mathbf{w}_{k+1})$,*

$$\|\boldsymbol{\zeta}_{\mathbf{p}}^{k+1}\| \le d_1 \left( \|\mathbf{x}_{k+1} - \mathbf{x}_k\| + \|\mathbf{u}_{k+1} - \mathbf{u}_k\| \right) + (M_g + 1)\delta_k,$$

*where*

$$d_1 = \max\{L_f + 1/\eta + \rho(M_g^2 + M_g), \ \rho(M_g + 1) + 1/\tau\}.$$

*Proof.* Writing down the optimality condition for the update of $\mathbf{x}_{k+1}$ in equation 9, we have

$$0 \in \nabla f(\mathbf{x}_k) + \partial g(\mathbf{x}_{k+1})^\top \boldsymbol{\lambda}_k + \frac{1}{\eta}(\mathbf{x}_{k+1} - \mathbf{x}_k) + v_{k+1}, \quad v_{k+1} \in \partial r(\mathbf{x}_{k+1}) \tag{39}$$

Using the subdifferential calculus rules, we have

$$\nabla f(\mathbf{x}_{k+1}) + \partial g(\mathbf{x}_{k+1})^\top \boldsymbol{\lambda}_{k+1} + v_{k+1} \in \partial_{\mathbf{x}} \mathcal{L}_{\alpha\beta}(\mathbf{w}_{k+1}) \tag{40}$$

By defining the quantity

$$\boldsymbol{\zeta}_{\mathbf{x}}^{k+1} = \nabla f(\mathbf{x}_{k+1}) - \nabla f(\mathbf{x}_k) + \partial g(\mathbf{x}_{k+1})^\top (\boldsymbol{\lambda}_{k+1} - \boldsymbol{\lambda}_k) - \frac{1}{\eta}(\mathbf{x}_{k+1} - \mathbf{x}_k) \tag{41}$$

and using equation 39 and equation 40, we obtain that $\boldsymbol{\zeta}_{\mathbf{x}}^{k+1} \in \partial_{\mathbf{x}} \mathcal{L}_{\alpha\beta}(\mathbf{w}_{k+1})$.

Next, define the quantity

$$\boldsymbol{\zeta}_{\mathbf{u}}^{k+1} := \mathbf{u}_{k+1} - \Pi_U[\mathbf{u}_{k+1} - \boldsymbol{\lambda}_{k+1}],$$

which is equivalent to the *projected gradient* of $\mathcal{L}_{\alpha\beta}$ in $\mathbf{u}$. It is a measure of optimality for the update of $\mathbf{u}_{k+1}$ Nesterov (2012):

$$\widetilde{\nabla}_{\mathbf{u}} \mathcal{L}_{\alpha\beta}(\mathbf{w}_{k+1}) := \mathbf{u}_{k+1} - \underset{v \in U}{\operatorname{argmin}} \left\{ \langle \nabla_{\mathbf{u}} \mathcal{L}_{\alpha\beta}(\mathbf{w}_{k+1}), v - \mathbf{u}_{k+1} \rangle + \frac{1}{2} \|v - \mathbf{u}_{k+1}\|^2 \right\}$$
$$= \mathbf{u}_{k+1} - \widetilde{\mathbf{u}}_{k+1}.$$

where we define $\widetilde{\mathbf{u}}_{k+1} := \operatorname{argmin}_{v \in U} \left\{ \langle \nabla_{\mathbf{u}} \mathcal{L}_{\alpha\beta}(\mathbf{w}_{k+1}), v - \mathbf{u}_{k+1} \rangle + \frac{1}{2} \|v - \mathbf{u}_{k+1}\|^2 \right\}$.

From the update of $\mathbf{z}_{k+1}$ in equation 6, we have

$$\nabla_{\mathbf{z}} \mathcal{L}_{\alpha\beta}(\mathbf{w}_{k+1}) = -(\boldsymbol{\lambda}_{k+1} - \mu_{k+1}) + \alpha \mathbf{z}_{k+1} = 0.$$

Hence, we obtain

$$\boldsymbol{\zeta}_{\mathbf{p}}^{k+1} := \begin{pmatrix} \zeta_{\mathbf{x}}^{k+1} \\ \zeta_{\mathbf{u}}^{k+1} \\ 0 \end{pmatrix} \quad \text{where} \quad \begin{pmatrix} \zeta_{\mathbf{x}}^{k+1} & \in \partial_{\mathbf{x}} \mathcal{L}_{\alpha\beta}(\mathbf{x}_{k+1}, \mathbf{u}_{k+1}, \mathbf{z}_{k+1}, \boldsymbol{\lambda}_{k+1}, \mu_{k+1}) \\ \zeta_{\mathbf{u}}^{k+1} & = \widetilde{\nabla}_{\mathbf{u}} \mathcal{L}_{\alpha\beta}(\mathbf{x}_{k+1}, \mathbf{u}_{k+1}, \mathbf{z}_{k+1}, \boldsymbol{\lambda}_{k+1}, \mu_{k+1}) \\ 0 & = \nabla_{\mathbf{z}} \mathcal{L}_{\alpha\beta}(\mathbf{x}_{k+1}, \mathbf{u}_{k+1}, \mathbf{z}_{k+1}, \boldsymbol{\lambda}_{k+1}, \mu_{k+1}) \end{pmatrix}.$$

We derive an upper estimate for $\boldsymbol{\zeta}_{\mathbf{p}}^{k+1}$. A direct calculation gives

$$\begin{aligned}
\|\boldsymbol{\zeta}_{\mathbf{x}}^{k+1}\| &\le \|\nabla f(\mathbf{x}_{k+1}) - \nabla f(\mathbf{x}_k)\| + (1/\eta)\|\mathbf{x}_k - \mathbf{x}_{k+1}\| + \|\partial g(\mathbf{x}_{k+1})\| \|\boldsymbol{\lambda}_{k+1} - \boldsymbol{\lambda}_k\| \\
&\le (L_f + 1/\eta)\|\mathbf{x}_{k+1} - \mathbf{x}_k\| + M_g \|\boldsymbol{\lambda}_{k+1} - \boldsymbol{\lambda}_k\| \\
&\le (L_f + 1/\eta)\|\mathbf{x}_{k+1} - \mathbf{x}_k\| + \rho M_g^2 \|\mathbf{x}_{k+1} - \mathbf{x}_k\| + \rho M_g \|\mathbf{u}_{k+1} - \mathbf{u}_k\| + M_g \delta_k \\
&\le (L_f + 1/\eta + \rho M_g^2)\|\mathbf{x}_{k+1} - \mathbf{x}_k\| + \rho M_g \|\mathbf{u}_{k+1} - \mathbf{u}_k\| + M_g \delta_k
\end{aligned} \tag{42}$$

Next, we estimate an upper bound for the component $\boldsymbol{\zeta}_{\mathbf{u}}^{k+1}$. The first-order optimality condition implies that

$$\langle \nabla_{\mathbf{u}} \mathcal{L}_{\alpha\beta}(\mathbf{u}_{k+1}) + (\widetilde{\mathbf{u}}_{k+1} - \mathbf{u}_{k+1}), \mathbf{u} - \widetilde{\mathbf{u}}_{k+1} \rangle \ge 0. \tag{43}$$

Here, $\nabla_{\mathbf{u}}\mathcal{L}_{\alpha\beta}(\mathbf{w}_{k+1})$ is denoted by $\nabla_{\mathbf{u}}\mathcal{L}_{\alpha\beta}(\mathbf{u}_{k+1})$. By the definition $\mathbf{u}_{k+1}$ in equation 10, we have

$$\left\langle \nabla_{\mathbf{u}}\mathcal{L}_{\alpha\beta}(\mathbf{u}_k) + \frac{1}{\tau}(\mathbf{u}_{k+1} - \mathbf{u}_k), \mathbf{u} - \mathbf{u}_{k+1} \right\rangle \geq 0, \tag{44}$$

where $\nabla_{\mathbf{u}}\mathcal{L}_{\alpha\beta}(\mathbf{u}_k) = \nabla_{\mathbf{u}}\mathcal{L}_{\alpha\beta}(\mathbf{x}_k, \mathbf{u}_k, \mathbf{z}_k, \boldsymbol{\lambda}_k, \boldsymbol{\mu}_k)$ for simplicity. Combining equation 43 and equation 44, with settings $\mathbf{u} = \mathbf{u}_{k+1}$ in equation 43 and $\mathbf{u} = \widetilde{\mathbf{u}}_{k+1}$ in equation 44, yields

$$\left\langle \nabla_{\mathbf{u}}\mathcal{L}_{\alpha\beta}(\mathbf{u}_k) - \nabla_{\mathbf{u}}\mathcal{L}_{\alpha\beta}(\mathbf{u}_{k+1}) + \frac{1}{\tau}(\mathbf{u}_{k+1} - \mathbf{u}_k) - (\widetilde{\mathbf{u}}_{k+1} - \mathbf{u}_{k+1}), \widetilde{\mathbf{u}}_{k+1} - \mathbf{u}_{k+1} \right\rangle \geq 0,$$

equivalently,

$$\left\langle \nabla_{\mathbf{u}}\mathcal{L}_{\alpha\beta}(\mathbf{u}_k) - \nabla_{\mathbf{u}}\mathcal{L}_{\alpha\beta}(\mathbf{u}_{k+1}) + \frac{1}{\tau}(\mathbf{u}_{k+1} - \mathbf{u}_k), \widetilde{\mathbf{u}}_{k+1} - \mathbf{u}_{k+1} \right\rangle \geq \|\widetilde{\mathbf{u}}_{k+1} - \mathbf{u}_{k+1}\|^2.$$

By applying the Cauchy-Schwarz inequality and triangle inequality yields

$$\left( \|\nabla_{\mathbf{u}}\mathcal{L}_{\alpha\beta}(\mathbf{u}_k) - \nabla_{\mathbf{u}}\mathcal{L}_{\alpha\beta}(\mathbf{u}_{k+1})\| + \frac{1}{\tau}\|\mathbf{u}_{k+1} - \mathbf{u}_k\| \right) \|\widetilde{\mathbf{u}}_{k+1} - \mathbf{u}_{k+1}\| \geq \|\widetilde{\mathbf{u}}_{k+1} - \mathbf{u}_{k+1}\|^2$$

and

$$\|\nabla_{\mathbf{u}}\mathcal{L}_{\alpha\beta}(\mathbf{u}_k) - \nabla_{\mathbf{u}}\mathcal{L}_{\alpha\beta}(\mathbf{u}_{k+1})\| \leq \|\boldsymbol{\lambda}_k - \boldsymbol{\lambda}_{k+1}\|$$
$$\leq \rho M_g\|\mathbf{x}_{k+1} - \mathbf{x}_k\| + \rho\|\mathbf{u}_{k+1} - \mathbf{u}_k\| + \delta_k.$$

Therefore,

$$\|\boldsymbol{\zeta}_{\mathbf{u}}^{k+1}\| = \|\widetilde{\mathbf{u}}_{k+1} - \mathbf{u}_{k+1}\| \leq \rho M_g\|\mathbf{x}_{k+1} - \mathbf{x}_k\| + (\rho + 1/\tau)\|\mathbf{u}_{k+1} - \mathbf{u}_k\| + \delta_k. \tag{45}$$

Combining equation 42 and equation 45, we obtain

$$\|\boldsymbol{\zeta}_{\mathbf{p}}^{k+1}\| \leq d_1(\|\mathbf{x}_{k+1} - \mathbf{x}_k\| + \|\mathbf{u}_{k+1} - \mathbf{u}_k\|) + (M_g + 1)\delta_k,$$

where $d_1 = \max\{L_f + 1/\eta + \rho(M_g^2 + M_g) + 1/\eta, \ \rho(M_g + 1) + 1/\tau\}$. This inequality, along with $\boldsymbol{\zeta}_{\mathbf{p}}^{k+1} \in \partial_{\mathbf{p}}\mathcal{L}_{\alpha\beta}(\mathbf{w}_{k+1})$, yields the desired result. □

# B  Proofs of Asymptotic Convergence for Algorithm 1

## B.1  Proof of Primal Stationarity (Lemma 12)

*Proof.* From Lemma 34, we have

$$C_{\mathbf{p}}\left(\|\mathbf{x}_{k+1} - \mathbf{x}_k\|^2 + \|\mathbf{u}_{k+1} - \mathbf{u}_k\|^2\right) \leq \mathcal{L}_{\alpha\beta}(\mathbf{w}_k) - \mathcal{L}_{\alpha\beta}(\mathbf{w}_{k+1}) + \widehat{\delta}_k, \tag{46}$$

where $C_{\mathbf{p}} = \max\{C_1, C_2\}$. Using Lemma 35 and the fact $(a + b + c)^2 \leq 3(a^2 + b^2 + c^2)$, we have

$$\|\boldsymbol{\zeta}_{\mathbf{p}}^{k+1}\|^2 \leq 3d_1^2(\|\mathbf{x}_{k+1} - \mathbf{x}_k\|^2 + \|\mathbf{u}_{k+1} - \mathbf{u}_k\|^2) + 3(M_g + 1)^2\delta_k^2,$$

which, combined with equation 46, yields

$$\|\boldsymbol{\zeta}_{\mathbf{p}}^{k+1}\|^2 \leq \frac{3d_1^2}{C_{\mathbf{p}}}\left(\mathcal{L}_{\alpha\beta}(\mathbf{w}_k) - \mathcal{L}_{\alpha\beta}(\mathbf{w}_{k+1}) + \widehat{\delta}_k\right) + 3(M_g + 1)^2\delta_k^2.$$

Summing up the above inequalities over $k = 0, \dots, T - 1$, we obtain

$$\sum_{k=0}^{T-1} \|\boldsymbol{\zeta}_{\mathbf{p}}^{k+1}\|^2 \leq \frac{3d_1^2}{C_{\mathbf{p}}}\left(\mathcal{L}_{\alpha\beta}(\mathbf{w}_0) - \mathcal{L}_{\alpha\beta}(\mathbf{w}_T) + \sum_{k=0}^{T-1}\widehat{\delta}_k\right) + 3(M_g + 1)^2 \sum_{k=0}^{T-1}\delta_k^2$$

Since $\sum_{k=0}^{\infty} \delta_k^2 < +\infty$, we denote $B_\delta = \sum_{k=0}^{\infty} \delta_k^2$. Therefore,

$$
\begin{aligned}
&\frac{1}{T} \sum_{k=0}^{T-1} \|\boldsymbol{\zeta}_{\mathbf{p}}^{k+1}\|^2 \\
&\leq \frac{\frac{3d_1^2}{C_{\mathbf{p}}} \left( \mathcal{L}_{\alpha\beta}(\mathbf{w}_0) - \mathcal{L}_{\alpha\beta}(\mathbf{w}_T) \right)}{T} + \frac{\frac{3d_1^2}{C_{\mathbf{p}}} \sum_{k=0}^{T-1} \widehat{\delta}_k}{T} + \frac{3(M_g+1)^2 \sum_{k=0}^{T-1} \delta_k^2}{T} \\
&\leq \frac{\frac{3d_1^2}{C_{\mathbf{p}}} \left( \mathcal{L}_{\alpha\beta}(\mathbf{w}_0) - \underline{\mathcal{L}_{\alpha\beta}} \right)}{T} + \frac{\left( \frac{3d_1^2}{\rho^3 C_{\mathbf{p}}} + 3(M_g+1)^2 \right) \sum_{k=0}^{T-1} \delta_k^2}{T} + \frac{\frac{1}{\rho^2} \sum_{k=0}^{T-1} \delta_k}{T},
\end{aligned}
\tag{47}
$$

where the second inequality holds by the the lower boundedness of $\mathcal{L}_{\alpha\beta}(\mathbf{w}_k)$, denoted by $\underline{\mathcal{L}_{\alpha\beta}}$, that is from the boundedness of generated sequences, and $\widehat{\delta}_k = \frac{\delta_k^2}{\rho^3} + \frac{\delta_k}{\rho^2}$.

Note that given $\delta_k = \kappa \cdot (k+1)^{-1}$ and $\kappa > 0$, for sufficiently large $T$, we know that

$$
\sum_{k=0}^{T-1} \delta_k \approx \kappa^{-1} \log(\kappa T).
$$

Since the last term on the right-hand side (RHS) of equation 47 dominates the other terms and $T$ grows faster than $\log(T)$, the RHS of equation 47 decreases to 0 as $T$ increase. $\square$

## B.2 Proof of Primal Feasibility (Lemma 14)

*Proof.* Note that the update rule $\boldsymbol{\mu}_{k+1} = \boldsymbol{\mu}_k + \sigma_k(\boldsymbol{\lambda}_k - \boldsymbol{\mu}_k)$ can be rewritten as a convex combination $\boldsymbol{\mu}_{k+1} = (1 - \sigma_k)\boldsymbol{\mu}_k + \sigma_k\boldsymbol{\lambda}_k$. Since the step size $\sigma_k \in [0,1]$, the sequence $\{\boldsymbol{\mu}_k\}$ remains within the convex hull of the initialization $\boldsymbol{\mu}_0$ and the sequence $\{\boldsymbol{\lambda}_k\}$. Since $\{\boldsymbol{\lambda}_k\}$ is bounded (by Assumption 7), $\{\boldsymbol{\mu}_k\}$ is also bounded.

Now, from the $\boldsymbol{\mu}$-update equation 11, notice that $\boldsymbol{\mu}_{k+1} = \boldsymbol{\mu}_0 + \frac{1}{\rho} \sum_{t=0}^{k} \sigma_t(\boldsymbol{\lambda}_t - \boldsymbol{\mu}_t)$. Using the fact that $\|a\| - \|b\| \leq \|a + b\|$ for any $a, b \in \mathbb{R}^m$, we have

$$
\left\| \sum_{t=0}^{\infty} \sigma_t(\boldsymbol{\lambda}_t - \boldsymbol{\mu}_t) \right\| \leq \|\boldsymbol{\mu}_{k+1}\| + \|\boldsymbol{\mu}_0\| < +\infty,
\tag{48}
$$

where the last inequality hold by the boundedness of $\{\boldsymbol{\mu}_k\}$ from Assumption 7 together with the boundedness of sequence $\{(\boldsymbol{\lambda}_k - \boldsymbol{\mu}_k) := \rho(g(\mathbf{x}_k) + u_k)\}$.

Since $g(\mathbf{x}_k) + u_k$ is bounded, the $\boldsymbol{\lambda}$-update implies that the sequence $\{\boldsymbol{\lambda}_k - \boldsymbol{\mu}_k\}$ is bounded. Thus, it has a limit point, denoted by $\overline{\boldsymbol{\lambda}} - \overline{\boldsymbol{\mu}}$. Then, $\{\boldsymbol{\lambda}_k - \boldsymbol{\mu}_k\}$ has a bounded subsequence $\{\boldsymbol{\lambda}_{k_j} - \boldsymbol{\mu}_{k_j}\}_{j \in \mathbb{N}}$ converging to $\overline{\boldsymbol{\lambda}} - \overline{\boldsymbol{\mu}}$. Without loss of generality, by restricting our analysis to such a subsequence, we may assume that $\lim_{j \to \infty} \|\boldsymbol{\lambda}_{k_j} - \boldsymbol{\mu}_{k_j}\| = \|e\|$, where $e = \overline{\boldsymbol{\lambda}} - \overline{\boldsymbol{\mu}}$. Focusing on such a subsequence allows us to exclude oscillatory cases (such as $\boldsymbol{\lambda}_k - \boldsymbol{\mu}_k = (-1)^k$). Note that subsequence convergence to a stationary point is a standard guarantee in non-convex optimization Sun & Sun (2024); Boob et al. (2022); Zhang & Luo (2020).

We prove that the limit point satisfies $\overline{\boldsymbol{\lambda}} - \overline{\boldsymbol{\mu}} = 0$ by contradiction. Assume that the limit point is non-zero. Since $\sum_{k=0}^{\infty} \sigma_k = \infty$, we see that

$$
\left\| \sum_{k=0}^{\infty} \sigma_k(\boldsymbol{\lambda}_k - \boldsymbol{\mu}_k) \right\| = \infty,
$$

which contradicts equation 48. This contradiction leads to the desired result that $\overline{\boldsymbol{\lambda}} - \overline{\boldsymbol{\mu}} = 0$. It directly follows the definitions of $\boldsymbol{\lambda}_{k+1}$ and $u_{k+1}$ that

$$
0 = \frac{1}{\rho} \left( \overline{\boldsymbol{\lambda}} - \overline{\boldsymbol{\mu}} \right) = g(\overline{\mathbf{x}}) + \overline{u} \quad \text{and} \quad \overline{u} \geq 0.
$$

Hence, any limit point $\overline{\mathbf{x}}$ is feasible, namely, $g(\overline{\mathbf{x}}) \leq 0$.

Now consider the dual stationarity defined as $\boldsymbol{\zeta}_{\mathbf{d}}^{k+1} := (\zeta_{\boldsymbol{\lambda}}^{k+1}, \zeta_{\boldsymbol{\mu}}^{k+1}) \in \partial_{\mathbf{d}}\mathcal{L}_\rho(\mathbf{w}_{k+1})$. Since $\boldsymbol{\lambda}$ update step equation 13 is an exact maximization, $\boldsymbol{\zeta}_{\boldsymbol{\lambda}}^{k+1} = 0$. Due to the convergence of the subsequence $\|\boldsymbol{\lambda}_{k_j} - \boldsymbol{\mu}_{k_j}\| \to 0$, we have $\boldsymbol{\zeta}_{\boldsymbol{\mu}}^{k_j+1} \to 0$ as well by the $\boldsymbol{\mu}$ update step equation 11. $\qquad\square$

### B.3 Proof of Dual Feasibility (Lemma 15)

*Proof.* By the update rule *equation* 10, $\mathbf{u} \in \mathbb{R}_+^m$ and $\bar{\mathbf{u}} \geq \mathbf{0}$. And the stationarity condition with respect to $\mathbf{u}$ implies that a fixed point $\bar{\mathbf{u}}$ must satisfy

$$\bar{\mathbf{u}} = \Pi_{\mathbb{R}_+^m}[\bar{\mathbf{u}} - \tau\bar{\boldsymbol{\lambda}}] \iff \langle\bar{\boldsymbol{\lambda}}, \mathbf{u} - \bar{\mathbf{u}}\rangle \geq 0, \quad \forall\mathbf{u} \in \mathbb{R}_+^m.$$

This condition can be stated as

$$\mathbf{0} \in \bar{\boldsymbol{\lambda}} + \mathcal{N}_{\mathbb{R}_+^m}(\bar{\mathbf{u}}) \iff -\bar{\boldsymbol{\lambda}} \in \mathcal{N}_{\mathbb{R}_+^m}(\bar{\mathbf{u}}), \tag{49}$$

where $\mathcal{N}_{\mathbb{R}_+^m}(\bar{\mathbf{u}})$ is the normal cone to $\mathbb{R}_+^m$ at the limit point $\bar{\mathbf{u}}$. For any component $j$, if $\bar{\mathbf{u}}_j > \mathbf{0}$, the point is in the interior of the set along this axis, and the normal cone contains only the zero vector, i.e., $(\mathcal{N}_{\mathbb{R}_+^m}(\bar{\mathbf{u}}))_j = \{\mathbf{0}\}$. Thus, equation 49 implies that if $\bar{\mathbf{u}}_j > \mathbf{0}$, we must have $\bar{\boldsymbol{\lambda}}_j = \mathbf{0}$. If $\bar{\mathbf{u}}_j = \mathbf{0}$, the point is at the boundary, and the normal cone consists of all non-positive scalars, i.e., $(\mathcal{N}_{\mathbb{R}_+^m}(\bar{\mathbf{u}}))_j = (-\infty, 0]$. In any case, we conclude that $\bar{\boldsymbol{\lambda}} \geq \mathbf{0}$. $\qquad\square$

### B.4 Proof of Complementary Slackness (Lemma 16)

*Proof.* We prove that for each component $j$, $\bar{\boldsymbol{\lambda}}_j g_j(\bar{\mathbf{x}}) = 0$. Consider two cases for $\bar{\mathbf{u}}_j \geq 0$, where the non-negativity follows from the update rule equation 10. First, if $\bar{\mathbf{u}}_j > \mathbf{0}$, $\bar{\boldsymbol{\lambda}}_j = \mathbf{0}$ by the stationarity condition used in the proof of Lemma 15. Thus,

$$\bar{\boldsymbol{\lambda}}_j g_j(\bar{\mathbf{x}}) = 0 \cdot g_j(\bar{\mathbf{x}}) = 0.$$

Second, if $\bar{\mathbf{u}}_j = 0$, from Lemma 14, $g_j(\bar{\mathbf{x}}) = -\bar{\mathbf{u}}_j = 0$. Thus,

$$\bar{\boldsymbol{\lambda}}_j g_j(\bar{\mathbf{x}}) = \bar{\boldsymbol{\lambda}}_j \cdot 0 = 0,$$

concluding the proof. $\qquad\square$

## C  Proofs of Non-asymptotic Convergence Rate for Algorithm 1

### C.1  Proof of Primal Stationarity Convergence Rate (Lemma 18)

*Proof.* The rate of average residual convergence established in Lemma 12 corresponds to using $\boldsymbol{\lambda}_k$. From the definition of primal residual,

$$\boldsymbol{\zeta}_{\mathbf{x}}^{k+1} = \nabla f(\mathbf{x}_{k+1}) - \nabla f(\mathbf{x}_k) + \partial g(\mathbf{x}_{k+1})^\top(\boldsymbol{\lambda}_{k+1} - \boldsymbol{\lambda}_k) - \frac{1}{\eta}(\mathbf{x}_{k+1} - \mathbf{x}_k).$$

And the optimality condition for the update of $\mathbf{x}_{k+1}$ in equation 9 gives

$$-(\nabla f(\mathbf{x}_k) + \partial g(\mathbf{x}_{k+1})^\top\boldsymbol{\lambda}_k + \frac{1}{\eta}(\mathbf{x}_{k+1} - \mathbf{x}_k)) \in \partial r(\mathbf{x}_{k+1}).$$

Rearranging this gives an expression for the stationarity residual with respect to $\boldsymbol{\lambda}_{k+1}$:

$$\boldsymbol{\zeta}_{\mathbf{x}}^{k+1} \in \nabla f(\mathbf{x}_{k+1}) + \partial r(\mathbf{x}_{k+1}) + \partial g(\mathbf{x}_{k+1})^\top\boldsymbol{\lambda}_{k+1} \tag{50}$$

The difference between the stationarity residuals for $\boldsymbol{\lambda}_k$ and $\boldsymbol{\nu}_k$ is bounded by $\|\partial g(\bar{\mathbf{x}}_T)^\top(\bar{\boldsymbol{\nu}}_T - \bar{\boldsymbol{\lambda}}_T)\|$, where $\bar{\mathbf{x}}_T = \frac{1}{T}\sum\mathbf{x}_k$ and $\bar{\boldsymbol{\lambda}}_T = \frac{1}{T}\sum\boldsymbol{\lambda}_k$. The difference in the averaged multipliers is:

$$\bar{\boldsymbol{\nu}}_T - \bar{\boldsymbol{\lambda}}_T = \frac{1}{T}\sum_{k=0}^{T-1}(\boldsymbol{\nu}_k - \boldsymbol{\lambda}_k) = \frac{1}{T}\sum_{k=0}^{T-1}\frac{1}{\tau}(\mathbf{u}_{k+1} - \mathbf{u}_k) = \frac{1}{\tau T}(\mathbf{u}_T - \mathbf{u}_0). \tag{51}$$

Since the iterates $\{\mathbf{u}_k\}$ are bounded, $\|\bar{\boldsymbol{\nu}}_T - \bar{\boldsymbol{\lambda}}_T\| = \mathcal{O}(1/T)$. This difference vanishes faster than the stationarity residual itself, which converges at $\tilde{\mathcal{O}}(1/\sqrt{T})$. Therefore, the stationarity condition holds for $\bar{\boldsymbol{\nu}}_T$ with the same convergence rate. $\qquad\square$

### C.2 Proof of Primal Feasibility Convergence Rate (Lemma 19)

*Proof.* The rate of convergence for primal feasibility is independent of the choice of multiplier and can be quantified by leveraging the convergence rate of the dual variables. From the update rule equation 13 for $\boldsymbol{\lambda}_{k+1}$, we have the relation $g(\mathbf{x}_{k+1}) + \mathbf{u}_{k+1} = \frac{1}{\rho}(\boldsymbol{\lambda}_{k+1} - \boldsymbol{\mu}_{k+1})$. Since $\mathbf{u}_{k+1} \geq \mathbf{0}$, the norm of the primal feasibility violation, $[g(\mathbf{x}_{k+1})]^+ = \max\{\mathbf{0}, g(\mathbf{x}_{k+1})\}$, is bounded:

$$\|[g(\mathbf{x}_{k+1})]^+\| \leq \|g(\mathbf{x}_{k+1}) + \mathbf{u}_{k+1}\| = \frac{1}{\rho}\|\boldsymbol{\lambda}_{k+1} - \boldsymbol{\mu}_{k+1}\|. \tag{52}$$

This inequality holds because for any component $j$, if $g_j(\mathbf{x}_{k+1}) \leq \mathbf{0}$, then $([g(\mathbf{x}_{k+1})]^+)_j^2 = \mathbf{0} \leq (g_j(\mathbf{x}_{k+1}) + \mathbf{u}_{j,k+1})^2$. If $g_j(\mathbf{x}_{k+1}) > \mathbf{0}$, then since $\mathbf{u}_{j,k+1} \geq \mathbf{0}$, we have $([g(\mathbf{x}_{k+1})]^+)_j^2 = g_j(\mathbf{x}_{k+1})^2 \leq (g_j(\mathbf{x}_{k+1}) + \mathbf{u}_{j,k+1})^2$. Summing over all components yields the squared norm inequality.

Now we establish that $\frac{1}{T}\sum_{k=0}^{T-1}\|\boldsymbol{\lambda}_{k+1} - \boldsymbol{\mu}_{k+1}\|^2 = \tilde{\mathcal{O}}(1/T)$. From the $\boldsymbol{\lambda}$-update, we know that

$$\|\boldsymbol{\lambda}_{k+1} - \boldsymbol{\mu}_{k+1}\| = \rho\|g(\mathbf{x}_{k+1}) + \mathbf{u}_{k+1}\|.$$

Let $(\mathbf{x}^*, \mathbf{u}^*) \in X^* \times U^*$ be a feasible point pair, where $X^*$ and $U^*$ denotes the optimal sets, then $g(\mathbf{x}^*) + \mathbf{u}^* = 0$. By Assumption 5 and triangle's inequality,

$$\|g(\mathbf{x}_{k+1}) + \mathbf{u}_{k+1}\| = \|g(\mathbf{x}_{k+1}) + \mathbf{u}_{k+1} - (g(\mathbf{x}^*) + \mathbf{u}^*)\|$$
$$\leq L_g\|\mathbf{x}_{k+1} - \mathbf{x}^*\| + \|\mathbf{u}_{k+1} - \mathbf{u}^*\|.$$

By the local error assumption, for sufficiently large $k$, we have

$$\text{dist}((\mathbf{x}_{k+1}, \mathbf{u}_{k+1}), X^* \times U^*) \leq \kappa \cdot \|\zeta_p^{k+1}\|.$$

Using Lemmas 35 and 39 and applying the inequality $(a + b)^2 \leq 2a^2 + 2b^2$, we have

$$\|\boldsymbol{\lambda}_{k+1} - \boldsymbol{\mu}_{k+1}\|^2 \leq C\left(d_1\left(\|\mathbf{x}_{k+1} - \mathbf{x}_k\| + \|\mathbf{u}_{k+1} - \mathbf{u}_k\|\right) + (M_g + 1)\delta_k\right)^2$$
$$\leq C\left(2d_1^2(\|\mathbf{x}_{k+1} - \mathbf{x}_k\| + \|\mathbf{u}_{k+1} - \mathbf{u}_k\|)^2 + 2(M_g + 1)^2\delta_k^2\right)$$
$$\leq C\left(4d_1^2(\|\mathbf{x}_{k+1} - \mathbf{x}_k\|^2 + \|\mathbf{u}_{k+1} - \mathbf{u}_k\|^2) + 2(M_g + 1)^2\delta_k^2\right),$$

for some constant $C$. Summing the inequality over $k = 0, \ldots, T - 1$,

$$\sum_{k=0}^{T-1}\|\boldsymbol{\lambda}_{k+1} - \boldsymbol{\mu}_{k+1}\|^2 \leq 4d_1^2 C \sum_{k=0}^{T-1}(\|\mathbf{x}_{k+1} - \mathbf{x}_k\|^2 + \|\mathbf{u}_{k+1} - \mathbf{u}_k\|^2) + 2(M_g + 1)^2 C \sum_{k=0}^{T-1}\delta_k^2.$$

Since Lemmas 34 and 38(a) give

$$\sum_{k=0}^{T-1}\|\mathbf{x}_{k+1} - \mathbf{x}_k\|^2 + \|\mathbf{u}_{k+1} - \mathbf{u}_k\|^2 \leq \mathcal{L}(\mathbf{w}_0) - \mathcal{L}(\mathbf{w}_T) + \sum_{k=0}^{T-1}\hat{\delta}_k.$$

Therefore we have the desired rate of

$$\frac{1}{T}\sum_{k=0}^{T-1}\|\boldsymbol{\lambda}_{k+1} - \boldsymbol{\mu}_{k+1}\|^2 = \tilde{\mathcal{O}}(1/T).$$

This allows us to bound the running average of the squared primal feasibility violation:

$$\frac{1}{T}\sum_{k=0}^{T-1}\|[g(\mathbf{x}_{k+1})]^+\|^2 \leq \frac{1}{\rho^2 T}\sum_{k=0}^{T-1}\|\boldsymbol{\lambda}_{k+1} - \boldsymbol{\mu}_{k+1}\|^2 = \tilde{\mathcal{O}}\left(\frac{\log(T)}{T}\right) = \tilde{\mathcal{O}}\left(\frac{1}{T}\right). \tag{53}$$

By applying Jensen's inequality, we find that the average primal feasibility violation converges at a rate of $\tilde{\mathcal{O}}(1/\sqrt{T})$. $\qquad\square$

### C.3 Proof of Complementary Slackness Convergence Rate (Lemma 20)

*Proof.* From the optimality of $\mathbf{u}$-update equation 10 and the construction of $\boldsymbol{\nu}$, $\mathbf{u}_{k+1} = \Pi_{\mathbb{R}^m_+}[\mathbf{u}_{k+1} - \tau\boldsymbol{\nu}_k]$ implies the complementarity condition $\langle\boldsymbol{\nu}_k, \mathbf{u} - \mathbf{u}_{k+1}\rangle \geq 0$ for all $\mathbf{u} \geq \mathbf{0}$. By choosing $\mathbf{u} = \mathbf{0}$, we have $-\langle\boldsymbol{\nu}_k, \mathbf{u}_{k+1}\rangle \geq 0$. Since both $\boldsymbol{\nu}_k$ and $\mathbf{u}_{k+1}$ are non-negative vectors, they are orthogonal component-wise:

$$\nu_{j,k}u_{j,k+1} = 0 \quad \forall j \in [m]. \tag{54}$$

Now consider the approximate complementary slackness at each iteration defined as Definition 2. By the $\boldsymbol{\lambda}$-update equation 13 and equation 54,

$$\nu_{j,k}g_j(\mathbf{x}_{k+1}) = \nu_{j,k}\left(-u_{j,k+1} + \frac{1}{\rho}(\lambda_{j,k+1} - \mu_{j,k+1})\right) = \frac{\nu_{j,k}}{\rho}(\lambda_{j,k+1} - \mu_{j,k+1}).$$

By applying Cauchy-Schwarz inequality on the approximate complementary slackness,

$$\sum_{j=1}^m |\nu_{j,k}g_j(\mathbf{x}_{k+1})| \leq \frac{1}{\rho}\sum_{j=1}^m |\nu_{j,k}||\lambda_{j,k+1} - \mu_{j,k+1}| \leq \frac{1}{\rho}\|\boldsymbol{\nu}_k\|\|\boldsymbol{\lambda}_{k+1} - \boldsymbol{\mu}_{k+1}\|. \tag{55}$$

Since $\{\boldsymbol{\lambda}_k\}$ and $\{\mathbf{u}_k\}$ are bounded, $\{\boldsymbol{\nu}_k\}$ is bounded by a constant $B_\nu := \max_{k\geq 1}\{\boldsymbol{\nu}_k\}$. Thus,

$$\sum_{j=1}^m |\nu_{j,k}g_j(\mathbf{x}_{k+1})| \leq \frac{B_\nu}{\rho}\|\boldsymbol{\lambda}_{k+1} - \boldsymbol{\mu}_{k+1}\|.$$

Now, we can analyze the running average:

$$\frac{1}{T}\sum_{k=0}^{T-1}\sum_{j=1}^m |\nu_{j,k}g_j(\mathbf{x}_{k+1})| \leq \frac{B_\nu}{\rho T}\sum_{k=0}^{T-1}\|\boldsymbol{\lambda}_{k+1} - \boldsymbol{\mu}_{k+1}\|.$$

Using Jensen's inequality and the established rate for the dual residual from , we get:

$$\frac{1}{T}\sum_{k=0}^{T-1}\|\boldsymbol{\lambda}_{k+1} - \boldsymbol{\mu}_{k+1}\| \leq \sqrt{\frac{1}{T}\sum_{k=0}^{T-1}\|\boldsymbol{\lambda}_{k+1} - \boldsymbol{\mu}_{k+1}\|^2} = \sqrt{\tilde{\mathcal{O}}\left(\frac{1}{T}\right)} = \tilde{\mathcal{O}}\left(\frac{1}{\sqrt{T}}\right).$$

This then establishes that the average of the sum of absolute values converges at the required rate of $\tilde{\mathcal{O}}(1/\sqrt{T})$. $\qquad\square$

## D   Supporting Lemmas for Convergence Analysis of Algorithm 2

Now, we establish several key properties of Algorithm 2, including important relations among the primal and dual sequences, the approximate decrease of the Proximal-Perturbed Augmented Lagrangian ($L_\rho$), and error bounds for its subgradient in the primal variables. These intermediate results are crucial stepping stones for proving the algorithm's overall convergence to an $\epsilon$-KKT point. We first provide basic yet crucial relations on the sequences $\boldsymbol{\lambda}^k$, $\boldsymbol{\mu}^k$, and $\mathbf{x}^k$.

**Lemma 36.** *Let $\{(\mathbf{x}_k, \mathbf{u}_k, \mathbf{z}_k, \boldsymbol{\lambda}_k, \boldsymbol{\mu}_k)\}$ be the sequence generated by Algorithm 2 with the choice of the sequence $\{\delta_k\}$ as in equation 23. Under Assumption 6, for any $k \geq 1$, the following relations hold:*

$$\|\boldsymbol{\mu}_{k+1} - \boldsymbol{\mu}_k\|^2 = \sigma_k^2\|\boldsymbol{\lambda}_k - \boldsymbol{\mu}_k\|^2 \leq \delta_k^2/4, \tag{56a}$$

$$\sigma_k\|\boldsymbol{\lambda}_k - \boldsymbol{\mu}_k\|^2 \leq \delta_k, \tag{56b}$$

$$\|\boldsymbol{\mu}_{k+1} - \boldsymbol{\lambda}_k\|^2 = (1 - \sigma_k)^2\|\boldsymbol{\lambda}_k - \boldsymbol{\mu}_k\|^2, \tag{56c}$$

$$\|\boldsymbol{\lambda}_{k+1} - \boldsymbol{\lambda}_k\|^2 \leq 3\rho^2 M_g^2\|\mathbf{x}_{k+1} - \mathbf{x}_k\|^2 + 3\rho^2\|\mathbf{u}_{k+1} - \mathbf{u}_k\|^2 + 3\delta_k^2/4, \tag{56d}$$

*where $M_g$ denotes the Lipschitz constant of $g$ from equation 3.*

*Proof.* Relations equation 56a, equation 56c and equation 56d follow the same proofs as equation 33a, equation 33b and equation 33c, respectively.

By the definitions $\sigma_k = \frac{\delta_k}{\|\boldsymbol{\lambda}_k - \boldsymbol{\mu}_k\|^2 + 1} \leq 1$ and $\delta_k \in (0, 1]$, we know that $\sigma_k \leq 1$. Thus, we obtain the relation equation 56b:

$$\sigma_k \|\boldsymbol{\lambda}_k - \boldsymbol{\mu}_k\|^2 = \frac{\delta_k}{1 + \frac{1}{\|\boldsymbol{\lambda}_k - \boldsymbol{\mu}_k\|^2}} \leq \delta_k.$$

Subtracting $\boldsymbol{\mu}_{k+1}$ from $\boldsymbol{\lambda}_k$ yields

$$\|\boldsymbol{\lambda}_k - \boldsymbol{\mu}_{k+1}\| = \|\boldsymbol{\lambda}_k - \boldsymbol{\mu}_k - \sigma_k(\boldsymbol{\lambda}_k - \boldsymbol{\mu}_k)\| = (1 - \sigma_k)\|\boldsymbol{\lambda}_k - \boldsymbol{\mu}_k\|.$$

Squaring both sides of the inequality yields the relation equation 56c. □

The relations in Lemma 36 are critical to our technique for proving convergence, bypassing the need for the surjectivity of the Jacobian $\nabla g(\mathbf{x})$ (or subgradient mapping $\partial g(\mathbf{x})$) as in Bolte et al. (2018); Boţ & Nguyen (2020).

**Lemma 37.** *Let $\{\mathbf{x}^k\}$ be the sequence generated by Algorithm 2. Under Assumptions 4, 5 and 6, there exists a constant $L_\ell > 0$ such that*

$$\ell_\rho(\mathbf{x}_{k+1}) \leq \ell_\rho(\mathbf{x}_k) + \langle \nabla_\mathbf{x} \ell_\rho(\mathbf{x}_k), \mathbf{x}_{k+1} - \mathbf{x}_k \rangle + \frac{L_\ell}{2}\|\mathbf{x}_{k+1} - \mathbf{x}_k\|^2, \tag{57}$$

*where $L_\ell := L_f + L_g B_{\boldsymbol{\lambda}} + \rho(L_g B_\mathbf{u} + L_g B_g + M_g^2)$ with $B_{\boldsymbol{\lambda}} = \max_{k \geq 0} \|\boldsymbol{\lambda}_k\|$, $B_\mathbf{u} = \max_{k \geq 0} \|\mathbf{u}_k\|$, $B_g = \max_{\mathbf{x} \in \mathrm{dom}(r)} \|g(\mathbf{x})\|$ and $M_g = \max_{\mathbf{x} \in \mathrm{dom}(r)} \|\nabla g(\mathbf{x})\|$ from equation 3.*

In the statement of Lemma 37, we omitted $(\mathbf{u}_k, \mathbf{z}_k, \boldsymbol{\lambda}_k, \boldsymbol{\mu}_k)$ in the argument of $\ell_\rho(\cdot)$ for simplicity.

*Proof.* Note that $\nabla_\mathbf{x} \ell_\rho(\mathbf{x}, \mathbf{u}, \mathbf{z}, \boldsymbol{\lambda}, \boldsymbol{\mu}) = \nabla f(\mathbf{x}) + \nabla g(\mathbf{x})(\boldsymbol{\lambda} + \rho(g(\mathbf{x}) + \mathbf{u}))$. A direct computation gives

$$\begin{aligned}
\|\nabla_\mathbf{x} \ell_\rho(\mathbf{x}_{k+1}) - \nabla_\mathbf{x} \ell_\rho(\mathbf{x}_k)\| &\leq \|\nabla f(\mathbf{x}_{k+1}) - \nabla f(\mathbf{x}_k)\| + \|(\nabla g(\mathbf{x}_{k+1}) - \nabla g(\mathbf{x}_k))(\boldsymbol{\lambda}_k + \rho \mathbf{u}_k)\| \\
&\quad + \rho\|\nabla g(\mathbf{x}_{k+1})g(\mathbf{x}_{k+1}) - \nabla g(\mathbf{x}_k)g(\mathbf{x}_{k+1})\| \\
&\quad + \rho\|\nabla g(\mathbf{x}_k)g(\mathbf{x}_{k+1}) - \nabla g(\mathbf{x}_k)g(\mathbf{x}_k)\| \\
&\leq L_f\|\mathbf{x}_{k+1} - \mathbf{x}_k\| + L_g(B_{\boldsymbol{\lambda}} + \rho B_\mathbf{u})\|\mathbf{x}_{k+1} - \mathbf{x}_k\| \\
&\quad + \rho L_g B_g\|\mathbf{x}_{k+1} - \mathbf{x}_k\| + \rho M_g^2\|\mathbf{x}_{k+1} - \mathbf{x}_k\| \\
&\leq \left(L_f + L_g B_{\boldsymbol{\lambda}} + \rho(L_g B_\mathbf{u} + L_g B_g + M_g^2)\right)\|\mathbf{x}_{k+1} - \mathbf{x}_k\|.
\end{aligned}$$

Hence, by the descent lemma (Bertsekas, 1999, Proposition A.24), we obtain the desired result. □

Now, we establish key properties that lead to our main convergence results.

**Lemma 38.** *Let the sequence $\{\mathbf{w}_k\}$ be generated by Algorithm 2. Under Assumptions 4-7, the Proximal-Perturbed Augmented Lagrangian $\mathcal{L}_\rho$ equation 17 satisfies:*

*(a)* **(Approximate Decrease of $\mathcal{L}_\rho$)**

$$\mathcal{L}_\rho(\mathbf{w}_{k+1}) - \mathcal{L}_\rho(\mathbf{w}_k) \leq -c_1\|\mathbf{x}_{k+1} - \mathbf{x}_k\|^2 - c_2\|\mathbf{u}_{k+1} - \mathbf{u}_k\|^2 + \widehat{\delta}_k,$$

*where $c_1 = \frac{1}{2}\left(\frac{1}{\eta} - L_\ell - 3\rho M_g^2\right) > 0$, $c_2 = \left(\frac{1}{\tau} - 2\rho\right) > 0$, and $\widehat{\delta}_k := \frac{\delta_k^2}{4\rho} + \frac{\delta_k}{\rho}$.*

*(b)* **(Convergence of $\mathcal{L}_\rho$)** *the sequence $\{\mathcal{L}_\rho(\mathbf{w}_k)\}$ is convergent, i.e., $\lim_{k \to \infty} \mathcal{L}_\rho(\mathbf{w}_{k+1}) := \underline{\mathcal{L}}_\rho > -\infty$.*

*Proof.* (a) The difference between two consecutive sequences of $\mathcal{L}_\rho$ can be divided into four parts:

$$\mathcal{L}_\rho(\mathbf{w}_{k+1}) - \mathcal{L}_\rho(\mathbf{w}_k) = [\mathcal{L}_\rho(\mathbf{x}_{k+1}, \mathbf{u}_k, \mathbf{z}_k, \boldsymbol{\lambda}_k, \boldsymbol{\mu}_k) - \mathcal{L}_\rho(\mathbf{w}_k)] \tag{58a}$$
$$+ [\mathcal{L}_\rho(\mathbf{x}_{k+1}, \mathbf{u}_{k+1}, \mathbf{z}_k, \boldsymbol{\lambda}_k, \boldsymbol{\mu}_k) - \mathcal{L}_\rho(\mathbf{x}_{k+1}, \mathbf{u}_k, \mathbf{z}_k, \boldsymbol{\lambda}_k, \boldsymbol{\mu}_k)] \tag{58b}$$
$$+ [\mathcal{L}_\rho(\mathbf{x}_{k+1}, \mathbf{u}_{k+1}, \mathbf{z}_k, \boldsymbol{\lambda}_{k+1}, \boldsymbol{\mu}_{k+1}) - \mathcal{L}_\rho(\mathbf{x}_{k+1}, \mathbf{u}_{k+1}, \mathbf{z}_k, \boldsymbol{\lambda}_k, \boldsymbol{\mu}_k)] \tag{58c}$$
$$+ [\mathcal{L}_\rho(\mathbf{w}_{k+1}) - \mathcal{L}_\rho(\mathbf{x}_{k+1}, \mathbf{u}_{k+1}, \mathbf{z}_k, \boldsymbol{\lambda}_{k+1}, \boldsymbol{\mu}_{k+1})]. \tag{58d}$$

First, we consider equation 58a. Writing $\mathcal{L}_\rho(\mathbf{x}_{k+1}) = \mathcal{L}_\rho(\mathbf{x}_{k+1}, \mathbf{u}_k, \mathbf{z}_k, \boldsymbol{\lambda}_k, \boldsymbol{\mu}_k)$, and using Lemma 37, we have

$$\ell_\rho(\mathbf{x}_{k+1}) \leq \ell_\rho(\mathbf{x}_k) + \langle \nabla_{\mathbf{x}} \ell_\rho(\mathbf{x}_k), \mathbf{x}_{k+1} - \mathbf{x}_k \rangle + \frac{L_\ell}{2} \|\mathbf{x}_{k+1} - \mathbf{x}_k\|^2. \tag{59}$$

From the definition of $\mathbf{x}_{k+1}$ in equation 22, it follows that

$$\mathcal{L}_\rho(\mathbf{x}_k) \geq \ell_\rho(\mathbf{x}_k) + \langle \nabla_x \ell_\rho(\mathbf{x}_k), \mathbf{x}_{k+1} - \mathbf{x}_k \rangle + \frac{1}{2\eta} \|\mathbf{x}_{k+1} - \mathbf{x}_k\|^2 + r(\mathbf{x}_{k+1}),$$

implying $\langle \nabla_x \ell_\rho(\mathbf{x}_k), \mathbf{x}_{k+1} - \mathbf{x}_k \rangle + r(\mathbf{x}_{k+1}) \leq -\frac{1}{2\eta} \|\mathbf{x}_{k+1} - \mathbf{x}_k\|^2 + r(\mathbf{x}_k)$. Combining the this expression with equation 59 yields

$$\mathcal{L}_\rho(\mathbf{x}_{k+1}, \mathbf{u}_k, \mathbf{z}_k, \boldsymbol{\lambda}_k, \boldsymbol{\mu}_k) - \mathcal{L}_\rho(\mathbf{x}_k, \mathbf{u}_k, \mathbf{z}_k, \boldsymbol{\lambda}_k, \boldsymbol{\mu}_k) \leq -\frac{1}{2} \left( \frac{1}{\eta} - L_\ell \right) \|\mathbf{x}_{k+1} - \mathbf{x}_k\|^2. \tag{60}$$

Next, consider the second part equation 58b. Noting that $\nabla_{\mathbf{u}} \mathcal{L}_\rho$ is $\rho$-Lipschitz continuous, we have

$$\mathcal{L}_\rho(\mathbf{u}_{k+1}) \leq \mathcal{L}_\rho(\mathbf{u}_k) + \langle \nabla_{\mathbf{u}} \mathcal{L}_\rho(\mathbf{u}_k), \mathbf{u}_{k+1} - \mathbf{u}_k \rangle + \frac{\rho}{2} \|\mathbf{u}_{k+1} - \mathbf{u}_k\|^2.$$

By using the property of the projection operator, $\langle \Pi_{[0,U]}[\mathbf{a}] - \mathbf{a}, \mathbf{b} - \Pi_{[0,U]}[\mathbf{a}] \rangle \geq 0$ for $\mathbf{b} \in \Pi_{[0,U]}$, $\forall \mathbf{a} \in \mathbb{R}^m$, with $\mathbf{a} = \mathbf{u}_k - \tau \nabla_{\mathbf{u}} \mathcal{L}_\rho(\mathbf{u}_k)$, and $\mathbf{b} = \mathbf{u}_k$, we get

$$\langle \mathbf{u}_{k+1} - \mathbf{u}_k + \tau \nabla_{\mathbf{u}} \mathcal{L}_\rho(\mathbf{u}_k), \mathbf{u}_k - \mathbf{u}_{k+1} \rangle \geq 0,$$

from which we have $\langle \nabla_{\mathbf{u}} \mathcal{L}_\rho(\mathbf{u}_k), \mathbf{u}_{k+1} - \mathbf{u}_k \rangle \leq -\frac{1}{\tau} \|\mathbf{u}_{k+1} - \mathbf{u}_k\|^2$. Therefore,

$$\mathcal{L}_\rho(\mathbf{x}_{k+1}, \mathbf{u}_{k+1}, \mathbf{z}_k, \boldsymbol{\lambda}_k, \boldsymbol{\mu}_k) - \mathcal{L}_\rho(\mathbf{x}_{k+1}, \mathbf{u}_k, \mathbf{z}_k, \boldsymbol{\lambda}_k, \boldsymbol{\mu}_k) \leq - \left( \frac{1}{\tau} - \frac{\rho}{2} \right) \|\mathbf{u}_{k+1} - \mathbf{u}_k\|^2. \tag{61}$$

Now consider equation 58c. We start by noting that

$$\mathcal{L}_\rho(\mathbf{x}_{k+1}, \mathbf{u}_{k+1}, \mathbf{z}_k, \boldsymbol{\lambda}_{k+1}, \boldsymbol{\mu}_{k+1}) - \mathcal{L}_\rho(\mathbf{x}_{k+1}, \mathbf{u}_{k+1}, \mathbf{z}_k, \boldsymbol{\lambda}_k, \boldsymbol{\mu}_k)$$
$$= \underbrace{\langle \boldsymbol{\lambda}_{k+1} - \boldsymbol{\lambda}_k, g(\mathbf{x}_{k+1}) + \mathbf{u}_{k+1} \rangle}_{\text{(I)}} + \underbrace{\langle (\boldsymbol{\lambda}_k - \boldsymbol{\mu}_k) - (\boldsymbol{\lambda}_{k+1} - \boldsymbol{\mu}_{k+1}), \mathbf{z}_k \rangle}_{\text{(II)}} \tag{62}$$
$$- \frac{\beta}{2} \|\boldsymbol{\lambda}_{k+1} - \boldsymbol{\mu}_{k+1}\|^2 + \frac{\beta}{2} \|\boldsymbol{\lambda}_k - \boldsymbol{\mu}_k\|^2.$$

Using the updates $\boldsymbol{\lambda}_{k+1} = \boldsymbol{\mu}_{k+1} + \rho(g(\mathbf{x}_{k+1}) + \mathbf{u}_{k+1})$ and $\mathbf{z}_k = \frac{1}{\alpha}(\boldsymbol{\lambda}_k - \boldsymbol{\mu}_k)$, and the fact that $\langle \mathbf{a} - \mathbf{b}, \mathbf{a} \rangle = \frac{1}{2} \|\mathbf{a} - \mathbf{b}\|^2 + \frac{1}{2} \|\mathbf{a}\|^2 - \frac{1}{2} \|\mathbf{b}\|^2$ with $\mathbf{a} = \boldsymbol{\lambda}_k - \boldsymbol{\mu}_k$ and $\mathbf{b} = \boldsymbol{\lambda}_{k+1} - \boldsymbol{\mu}_{k+1}$, we have

$$\text{(I)} = \frac{1}{2\rho} \|\boldsymbol{\lambda}_{k+1} - \boldsymbol{\lambda}_k\|^2 + \frac{1}{2\rho} \|\boldsymbol{\lambda}_{k+1} - \boldsymbol{\mu}_{k+1}\|^2 - \frac{1}{2\rho} \|\boldsymbol{\mu}_{k+1} - \boldsymbol{\lambda}_k\|^2,$$
$$\text{(II)} = \frac{1}{2\alpha} \|(\boldsymbol{\lambda}_{k+1} - \boldsymbol{\mu}_{k+1}) - (\boldsymbol{\lambda}_k - \boldsymbol{\mu}_k)\|^2 + \frac{1}{2\alpha} \|\boldsymbol{\lambda}_k - \boldsymbol{\mu}_k\|^2 - \frac{1}{2\alpha} \|\boldsymbol{\lambda}_{k+1} - \boldsymbol{\mu}_{k+1}\|^2 \tag{63}$$
$$= \frac{\alpha}{2} \|\mathbf{z}_{k+1} - \mathbf{z}_k\|^2 + \frac{1}{2\alpha} \|\boldsymbol{\lambda}_k - \boldsymbol{\mu}_k\|^2 - \frac{1}{2\alpha} \|\boldsymbol{\lambda}_{k+1} - \boldsymbol{\mu}_{k+1}\|^2.$$

Substituting equation 63 into equation 62 yields

$$
\begin{aligned}
&\mathcal{L}_\rho(\mathbf{x}_{k+1}, \mathbf{u}_{k+1}, \mathbf{z}_k, \boldsymbol{\lambda}_{k+1}, \boldsymbol{\mu}_{k+1}) - \mathcal{L}_\rho(\mathbf{x}_{k+1}, \mathbf{u}_{k+1}, \mathbf{z}_k, \boldsymbol{\lambda}_k, \boldsymbol{\mu}_k) \\
&\leq \frac{1}{2\rho}\|\boldsymbol{\lambda}_{k+1} - \boldsymbol{\lambda}_k\|^2 - \frac{1}{2\rho}\|\boldsymbol{\mu}_{k+1} - \boldsymbol{\lambda}_k\|^2 + \frac{1}{2\rho}\|\boldsymbol{\lambda}_k - \boldsymbol{\mu}_k\|^2 + \frac{\alpha}{2}\|\mathbf{z}_{k+1} - \mathbf{z}_k\|^2 \\
&\overset{(i)}{\leq} \frac{1}{2\rho}\left(3\rho^2 M_g^2\|\mathbf{x}_{k+1} - \mathbf{x}_k\|^2 + 3\rho^2\|\mathbf{u}_{k+1} - \mathbf{u}_k\|^2 + 3\|\boldsymbol{\mu}_{k+1} - \boldsymbol{\mu}_k\|^2\right) \\
&\quad + \frac{1}{2\rho}\left(2\sigma_k - \sigma_k^2\right)\|\boldsymbol{\lambda}_k - \boldsymbol{\mu}_k\|^2 + \frac{\alpha}{2}\|\mathbf{z}_{k+1} - \mathbf{z}_k\|^2 \\
&\overset{(ii)}{\leq} \frac{1}{2}\left(3\rho M_g^2\|\mathbf{x}_{k+1} - \mathbf{x}_k\|^2 + 3\rho\|\mathbf{u}_{k+1} - \mathbf{u}_k\|^2\right) \\
&\quad + \frac{1}{2\rho}\left(2\sigma_k + 2\sigma_k^2\right)\|\boldsymbol{\lambda}_k - \boldsymbol{\mu}_k\|^2 + \frac{\alpha}{2}\|\mathbf{z}_{k+1} - \mathbf{z}_k\|^2 \\
&\overset{(iii)}{\leq} \frac{1}{2}\left(3\rho M_g^2\|\mathbf{x}_{k+1} - \mathbf{x}_k\|^2 + 3\rho\|\mathbf{u}_{k+1} - \mathbf{u}_k\|^2\right) + \frac{\delta_k^2}{4\rho} + \frac{\delta_k}{\rho} + \frac{\alpha}{2}\|\mathbf{z}_{k+1} - \mathbf{z}_k\|^2,
\end{aligned}
\tag{64}
$$

where (i) follows from equation 56c and equation 56d in Lemma 36; (ii) follows from equation 56a in Lemma 36; and (iii) is from equation 56a and equation 56b in Lemma 36.

Lastly, we consider equation 58d. Write down $\mathcal{L}_\rho(\mathbf{z}_{k+1}) = \mathcal{L}_\rho(\mathbf{x}_{k+1}, \mathbf{u}_{k+1}, \mathbf{z}_{k+1}, \boldsymbol{\lambda}_{k+1}, \boldsymbol{\mu}_{k+1})$ for notational simplicity. From the $\alpha$-strong convexity of $\mathcal{L}_\rho$ in $\mathbf{z}$, we have

$$
\mathcal{L}_\rho(\mathbf{z}_k) \geq \mathcal{L}_\rho(\mathbf{z}_{k+1}) + \langle \nabla_\mathbf{z}\mathcal{L}_\rho(\mathbf{z}_{k+1}), \mathbf{z}_k - \mathbf{z}_{k+1}\rangle + \frac{\alpha}{2}\|\mathbf{z}_{k+1} - \mathbf{z}_k\|^2.
$$

Since $\mathbf{z}_{k+1}$ minimizes $\mathcal{L}_\rho(\mathbf{x}_{k+1}, \mathbf{u}_{k+1}, \mathbf{z}, \boldsymbol{\lambda}_{k+1}, \boldsymbol{\mu}_{k+1})$, we have that $\nabla_\mathbf{z}\mathcal{L}_\rho(\mathbf{z}_{k+1}) = \mathbf{0}$. Thus,

$$
\mathcal{L}_\rho(\mathbf{z}_{k+1}) - \mathcal{L}_\rho(\mathbf{z}_k) \leq -\frac{\alpha}{2}\|\mathbf{z}_{k+1} - \mathbf{z}_k\|^2.
\tag{65}
$$

Combining equation 60, equation 61, equation 64, and equation 65 yields the desired result.

(b) By using the update of $\mathbf{z}_{k+1} = \frac{\boldsymbol{\lambda}_{k+1} - \boldsymbol{\mu}_{k+1}}{\alpha}$, we deduce

$$
\begin{aligned}
\mathcal{L}_\rho(\mathbf{w}_{k+1}) &= f(\mathbf{x}_{k+1}) + \langle \boldsymbol{\lambda}_{k+1}, g(\mathbf{x}_{k+1}) + \mathbf{u}_{k+1}\rangle \\
&\quad \underbrace{-\frac{1}{2\rho}\|\boldsymbol{\lambda}_{k+1} - \boldsymbol{\mu}_{k+1}\|^2 + \frac{\rho}{2}\|g(\mathbf{x}_{k+1}) + \mathbf{u}_{k+1}\|^2}_{=0} + r(\mathbf{x}_{k+1}) \\
&= f(\mathbf{x}_{k+1}) + \frac{1}{2\rho}\|\boldsymbol{\lambda}_{k+1}\|^2 + \frac{1}{2\rho}\|\boldsymbol{\lambda}_{k+1} - \boldsymbol{\mu}_{k+1}\|^2 - \frac{1}{2\rho}\|\boldsymbol{\mu}_{k+1}\|^2 + r(\mathbf{x}_{k+1}) > -\infty,
\end{aligned}
$$

where the last inequality holds by the boundedness of $\{\boldsymbol{\mu}_k\}$ (Assumption 7) and the lower boundedness of $f$ and $r$ over $\mathrm{dom}(r)$ (Assumption 6). Given the step sizes $0 < \eta < 1/(L_\ell + 3\rho M_g^2)$ and $0 < \tau < 1/2\rho$, we already know the sequence $\{\mathcal{L}_\rho(\mathbf{w}_{k+1})\}$ is approximately nonincreasing (Lemma 38(a)); Although it may not decrease monotonically at every step, it tends to decrease over iterations. As $\{\delta_k\}$ goes to 0 as $k \to \infty$, $\{\mathcal{L}_\rho(\mathbf{w}_{k+1})\}$ converges to a finite value $\underline{\mathcal{L}_\rho} > -\infty$. $\qquad\square$

**Lemma 39** (Subgradient Error Bound). *Let the sequence $\{\mathbf{w}_k := (\mathbf{x}_k, \mathbf{u}_k, \mathbf{z}_k, \boldsymbol{\lambda}_k, \boldsymbol{\mu}_k)\}$ be generated by Algorithm 2, and let $\{\mathbf{p}_k := (\mathbf{x}_k, \mathbf{u}_k, \mathbf{z}_k)\}$ be the primal sequence. Under Assumptions 4-7, there exists a constant $d_1 > 0$ such that for the primal subgradient $\boldsymbol{\zeta}_\mathbf{p}^{k+1} = (\zeta_\mathbf{x}^{k+1}, \zeta_\mathbf{u}^{k+1}, \mathbf{0}) \in \partial_\mathbf{p}\mathcal{L}_\rho(\mathbf{w}_{k+1})$,*

$$
\|\boldsymbol{\zeta}_\mathbf{p}^{k+1}\| \leq d_1\left(\|\mathbf{x}_{k+1} - \mathbf{x}_k\| + \|\mathbf{u}_{k+1} - \mathbf{u}_k\|\right) + (M_g + 1)\delta_k,
$$

*where*

$$
d_1 = \max\left\{L_f + B_{\boldsymbol{\lambda}}L_g + \rho(M_g + L_g(B_g + B_\mathbf{u}) + 2M_g^2) + 1/\eta,\ 2\rho(M_g + 1) + 1/\tau\right\}.
$$

*Proof.* From the proof of Lemma 35, we have that for all $k \geq 0$

$$\zeta_{\mathbf{x}}^{k+1} := \nabla_{\mathbf{x}}\ell_\rho(\mathbf{w}_{k+1}) - \nabla_{\mathbf{x}}\ell_\rho(\mathbf{w}_k) + \frac{1}{\eta}(\mathbf{x}_k - \mathbf{x}_{k+1}) \in \partial_{\mathbf{x}}\mathcal{L}_\rho(\mathbf{w}_{k+1}),$$

$$\zeta_{\mathbf{u}}^{k+1} := \mathbf{u}_{k+1} - \Pi_{[0,U]}[\mathbf{u}_{k+1} - (\boldsymbol{\lambda}_{k+1} + \rho(g(\mathbf{x}_{k+1}) + \mathbf{u}_{k+1}))] = \widetilde{\nabla}_{\mathbf{u}}\mathcal{L}_\rho(\mathbf{w}_{k+1}),$$

$$\nabla_{\mathbf{z}}\mathcal{L}_\rho(\mathbf{w}_{k+1}) = \alpha\mathbf{z}_{k+1} - (\boldsymbol{\lambda}_{k+1} - \boldsymbol{\mu}_{k+1}) = \mathbf{0}.$$

Hence, we obtain

$$\boldsymbol{\zeta}_{\mathbf{p}}^{k+1} := \begin{pmatrix} \zeta_{\mathbf{x}}^{k+1} & \in \partial_{\mathbf{x}}\mathcal{L}_\rho(\mathbf{x}_{k+1}, \mathbf{u}_{k+1}, \mathbf{z}_{k+1}, \boldsymbol{\lambda}_{k+1}, \boldsymbol{\mu}_{k+1}) \\ \zeta_{\mathbf{u}}^{k+1} & = \widetilde{\nabla}_{\mathbf{u}}\mathcal{L}_\rho(\mathbf{x}_{k+1}, \mathbf{u}_{k+1}, \mathbf{z}_{k+1}, \boldsymbol{\lambda}_{k+1}, \boldsymbol{\mu}_{k+1}) \\ \mathbf{0} & = \nabla_{\mathbf{z}}\mathcal{L}_\rho(\mathbf{x}_{k+1}, \mathbf{u}_{k+1}, \mathbf{z}_{k+1}, \boldsymbol{\lambda}_{k+1}, \boldsymbol{\mu}_{k+1}) \end{pmatrix}.$$

We derive upper estimates for $\zeta_{\mathbf{x}}^{k+1}$ and $\zeta_{\mathbf{u}}^{k+1}$. A straightforward calculation yields

$$
\begin{aligned}
\|\zeta_{\mathbf{x}}^{k+1}\| \leq & \|\nabla f(\mathbf{x}_{k+1}) - \nabla f(\mathbf{x}_k)\| + (1/\eta)\|\mathbf{x}_k - \mathbf{x}_{k+1}\| \\
& + \|\nabla g(\mathbf{x}_{k+1})(\boldsymbol{\lambda}_{k+1} + \rho(g(\mathbf{x}_{k+1}) + \mathbf{u}_{k+1}) - \nabla g(\mathbf{x}_k)(\boldsymbol{\lambda}_k + \rho(g(\mathbf{x}_k) + \mathbf{u}_k))\| \\
\leq & (L_f + 1/\eta)\|\mathbf{x}_{k+1} - \mathbf{x}_k\| \\
& + \|\nabla g(\mathbf{x}_{k+1})\boldsymbol{\lambda}_{k+1} - \nabla g(\mathbf{x}_k)\boldsymbol{\lambda}_{k+1} + \nabla g(\mathbf{x}_k)\boldsymbol{\lambda}_{k+1} - \nabla g(\mathbf{x}_k)\boldsymbol{\lambda}_k\| & \text{(66a)} \\
& + \rho\|\nabla g(\mathbf{x}_{k+1})g(\mathbf{x}_{k+1}) - \nabla g(\mathbf{x}_k)g(\mathbf{x}_{k+1}) + \nabla g(\mathbf{x}_k)g(\mathbf{x}_{k+1}) - \nabla g(\mathbf{x}_k)g(\mathbf{x}_k)\| & \text{(66b)} \\
& + \rho\|\nabla g(\mathbf{x}_{k+1})\mathbf{u}_{k+1} - \nabla g(\mathbf{x}_k)\mathbf{u}_{k+1} + \nabla g(\mathbf{x}_k)\mathbf{u}_{k+1} - \nabla g(\mathbf{x}_k)\mathbf{u}_k\|, & \text{(66c)}
\end{aligned}
$$

in which equation 66a, equation 66b, and equation 66c can be bounded by

$$
\begin{aligned}
\text{equation } 66a \leq & B_{\boldsymbol{\lambda}}L_g\|\mathbf{x}_{k+1} - \mathbf{x}_k\| + M_g\|\boldsymbol{\lambda}_{k+1} - \boldsymbol{\lambda}_k\| \\
\leq & B_{\boldsymbol{\lambda}}L_g\|\mathbf{x}_{k+1} - \mathbf{x}_k\| + \rho M_g^2\|\mathbf{x}_{k+1} - \mathbf{x}_k\| \\
& + \rho M_g\|\mathbf{u}_{k+1} - \mathbf{u}_k\| + M_g\|\boldsymbol{\mu}_{k+1} - \boldsymbol{\mu}_k\| \\
\leq & \left(B_{\boldsymbol{\lambda}}L_g + \rho M_g^2\right)\|\mathbf{x}_{k+1} - \mathbf{x}_k\| \\
& + \rho M_g\|\mathbf{u}_{k+1} - \mathbf{u}_k\| + M_g\delta_k; \\
\text{equation } 66b \leq & (\rho B_g L_g + \rho M_g^2)\|\mathbf{x}_{k+1} - \mathbf{x}_k\|; \\
\text{equation } 66c \leq & \rho B_{\mathbf{u}}L_g\|\mathbf{x}_{k+1} - \mathbf{x}_k\| + \rho M_g\|\mathbf{u}_{k+1} - \mathbf{u}_k\|.
\end{aligned}
$$

where for bounding equation 66a, we used the $\boldsymbol{\lambda}$-update and $\|\boldsymbol{\mu}_{k+1} - \boldsymbol{\mu}_k\| = \frac{\delta_k}{\|\boldsymbol{\lambda}_k - \boldsymbol{\mu}_k\| + \frac{1}{\|\boldsymbol{\lambda}_k - \boldsymbol{\mu}_k\|}} \leq \delta_k$. Hence,

$$
\begin{aligned}
\|\zeta_{\mathbf{x}}^{k+1}\| \leq & (L_f + 1/\eta + B_{\boldsymbol{\lambda}}L_g + \rho L_g(B_g + B_{\mathbf{u}} + 2M_g^2))\|\mathbf{x}_{k+1} - \mathbf{x}_k\| \\
& + 2\rho M_g\|\mathbf{u}_{k+1} - \mathbf{u}_k\| + M_g\delta_k.
\end{aligned}
$$

Next, we estimate an upper bound for the component $\zeta_{\mathbf{u},k+1}$. From the proof of Lemma 35, we have

$$\left(\|\nabla_{\mathbf{u}}\mathcal{L}_\rho(\mathbf{u}_k) - \nabla_{\mathbf{u}}\mathcal{L}_\rho(\mathbf{u}_{k+1})\| + \tau^{-1}\|\mathbf{u}_{k+1} - \mathbf{u}_k\|\right) \cdot \|\widetilde{\mathbf{u}}_{k+1} - \mathbf{u}_{k+1}\| \geq \|\widetilde{\mathbf{u}}_{k+1} - \mathbf{u}_{k+1}\|^2,$$

where

$$
\begin{aligned}
\|\nabla_{\mathbf{u}}\mathcal{L}_\rho(\mathbf{u}_k) - \nabla_{\mathbf{u}}\mathcal{L}_\rho(\mathbf{u}_{k+1})\| = & \|\nabla_{\mathbf{u}}\mathcal{L}_\rho(\mathbf{x}_{k+1}, \mathbf{u}_k, \mathbf{z}_k, \boldsymbol{\lambda}_k, \boldsymbol{\mu}_k) \\
& - \nabla_{\mathbf{u}}\mathcal{L}_\rho(\mathbf{x}_{k+1}, \mathbf{u}_{k+1}, \mathbf{z}_{k+1}, \boldsymbol{\lambda}_{k+1}, \boldsymbol{\mu}_{k+1})\| \\
\leq & \|\boldsymbol{\lambda}_k + \rho(g(\mathbf{x}_{k+1}) + \mathbf{u}_k) - \boldsymbol{\lambda}_{k+1} - \rho(g(\mathbf{x}_{k+1}) + \mathbf{u}_{k+1})\| \\
\leq & \rho(M_g\|\mathbf{x}_{k+1} - \mathbf{x}_k\| + 2\|\mathbf{u}_{k+1} - \mathbf{u}_k\| + \delta_k).
\end{aligned}
$$

Therefore,

$$\|\zeta_{\mathbf{u}}^{k+1}\| = \|\widetilde{\mathbf{u}}_{k+1} - \mathbf{u}_{k+1}\| \leq \rho M_g\|\mathbf{x}_{k+1} - \mathbf{x}_k\| + \left(2\rho + \tau^{-1}\right)\|\mathbf{u}_{k+1} - \mathbf{u}_k\| + \delta_k. \tag{67}$$

Combining equation 67 and equation 67, we obtain

$$\|\boldsymbol{\zeta}_{\mathbf{p}}^{k+1}\| \leq d_1(\|\mathbf{x}_{k+1} - \mathbf{x}_k\| + \|\mathbf{u}_{k+1} - \mathbf{u}_k\|) + (M_g + 1)\delta_k,$$

where $d_1 = \max\{L_f + B_{\boldsymbol{\lambda}}L_g + \rho(M_g + B_gL_g + B_{\mathbf{u}}L_g + 2M_g^2) + 1/\eta, 2\rho(M_g + 1) + 1/\tau\}$. This inequality, combined with $\boldsymbol{\zeta}_{\mathbf{p}}^{k+1} \in \partial_{\mathbf{p}}\mathcal{L}_\rho(\mathbf{w}_{k+1})$, yields the desired result. $\qquad\square$

It can be easily verified that if $\frac{1}{T}\sum_{k=0}^{T-1}\|\boldsymbol{\zeta}_{\mathbf{p}}^{k+1}\| \to 0$, then a point that satisfies stationarity in the KKT conditions equation 2,

$$\mathbf{0} \in \nabla f(\mathbf{x}^*) + \partial r(\mathbf{x}^*) + \nabla g(\mathbf{x}^*)\boldsymbol{\lambda}^*,$$

is obtained. Specifically,

$$\begin{cases} \mathbf{0} \in \nabla f(\overline{\mathbf{x}}) + \partial r(\overline{\mathbf{x}}) + \nabla g(\overline{\mathbf{x}})\overline{\boldsymbol{\lambda}}, \\ \mathbf{0} = \overline{\mathbf{u}} - \Pi_{\mathbb{R}_+^m}[\overline{\mathbf{u}} - (\overline{\boldsymbol{\lambda}} + \rho(g(\overline{\mathbf{x}}) + \overline{\mathbf{u}}))], \end{cases}$$
$$\iff \quad \mathbf{0} \in \nabla f(\overline{\mathbf{x}}) + \partial r(\overline{\mathbf{x}}) + \nabla g(\overline{\mathbf{x}})\overline{\boldsymbol{\lambda}}.$$

We will use this part to establish convergence to a KKT point in Theorem 28. Note that we need not consider the gradient of $\mathcal{L}_\rho$ with respect to $\boldsymbol{\lambda}$, i.e., $\xi_{\boldsymbol{\lambda}}^{k+1} := \nabla_{\boldsymbol{\lambda}}\mathcal{L}_\rho(\mathbf{w}_{k+1})$, since we know from the $\boldsymbol{\lambda}$-update step equation 13 that $\nabla_{\boldsymbol{\lambda}}\mathcal{L}_\rho(\mathbf{w}_{k+1}) = g(\mathbf{x}_{k+1}) + \mathbf{u}_{k+1} - \mathbf{z}_{k+1} - \beta(\boldsymbol{\lambda}_{k+1} - \boldsymbol{\mu}_{k+1}) = \mathbf{0}$.

# E  Proofs of Asymptotic Convergence for Algorithm 2

## E.1  Proof of Primal Stationarity (Lemma 22)

*Proof.* Analogous to the proof of Lemma 12 in Section B.1, by using Lemma 38 and 39 with $\widehat{\delta}_k = \frac{\delta_k^2}{4\rho} + \frac{\delta_k}{\rho}$, we have

$$\frac{1}{T}\sum_{k=0}^{T-1}\|\boldsymbol{\zeta}_{\mathbf{p}}^{k+1}\|^2 \le \frac{\frac{3d_1^2}{c_3}\left(\mathcal{L}_\rho(\mathbf{w}_0) - \underline{\mathcal{L}_\rho}\right)}{T} + \frac{\left(\frac{3d_1^2}{4\rho c_3} + 3(M_g + 1)^2\right)\sum_{k=0}^{T-1}\delta_k^2}{T} + \frac{\frac{1}{\rho}\sum_{k=0}^{T-1}\delta_k}{T}. \tag{68}$$

Given $\delta_k = \frac{1}{p \cdot k^q + 1}$ with $2/3 < q \le 1$ and $p > 0$, the third term on the RHS of the above inequality dominates the second term. Moreover, for sufficiently large $T$, one can easily show that

$$\sum_{k=0}^{T-1}\delta_k \approx \begin{cases} p^{-1}\log(pT) & \text{if } q = 1, \\ (p - qp)^{-1}T^{1-q} & \text{if } \frac{2}{3} < q < 1. \end{cases}$$

Thus, for $q = 1$, the sum grows logarithmically, while for $2/3 < q < 1$, the sum grows polynomially with $T$. Therefore, for each choice of $q$, the RHS of equation 68 goes to 0 as $T$ increases, which proves that the primal sequences are convergent. $\qquad\square$

## E.2  Proof of Primal Feasibility (Lemma 25)

*Proof.* Follow the same proof as that of Lemma 14 in Section C.2. $\qquad\square$

## E.3  Proof of Dual Feasibility (Lemma 26)

*Proof.* The proof is analogous to that of Lemma 15 in Section B.3. By the update rule equation 20, $\mathbf{u} \in \mathbb{R}_+^m$ and $\overline{\mathbf{u}} \ge 0$. Thus, there are two cases for $\overline{\mathbf{u}} > 0$ and $\overline{\mathbf{u}} = 0$. If $\overline{\mathbf{u}} > 0$, by Lemma 25 and the stationarity of $\overline{\mathbf{u}}$ and $\overline{\overline{\boldsymbol{\lambda}}}$, $\overline{\boldsymbol{\lambda}} + \rho(g(\overline{\mathbf{x}}) + \overline{\mathbf{u}}) = \overline{\boldsymbol{\lambda}} = 0$. Similarly, if $\overline{\mathbf{u}} = 0$, $\overline{\boldsymbol{\lambda}} + \rho(g(\overline{\mathbf{x}}) + \overline{\mathbf{u}}) = \overline{\boldsymbol{\lambda}} \ge 0$. In any case, we conclude that $\overline{\boldsymbol{\lambda}} \ge 0$. $\qquad\square$

## E.4  Proof of Complementary Slackness (Lemma 27)

*Proof.* It follows the same proof as that of Lemma 16 in Section B.4. $\qquad\square$

# F   Proofs of Non-asymptotic Convergence Rate of Algorithm 2

## F.1   Proof of Primal Stationarity Convergence Rate (Lemma 29)

*Proof.* The proof is analogous to that of Lemma 18 using Lemma 22 and equation 26. By triangle inequality, Jensen's inequality and the result from Lemma 25,

$$\bar{\boldsymbol{\nu}}_T - \bar{\boldsymbol{\lambda}}_T = \frac{1}{T} \sum_{k=0}^{T-1} \left( \frac{1}{\tau} - \rho \right) (\mathbf{u}_{k+1} - \mathbf{u}_k) + \boldsymbol{\lambda}_{k+1} - \boldsymbol{\mu}_{k+1} \tag{69}$$

$$\leq \frac{1-\tau\rho}{\tau T}(\mathbf{u}_T - \mathbf{u}_0) + \frac{1}{T} \sum_{k=0}^{T-1} \|\boldsymbol{\lambda}_{k+1} - \boldsymbol{\mu}_{k+1}\| \tag{70}$$

$$= \mathcal{O}(1/T) + \mathcal{O}(1/\sqrt{T}) = \mathcal{O}(1/\sqrt{T}). \tag{71}$$

Therefore, the stationarity condition holds for $\bar{\boldsymbol{\nu}}_T$ with the same convergence rate. $\square$

## F.2   Proof of Primal Feasibility Convergence Rate (Lemma 30)

*Proof.* The proof is analogous to that of Lemma 19 using Lemma 25. $\square$

## F.3   Proof of Complementary Slackness Convergence Rate (Lemma 31)

*Proof.* From the optimality of $\mathbf{u}$-update equation 20 and the construction of $\boldsymbol{\nu}$ equation 26, we have $\mathbf{u}_{k+1} = \Pi_{\mathbb{R}_+^m}[\mathbf{u}_{k+1} - \tau\boldsymbol{\nu}_k]$, which implies the complementarity condition $\langle \boldsymbol{\nu}_k, \mathbf{u} - \mathbf{u}_{k+1} \rangle \geq 0$ for all $\mathbf{u} \geq \mathbf{0}$. The rest of the proof is the same as that of Lemma 20. $\square$

# G   Proof of Corollary 23

*Proof.* By using Jensen's inequality, $\left( \frac{1}{T} \sum_{k=0}^{T-1} \|\boldsymbol{\zeta}_{\mathbf{p}}^{k+1}\| \right)^2 \leq \frac{1}{T} \sum_{k=0}^{T-1} \|\boldsymbol{\zeta}_{\mathbf{p}}^{k+1}\|^2$, and taking the square root, we obtain

$$\frac{1}{T} \sum_{k=0}^{T-1} \|\boldsymbol{\zeta}_{\mathbf{p}}^{k+1}\| \leq \frac{1}{\sqrt{T}} \sqrt{\sum_{k=0}^{T-1} \|\boldsymbol{\zeta}_{\mathbf{p}}^{k+1}\|^2}.$$

Denoting the RHS of inequality equation 68 by $\Delta_T$ and combining Lemma 22 with the above inequality give

$$\frac{1}{T} \sum_{k=0}^{T-1} \|\boldsymbol{\zeta}_{\mathbf{p}}^{k+1}\| \leq \frac{\sqrt{\Delta_T}}{\sqrt{T}} \leq \epsilon,$$

which, along with the result in equation 24, gives $\widetilde{\mathcal{O}}\left( \frac{1}{\sqrt{T}} \right)$. Therefore, the following number of iterations is required to have $\epsilon$-primal stationarity:

$$T := \left\lceil \frac{\Delta_T}{\epsilon^2} \right\rceil = \widetilde{\mathcal{O}}\left( \frac{1}{\epsilon^2} \right).$$

$\square$

# H   Experimental Details

## H.1   Classification Problems Under Non-smooth Fairness Constraints

**Experimental Setup**

We benchmark PLADA against four state-of-the-art algorithms: the single-loop switching subgradient (SSG) algorithm Huang & Lin (2023), two double-loop inexact proximal point (IPP) algorithms (IPP-ConEx Boob

et al. (2022) and IPP-SSG Huang & Lin (2023)) and the multiplier model approach Narasimhan et al. (2020). For the benchmark algorithms, we followed the hyperparameter settings of Huang & Lin (2023) and Narasimhan et al. (2020) and we provide detailed descriptions of hyperparameters in Table 1. Note that we only used two hyper-parameter sets for 8 different experiments, while our benchmark algorithms used different hyper-parameters for every datasets, objectives and constraints.

Table 1: Hyper-parameters of PLADA used in experiments

| Problem | $\eta_w$ | $\eta_u$ | $\alpha$ | $\beta$ | $\delta_0$ |
|---|---|---|---|---|---|
| Models 5.1.1 and 5.1.2 | 0.001 | 0.1 | 10.0 | 0.1 | 0.1 |
| Neural network 5.1.3 | 0.1 | 0.01 | 10.0 | 0.5 | 0.1 |

**Datasets**

Our evaluation uses several standard real-world datasets: **Adult** Kohavi et al. (1996), **Bank** Moro et al. (2014), **COMPAS** Angwin et al. (2022) and **Communities and Crime** Redmond (2009). The descriptions of the datasets are presented in Table 2.

Table 2: Real-world fairness datasets used in experiments

| Dataset | n | d | Label | Sensitive Group |
|---|---|---|---|---|
| Adult (a9a) | 48,842 | 123 | Income | Gender |
| Bank | 41,188 | 54 | Subscription | Age |
| COMPAS | 6,172 | 16 | Recidivism | Race |
| Communities and Crime | 1,994 | 140 | Crime | Race |
| MSLR-WEB10K | 1.2 M | 136 | Relevance | Quality Score |

### H.2 Non-convex Multi-class Neyman-Pearson Classification

**Implementation details**

We employ a two-layer feed-forward neural network with sigmoid activation for the classition faction on the Fashion-MNIST Xiao et al. (2017) and CIFAR10 Krizhevsky et al. (2009) datasets. We compare Algorithm 2 (PPALA) with GDPA, as NL-IAPIAL can only handle convex constraints. A sigmoid function $\phi(y) = 1/(1 + \exp(y))$ is used for the loss function as in Lu (2022). For the experiments, we use 4 classes and set $\theta = 1$ with $\kappa_i = 1$ for Fashion-MNIST and $\kappa_i = 2$ for CIFAR-10.[4] PPALA uses a fixed learning rate $10^{-3}$ and parameters $\alpha = 10, \beta = 0.2$ for both datasets. The initial point $\mathbf{x}_0$ is randomly generated in each experiment. These numerical experiments were conducted using an A100 GPU and were implemented with Pytorch Paszke et al. (2019).

## I Additional Experiments

In this section, we further validate the claims on hyperparameter sensitivity and dual variable convergence by conducting additional experiments. We also show the empirical performance of Algorithm 1 by extending the application to stochastic setting.

**Hyperparameter robustness**

Although PLADA requires the selection of multiple hyperparameters $(\alpha, \beta, \rho, \eta, \tau)$, we show that it is straight-forward to select appropriate values for each hyperparameter. Figures 5 and 6 show the performance of PLADA across a wide range of values for the key parameters $\alpha > 0$ and $\beta > 0$, respectively. The results demonstrate that the algorithm's convergence behavior is remarkably stable and can still provide solutions that minimize the objective while remaining feasible.

---

[4]Note that the parameter settings for GDPA, as in (Lu, 2022, Section F in Appendix), lead to a lack of convergence in the neural network setting, particularly with a small value of the threshold $\kappa_i$ and a large increase ratio for updating the penalty parameter.

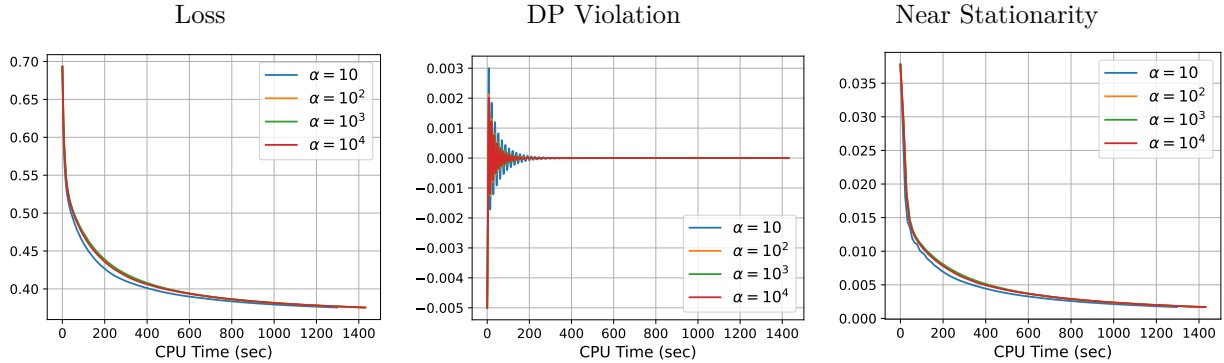

Figure 5: Comparison of the performance of PLADA with different $\alpha$ on the logistic loss objective with demographic parity (DP) constraint on Adult dataset. The results show the performance of PLADA is not sensitive to the value of $\alpha$ ($\beta = 0.1$ is fixed).

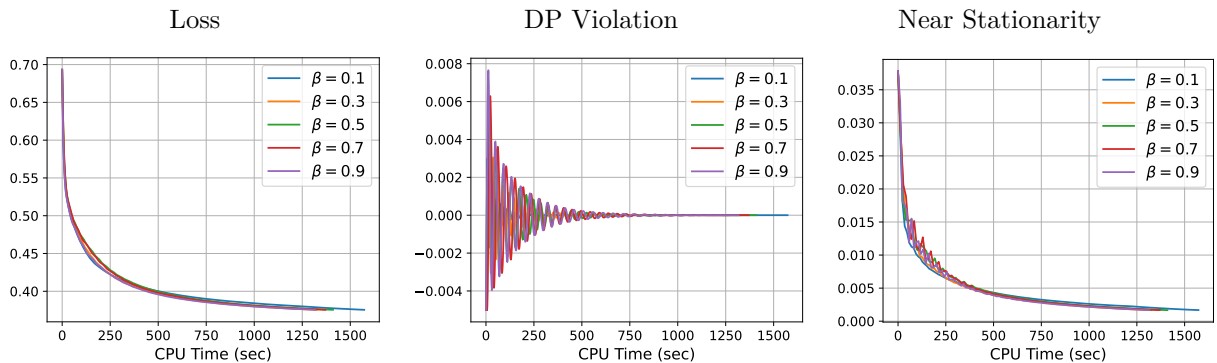

Figure 6: Comparison of the performance of PLADA with different $\beta$ on the logistic loss objective with demographic parity (DP) constraint on Adult dataset. The results show that the performance of PLADA is very slightly sensitive to the choice of $\beta$, as it affect dual parameter defined by $\rho = \frac{\alpha}{1+\alpha\beta}$ ($\alpha = 10$ is fixed).

**Dual variables convergence**

Lemma 25 claims that the gap between the dual variables $\|\boldsymbol{\lambda} - \boldsymbol{\mu}\|$ should converge to zero, ensuring feasibility. Figure 7 provides empirical validation of this result. The plots clearly show the convergence of $|\boldsymbol{\lambda} - \boldsymbol{\mu}|$ to zero, as well as the individual convergence of $\boldsymbol{\lambda}$ and $\boldsymbol{\mu}$.

**Highly stochastic setting**

To evaluate PLADA in a more challenging setting, we applied it to a ranking fairness problem using the large-scale MSLR-WEB10K dataset, which involves over 470,000 pairwise constraints. In this highly stochastic environment, where mini-batching is inevitable, PLADA was benchmarked against the method of Narasimhan et al. (2020). As shown in Figure 8, our algorithm achieves lower error rate and better constraint satisfaction, demonstrating its effectiveness even under highly stochastic conditions.

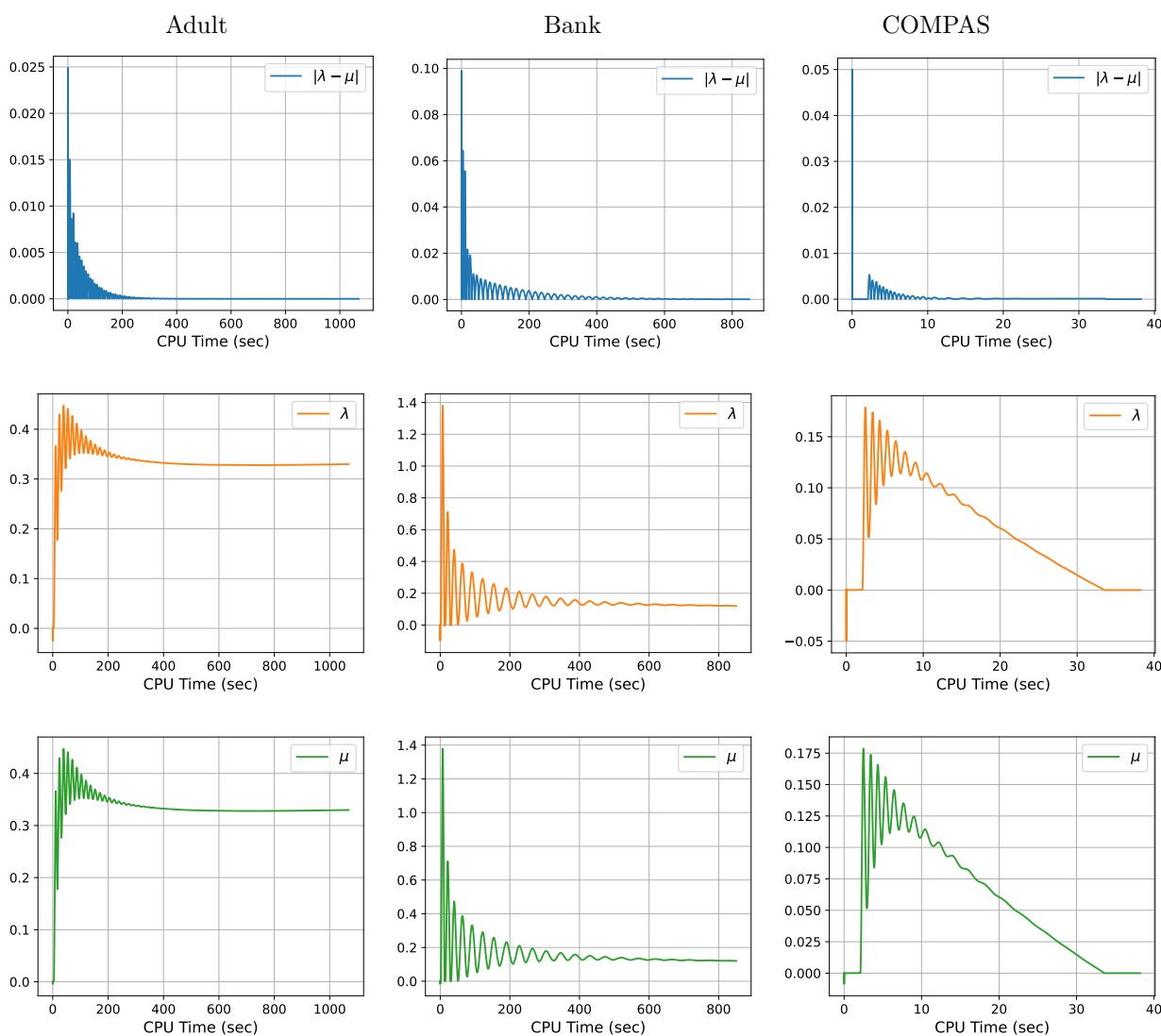

Figure 7: The values of $|\lambda - \mu|, \lambda, \mu$ of PLADA on the logistic loss objective with demographic parity (DP) constraint. The results show the converging behavior of the dual variables and their difference.

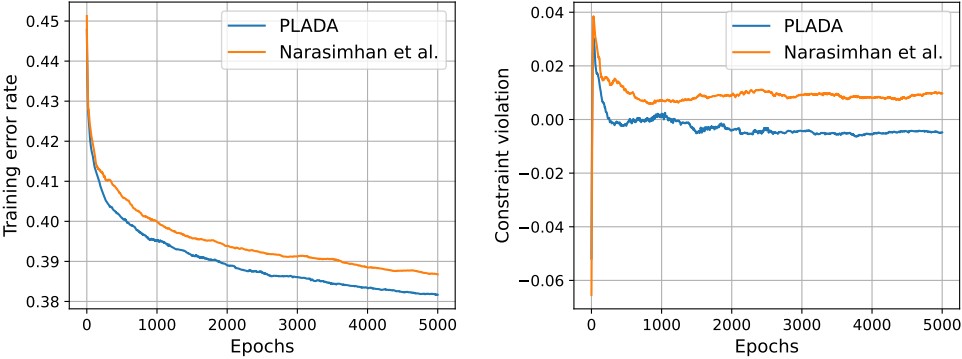

Figure 8: The average performance of PLADA and Narasimhan et al. (2020) on the ranking fairness versus Epochs after three repetitions. MSLR-WEB10K dataset has over 1.2M data points, from which over 470k pairs are created. PLADA achieves better constraint satisfaction with comparable error rate against approximate methods for the stochastic setting.

