# OpenReview forum: "Beyond Convexity: Proximal-Perturbed Lagrangian Methods for Efficient Functional Constrained Optimization"
_TMLR — Rejected by TMLR_

### Review · Reviewer_necn · 2025-11-13

**Summary Of Contributions:**

This paper develops a single-loop primal–dual framework based on a novel Proximal–Perturbed Lagrangian for nonconvex functional constrained optimization. The authors propose two algorithms (PLADA for non-smooth constraints and PPALA for differentiable constraints) that use fixed penalty/proximal parameters and diminishing auxiliary-step sizes. They provide asymptotic convergence to KKT points and non-asymptotic ergodic complexity guarantees (claimed to be $\mathcal{O}(\epsilon^{-2})$ under suitable choices). Numerical experiments on fairness-constrained classification and multi-class Neyman-Pearson tasks support the practical effectiveness of the methods.

**Additional Comments:**

Please see above.

**Audience:**

Yes

**Audience Explanation:**

This paper addresses nonconvex optimization problems with functional constraints, a class with wide applications in machine learning, data science, and signal processing. The authors propose new algorithms that achieve an improved iteration complexity bound and feature computationally efficient update rules.

**Claims And Evidence:**

Yes

**Claims Explanation:**

The paper is well-structured and logically organized, making it easy to follow.

**Requested Changes:**

1. Technical and Typographical Issues
 * In  Algorithm 1, the step size in Step 5 should be $\sigma_k / \rho$ according to equation (10).
* In Equation (10), the variable $\mathbf{z}_k$ should be bold to remain consistent with the vector notation.
* In the proof of Lemma 33, the first equation should use $r(x_k)$ instead of $r(x)$; a similar issue occurs in the inequality following equation (36).
* In Equation (38), the coefficient in front of $||\lambda_k - \mu_k||^2$ should be $\frac{1}{\rho}\left(1 - (1 - \sigma / \rho)^2\right)$; the denominator $\rho$ is missing.
* In the proof of Lemma 17, the line after the first inequality should be $x_{k+1}$.

2. Conceptual Questions and Clarifications}
* In the proof of Lemma 13, the boundedness of $\mu_k$ is claimed to follow from Assumption~7, but this assumption only ensures the boundedness of $\lambda_k$. Clarifying this step or providing an additional argument would improve the rigor of the proof.
 * Equation (48) may not directly imply the convergence of $|\lambda_t - \mu_t|$. Since the step size $\sigma_t$ is proportional to $\delta_t / (|\lambda_t - \mu_t|^2 + 1)$, the bound in Eq (48) effectively ensures only that $\sum_{t=0}^{\infty} \frac{\lambda_t - \mu_t}{t |\lambda_t - \mu_t|^2} < \infty$, which does not necessarily imply that $|\lambda_t - \mu_t| \to 0$. For instance, in oscillatory cases such as $\lambda_t - \mu_t = (-1)^t$ (or even slowly growing sequences like $\lambda_t - \mu_t = \sqrt{t}$), the above series can still converge while the norm does not vanish. Moreover, it appears that we only have a subsequence convergence rather than convergence of the entire sequences ${x_t}$ and ${u_t}$ since Lemma 33 is not a ``pure'' decreasing property and $\delta_t=O(1/t)$. Therefore, I think Eq.~(48) alone is insufficient to guarantee $|\lambda_t - \mu_t|\to 0$. Please correct me if I have misunderstood the derivation.
* Please elaborate on how $\frac{1}{T}\sum_{k=0}^{T-1} |\lambda_k - \mu_k|^2 = O(1/T)$ is derived, as this appears to be a crucial step in Lemmas~18 and~19.
 * Assumption~6 seems restrictive, as it excludes simple cases such as $r(x) = 0$. While this assumption simplifies the boundedness arguments, it may limit generality.
* It would be helpful to elaborate on how the proposed framework compares with [1].
[1] Kim, Jong Gwang. "Equilibrium computation of generalized Nash games: A new Lagrangian-based approach." arXiv preprint arXiv:2106.00109 (2021).

---

> ### Author Response · Authors · 2026-01-23
> **Official Comment by Authors (1/3)**
>
> We thank the reviewer for their detailed and insightful feedback. We have carefully considered the points raised by the reviewer, revised the manuscript, and provide our responses below.
>
> ### Technical and Typographical Issues
>
> We sincerely thank the reviewer for finding the typographical issues. The reviewer is correct regarding the missing denominator $\rho$ and we have modified the occurrences in equation (10) as well as in the proof of Lemma 33. We also rectified all other typos found by the reviewer. Fortunately, they were either correctly considered in our algorithm and analysis or only affected the analysis by a constant factor.
>
> ### Conceptual Questions and Clarifications
>
> 1. We agree that the boundedness of the sequence $\{\mu_k\}$ requires a more explicit justification, as Assumption 7 only defines the boundedness of the multiplier sequence $\{\lambda_k\}$. We will clarify this in the proof of Lemma 13 using the convexity of the update rule.
>
>     The $\mu$-update, $\mu_{k+1} = \mu_k + \frac{\sigma_k}{\rho}(\lambda_k - \mu_k)$, can be rewritten as a convex combination: $\mu_{k+1} = \left(1 - \frac{\sigma_k}{\rho}\right)\mu_k + \left(\frac{\sigma_k}{\rho}\right)\lambda_k.$
>
>     With the initialization of $\mu_0 = \lambda_0  \in \Lambda$ in the proposed algorithm, we have
>     $$
>     \mu_{1} = \left(1- \frac{\sigma_0}{\rho}\right)\mu_0 + \left(\frac{\sigma_0}{\rho}\right) \lambda_0.
>     $$
>     Since $\mu_0 \in \Lambda$ and $\lambda_0 \in \Lambda$ with $\sigma_k / \rho \in (0,1)$ for all $k$, $\mu_1$ remains in the bounded set $\Lambda \subset \mathbb{R}^m$ (Assumption 7). By induction, the dual sequence $\{\mu_k\}$ is also bounded.
>
> 2. We thank the reviewer for the precise observation regarding the convergence behavior of $\{\lambda_k - \mu_k\}$. We agree that our derivation primarily focuses on the convergence of a subsequence $\{\lambda_{k_j} - \mu_{k_j}\} \rightarrow 0$ instead of the entire sequence. Nonetheless, we would like to highlight that subsequence convergence to a stationary point is a standard guarantee in non-convex optimization [1,2,3]. We will clarify this distinction in the revised proof in terms of a convergent subsequence.
>
>     Since $g(x_k) + u_k$ is bounded, the $\lambda$-update implies that the sequence $\{ \lambda_k - \mu_k \}$ has a limit point, denoted by $\overline{\lambda}- \overline{\mu}$. Then, $\{ \lambda_{k_j} - \mu_{k_j} \}$ for $j \in \mathbb{N}$ has a bounded subsequence converging to $\overline{\lambda}- \overline{\mu}$. Without loss of generality, by restricting to such a subsequence $\{ \lambda_{k_j} - \mu_{k_j} \}$ for $j \in \mathbb{N}$ that converges to $\overline{\lambda}- \overline{\mu}$, we may assume that $\lim_{j \to \infty}  \| \lambda_{k_j} - \mu_{k_j}\|= \|e\|,$ where $e=\overline{\lambda} - \overline{\mu}$. Thus, focusing on such a subsequence allows us to exclude the oscillatory cases such as $\lambda_k -\mu_k=(-1)^k$ (or slowly growing sequences like $\|\lambda_k - \mu_k \|=\sqrt{k}$). Therefore, we can prove that $\{ \lambda_k - \mu_k\} \rightarrow 0$  by contradiction, by assuming that there exists a subsequence such that $\{ \lambda_k - \mu_k\}  \rightarrow e$ as $k\rightarrow \infty$ for some $e \neq 0$. The remaining proof follows from here.
>
>     [1] Kaizhao Sun and Xu Andy Sun. Dual descent augmented Lagrangian method and alternating direction method of multipliers. SIAM Journal on Optimization, 34(2):1679–1707, 2024.
>
>     [2] Digvijay Boob, Qi Deng, and Guanghui Lan. Stochastic first-order methods for convex and nonconvex functional constrained optimization. Mathematical Programming, 197(1):215-279, 2022.
>
>     [3] Jiawei Zhang and Zhi-Quan Luo. A proximal alternating direction method of multiplier for linearly constrained nonconvex minimization. SIAM Journal on Optimization, 30(3):2272-2302. 2020.

---

> ### Author Response · Authors · 2026-01-23
> **Official Comment by Authors (2/3)**
>
> 3. We thank the reviewer for pointing out this issue. To explicitly obtain $\frac{1}{T}\sum \|\lambda_k-\mu_k\|^2 = \tilde{\mathcal{O}}(1/T)$ in Lemmas 18 and 19, we need to invoke a local error bound condition. This assumption relates the distance to the solution set to the stationarity residual. It holds for broad classes of structured non-convex functions under mild assumptions and is commonly used in convergence analysis in non-convex non-smooth optimization [1,2]. Specifically, there exists a constant $\kappa>0$ and a neighborhood $U$ of the stationary set $X^\ast$ such that $\text{dist}(x,X^\ast)\le\kappa\cdot\text{dist}(0,\partial f(x))$ for all $x\in U$.
>
>     From the $\lambda$-update, we know that
>     $$\|\lambda_{k+1}-\mu_{k+1}\| = \rho\|g(x_{k+1}) + u_{k+1}\|.$$
>     Let $(x^\ast, u^\ast) \in X^\ast \times U^\ast $be a feasible point pair, where $X^\ast$ and $U^\ast$ denotes the optimal sets,  then $g(x^\ast)+u^\ast=0$. By Assumption 5 and triangle inequality,
>     $$
>     \begin{aligned}
>         \|g(x_{k+1})+u_{k+1}\| &= \|g(x_{k+1})+u_{k+1} - (g(x^\ast)+u^\ast)\| \\\\
>         &\le L_g\|x_{k+1}-x^\ast\| + \|u_{k+1}-u^\ast\|.
>     \end{aligned}
>     $$
>     By the local error assumption, for sufficiently large $k$, we have
>     $$\text{dist}((x_{k+1},u_{k+1}),X^\ast \times U^\ast) \le \kappa \cdot \|\zeta_p^{k+1}\|.$$
>     Using Lemmas 34 and 38 and applying the inequality $(a+b)^2\le2a^2+2b^2$, we have
>     $$\|\lambda_{k+1}-\mu_{k+1}\|^2 \le C\left(4d_1^2(\|x_{k+1}-x_k\|^2+\|u_{k+1}-u_k\|^2) + 2(M_g+1)^2\delta_k^2\right),$$
>     for some constant $C$. Summing the inequality over $k=0,\dots,T-1$,
>     $$\sum_{k=0}^{T-1}\|\lambda_{k+1}-\mu_{k+1}\|^2 \le 4d_1^2C\sum_{k=0}^{T-1}(\|x_{k+1}-x_k\|^2+\|u_{k+1}-u_k\|^2) + 2(M_g+1)^2C\sum_{k=0}^{T-1}\delta_k^2.$$
>     Since Lemmas 33 and 37 give
>     $$\sum_{k=0}^{T-1} \|x_{k+1}-x_k\|^2+\|u_{k+1}-u_k\|^2 \le \mathcal{L}(w_0) - \mathcal{L}(w_T) + \sum_{k=0}^{T-1} \hat{\delta}_k,$$
>     the desired rate follows.
>
>     We emphasize that while this local error bound assumption yields the fast $\mathcal{O}(1/T)$ rate, our method's asymptotic convergence (Theorem 16) holds without this assumption, relying solely on the properties of the proximal-perturbed update. This distinguishes our framework from prior works that may require strong regularity conditions just to ensure feasibility. We will add this assumption to the corresponding Lemmas and Theorems.
>
>     [1] Dmitriy Drusvyatskiy and Adrian S. Lewis. Error Bounds, Quadratic Growth, and Linear Convergence of Proximal Methods. Mathematics of Operations Research, 43(3):919-948, 2018.
>
>     [2] Damek Davis, Dmitriy Drusvyatskiy, Kellie J. MacPhee and Courtney Paquette. Subgradient Methods for Sharp Weakly Convex Functions. Journal of Optimization Theory and Applications, 179(3):962-982, 2018.

---

> > ### Author Response · Authors · 2026-01-23
> > **Official Comment by Authors (3/3)**
> >
> > 4. We appreciate the reviewer for identifying the restrictiveness of Assumption 6. We employ this assumption to ensure the problem is well-defined and the primal iterates remain bounded. Similar assumptions are standard in the design and analysis of algorithms for constrained non-convex optimization problems [1,2,3].
> >
> >     In our framework, we utilize regularizers $r(x)$, such as $\ell_1$ and $\ell_2$ norms or box constraints, to impose structural properties on the solution. From a theoretical perspective, Assumption 6 formalizes the effective boundedness induced by these norms, essentially restricting the analysis to a compact set.
> >
> >     Relaxation of Assumption 6 to allow the domain of $x$ to be $\mathbb{R}^n$ would require alternative assumptions. One possibility is to assume explicit boundedness of the sequence  $\{x_k\}$ generated by the algorithm, which is a much stronger assumption. Alternatively, one may impose a coercivity assumption on the objective function $f$ to guarantee the boundedness of $\{ x_k \}$. However, coercivity is often restrictive in practical machine learning applications.
> >
> >     Instead, we can adopt a compact set $X$ as the domain of $x$ to ensure boundedness, which is both practical and standard. Then, the regularizer can be written in more general form: $r(x) = \mathbb{I}_X(x) + \bar{r}(x),$ where $\mathbb{I}_X(x)$ is the indicator function of the set $X$ and $\bar{r}(x)$ denotes the remaining part of the regularizer [4,5].
> >
> >     [1] Weiwei Kong, Jefferson G Melo, and Renato DC Monteiro. Iteration complexity of a proximal augmented lagrangian method for solving nonconvex composite optimization problems with nonlinear convex constraints. Mathematics of Operations Research, 48(2):1066-1094, 2022.
> >
> >     [2] Zhaosong Lu and Zirui Zhou. Iteration-complexity of first-order augmented lagrangian methods for convex conic programming. SIAM Journal on Optimization, 33(2):1159–1190, 2023.
> >
> >     [3] Digvijay Boob, Qi Deng, and Guanghui Lan. Level constrained first order methods for function constrained optimization. Mathematical Programming 209(1): 1-61, 2025.
> >
> >     [4] Zichong Li, Pin-Yu Chen, Sijia Liu, Songtao Lu, and Yangyang Xu. Stochastic inexact augmented Lagrangian method for nonconvex expectation constrained optimization. Computational Optimization and Applications 87(1): 117-147, 2024.
> >
> >     [5] Kaizhao Sun and Xu Andy Sun. Dual descent augmented Lagrangian method and alternating direction method of multipliers. SIAM Journal on Optimization, 34(2):1679–1707, 2024.
> >
> > 5. We thank the reviewer for pointing out the connection to Kim (2021). We will add a subsection detailing this comparison. The Proximal-Perturbed Lagrangian algorithm proposed by Kim (2021) is designed for generalized Nash equilibrium problems (GNEPs), where $n$ optimization problems of $n$ players are solved simultaneously. In contrast, our algorithm in this work solves a single non-convex optimization problem based on the Proximal-Perturbed Lagrangian.
> >
> >     The algorithm in Kim (2021) heavily relies on a quadratic approximation for all players within an inner loop. As described in their Algorithm 1, Step 1 involves inner iterations $l = 0,1,2,\dots$, where a fixed-point subproblem is solved using gradient projection until specific error conditions are satisfied. This can add up, even though the inner gradient steps are computationally cheap due to the distributed implementation. In contrast, our method performs a proximal-gradient-like update for the primal variables with a simple stopping condition, providing significant computational efficiency.
> >
> >     In Kim (2021), the initialization with $\lambda_0 = \mu_0$ leads to dual updates identical to those in standard augmented Lagrangian-based algorithms, which can suffer from slow outer-loop convergence. Our algorithm uses proximal-like dual updates with a fixed step size, which guarantees faster outer-loop convergence to a stationary point.
> >
> >     Finally, while Kim (2021) focuses on smooth coupling constraints, our algorithm (PPALA) can handle non-smooth constraints and provides rigorous analysis for this broader class of problems, as opposed to the GNEP study in Kim (2021).

---

### Review · Reviewer_gXqk · 2025-12-08

**Summary Of Contributions:**

This paper proposes a new primal-dual framework for solving non-convex optimization problems with possibly non-smooth functional constraints, based on a proximal-perturbed augmented Lagrangian (PPAL) construction. Two algorithms are developed: P-Lagrangian based Alternating Direction Algorithm (PLADA) for non-smooth constraints and PPAL-based first-order Algorithm (PPALA) for smooth constraints. Numerical experiments include fair logistic regression, intersectional group fairness in neural networks, and multi-class Neyman-Pearson classification.

**Audience:**

Yes

**Audience Explanation:**

The numerical experiments evaluate PLADA and PPALA on a variety of machine learning tasks involving non-convex objectives and non-smooth or highly non-convex functional constraints, including fair logistic regression on the Adult, Bank, and COMPAS datasets, deep neural network training with intersectional group fairness constraints on Communities and Crime, and multi-class Neyman-Pearson classification on Fashion-MNIST and CIFAR-10. The experimental coverage is broad, spanning both traditional and deep models as well as multiple types of functional constraints, and the reported results are abundant, providing extensive empirical evidence for the claimed performance of the proposed methods.

**Broader Impact Concerns:**

No ethical concern founded.

**Claims And Evidence:**

No

**Claims Explanation:**

1. **Intractable primal update in PLADA (Algorithm 1).** In Line 3 of Algorithm 1 (PLADA), the update of the primal variable requires solving a subproblem containing the term $\langle\boldsymbol{\lambda}\_k,g(\mathbf{x})\rangle$, where $g$ is allowed to be non-convex non-smooth. Under Assumption 8, $g$ is merely $M\_g$-Lipschitz meaning its subgradients are bounded, but this provides no curvature control and does not imply convexity or weak convexity. Consequently, the subproblem in Line 3 remains fully non-convex, and the quadratic regularization term $\frac{1}{2\eta}||\mathbf{x}-\mathbf{x}\_{k}||^2$ cannot guarantee convexity. Therefore, computing $\mathbf{x}\_{k+1}$ requires globally minimizing a general non-convex, non-smooth function at every iteration, which is computationally intractable.

2. **Unjustified boundedness of dual iterates (Assumption 7).** Assumption 7 assumes that the dual iterates $\{\boldsymbol{\lambda}_k\}$ lie in a fixed compact convex set, but the paper provides no justification for why this assumption holds in the non-convex, non-smooth setting considered. The references cited (e.g., Boob et al. 2022; Huang & Lin 2023) do not assume boundedness of the Lagrange multipliers; instead, they prove boundedness as a consequence of constraint qualifications. Thus, Assumption 7 is not "standard" as claimed, and its use is circular: the convergence analysis requires bounded multipliers, but their boundedness is neither established nor derivable from the stated assumptions, making the analysis incomplete. Moreover, the cited references pertain to sequential quadratic programming (SQP) methods, which are only loosely related to the context of the present primal-dual framework and therefore do not meaningfully support the assumption.

3. **Missing justification of multiplier boundedness in PPALA (Algorithm 2).** For Algorithm 2 (PPALA), although the primal subproblem in Line 3 is well defined and computationally tractable as a standard proximal gradient step, the convergence analysis still relies critically on the existence of a uniform bound on the dual iterates $\{\boldsymbol{\lambda}\_k\}$, say $B_{\lambda}:=\sup_{k}||\boldsymbol{\lambda}\_k||$, since the stepsize condition $0<\eta<\frac{1}{L\_{\ell}+3\rho M\_g^2}$ is meaningful only when $L_{\ell}$, which depends on $B_{\lambda}$, is finite. However, Lemma 21 neither establishes boundedness of $\{\boldsymbol{\lambda}\_k\}$ nor invokes Assumption 7, and in Appendix F the non-asymptotic rate proofs merely cite earlier lemmas without providing any structural argument or constraint qualification that would imply such boundedness. Consequently, the convergence guarantees for PPALA depend on an unproved and essential property of the multiplier sequence, and thus the theoretical development remains incomplete.

**Requested Changes:**

See the three point above.

---

> ### Author Response · Authors · 2026-01-13
>
> We thank the reviewer for their detailed and constructive feedback. Our responses to each point are provided below.
>
> 1.  **Intractable primal update in PLADA (Algorithm 1).** We acknowledge the valid concern regarding the computational challenge of the primal subproblem in Algorithm 1 when $g(x)$ is non-convex and non-smooth.
>
>     The primary purpose of Algorithm 1 (PLADA) is to establish a theoretical baseline for the proposed P-Lagrangian framework under non-smooth constraints. While we utilize the $\text{argmin}$ operator to define the subproblem and drive the stationarity analysis, we agree that globally minimizing a general non-convex function is intractable. To address this inherent challenge, we introduced Algorithm 2 (PPALA) in Section 4 specifically for smooth constraints. Algorithm 2 employs a fully tractable proximal gradient update, which allows us to maintain the theoretical benefits of the P-Lagrangian framework while ensuring computational feasibility.
>
>     We will clarify in the revised manuscript that the convergence rate guarantees provided for Algorithm 1 are with respect to the outer loop iterations, under the assumption that the subproblem is solved to global optimality. This analysis serves to bound the iteration complexity of the framework itself, separating it from the complexity of the inner subproblem.
>
>     In our experiments (Section 5), we bridge the gap between theory and practice by approximating the primal update of Algorithm 1 with a single-step first-order update. Despite the theoretical intractability of the exact update, this approximation has proven empirically effective and computationally efficient, consistently guiding the iterates toward stationarity and feasibility across the tested benchmarks.
>
>     We will revise the paper to explicitly highlight that Algorithm 1 serves as the conceptual framework for non-smooth constraints, while Algorithm 2 provides the tractable realization for smooth problems.
>
> 2.  **Unjustified boundedness of dual iterates (Assumption 7).** We appreciate the reviewer’s comment on the distinction between the assumptions used in SQP methods and those in the Augmented Lagrangian methods. To better support Assumption 7 within the context of our proposed framework, we will replace the previous references with works that specifically utilize similar boundedness assumptions within the Augmented Lagrangian and Exact Penalty literature: Section 3 of [1], Section 6 of [2] and Section 4.3 of [3].
>
>     [1] Pillo GD, Grippo L, Lampariello F (1980) A method for solving equality constrained optimization problems by unconstrained minimization. In: *Optimization Techniques. Lecture Notes in Control and Information Sci.*, vol 23, pp 96–105, Springer-Verlag.
>
>     [2] Pillo GD, Lucidi S (2002) An augmented lagrangian function with improved exactness properties. *SIAM Journal on Optimization*, 12(2), 376–406.
>
>     [3] Bertsekas D (1982) *Constrained Optimization and Lagrange Multiplier Methods*. Elsevier, Belmont, Mass.
>
> 3.  **Missing justification of multiplier boundedness in PPALA (Algorithm 2).** We fully accept the reviewer's observation. We agree that the definition of the Lipschitz constant $L_l$ (and consequently the stepsize $\eta$) implicitly relies on the uniform boundedness of the dual iterates $B_\lambda$.
>
>     The convergence analysis in Lemmas 37 and 38 utilize the bound $B_\lambda$ to establish the descent properties of the Lagrangian. Therefore, these lemmas, along with the main Lemma 21, must formally inherit the boundedness assumption. We will revise Lemma 21 and the supporting Lemmas 37 and 38 to explicitly include Assumption 7 in their hypotheses. This revision will ensure that $L_l$ is well-defined and close the logical gap in the theoretical development of Algorithm 2. We thank the reviewer for pointing out this subtlety.

---

### Review · Reviewer_FvRi · 2025-12-08

**Summary Of Contributions:**

Summary of contributions: This paper introduces a new primal–dual optimization framework that solves non-convex functional constraint problems with simple first-order algorithms and improved complexity than previous methods and demonstrates strong performance on practical machine-learning tasks.

Strengths:

1. Significant theoretical contribution: The paper delivers a single-loop primal–dual algorithm that improves the best-known iteration complexity from the existing $\tilde{O}(\epsilon^{-3})$ to $\tilde{O}(\epsilon^{-2})$. Sections 3 and 4 provide rigorous convergence analyses for both non-smooth and smooth constraints.

2. The experimental results also demonstrate the practical efficiency of the algorithm proposed in this paper.

Weaknesses:

1. Assumption 7, which assumes that the sequence $\lambda_k$ generated by the algorithm remains bounded, seems somewhat strong and lacks justification. Although the authors cite several papers that adopt similar assumptions, this assumption is algorithm-dependent, and it is not clear that the proposed algorithm necessarily satisfies it.

2. In Algorithms 1 and 2, the additional variable z is unnecessary. While z may be useful for the theoretical analysis, it is not needed in the actual implementation, and including it in the algorithm description may lead to confusion.

**Audience:**

Yes

**Audience Explanation:**

The proposed framework is broadly applicable to constrained machine learning problems, and the techniques developed in this paper are likely to be of interest to audiences in related fields from both theoretical and practical perspectives.

**Broader Impact Concerns:**

No concern.

**Claims And Evidence:**

Yes

**Claims Explanation:**

The theoretical analysis and experimental results in this paper appear to be sound.

**Requested Changes:**

1. In Algorithms 1 and 2, the additional variable z is unnecessary. While z may be useful for the theoretical analysis, it is not needed in the actual implementation, and including it in the algorithm description may lead to confusion. I recommend revising the descriptions of Algorithms 1 and 2 accordingly.

2. In my view, Assumption 7, which assumes that the sequence $\lambda_k$ generated by the algorithm remains bounded, seems somewhat strong and lacks justification. Although the authors cite several papers that adopt similar assumptions, this assumption is algorithm-dependent, and it is not clear that the proposed algorithm necessarily satisfies it. It would be better if the authors could provide a justification for Assumption 7.

---

> ### Author Response · Authors · 2026-01-13
>
> We thank the reviewer for their valuable feedback. We agree with the suggestions to streamline the algorithm descriptions and appreciate the opportunity to clarify the theoretical basis for our boundedness assumption. Our detailed responses are provided below.
>
> * **[Weakness 1 / Requested Change 2]** We acknowledge the reviewer’s concern that assuming the boundedness of $\lambda_k$ (Assumption 7) appears strong and algorithm-dependent. We would like to clarify that this assumption is a standard structural requirement within the Augmented Lagrangian and Exact Penalty method literature, rather than an unique property of our specific algorithm.
>
>     Augmented Lagrangian methods rely on the equivalence between the constrained problem and the unconstrained minimization of the Lagrangian. Classical theoretical results [1, 2, 3] establish that this equivalence holds for a finite penalty parameter $\rho$ only if the corresponding Lagrange multipliers are bounded. If the multipliers were unbounded, $\rho$ would need to grow to infinity to enforce feasibility.
>
>     Assumption 7 acts as a proxy for standard geometric conditions. Specifically, Constraint Qualifications such as MFCQ or CPLD guarantee that the set of optimal multipliers is bounded.
>
>     To address the reviewer's concern, we have revised the discussion of Assumption 7 in the manuscript. We now explicitly state that this assumption is satisfied under standard conditions. Furthermore, we have added three classical references [1, 2, 3] that utilize this specific assumption to establish the exactness of Augmented Lagrangian functions.
>
>     [1] Pillo GD, Grippo L, Lampariello F (1980) A method for solving equality constrained optimization problems by unconstrained minimization. In: *Optimization Techniques. Lecture Notes in Control and Information Sci.*, vol 23, pp 96–105, Springer-Verlag.
>
>     [2] Pillo GD, Lucidi S (2002) An augmented lagrangian function with improved exactness properties. *SIAM Journal on Optimization*, 12(2), 376–406.
>
>     [3] Bertsekas D (1982) *Constrained Optimization and Lagrange Multiplier Methods*. Elsevier, Belmont, Mass.
>
> * **[Weakness 2 / Requested Change 1]** We completely agree with the reviewer. The auxiliary variable $z$ serves only to facilitate the theoretical analysis (specifically, tracking the stationarity and dual gap) and is not required for the practical execution of the algorithm.
>
>     We have modified the pseudocode for Algorithm 1 (PLADA) and Algorithm 2 (PPALA) to remove the explicit update of $z$. We have moved the definition of $z$ to the Theoretical Analysis section, introducing it strictly as a virtual sequence for the proofs. This change makes the algorithm descriptions more concise and easier to implement without altering the mathematical properties of the method.

---

### Decision · Action_Editor_U53v · 2026-01-25

**Recommendation:** Reject

**Audience:**

Yes

**Audience Explanation:**

Nonconvex constrained optimization is of interest to the machine learning society due to many applications in fairness constrained problems, continual learning and so on.

**Claims And Evidence:**

No

**Claims Explanation:**

Based on the reviewer’s comments and my own reading, I believe there are still major issues in the theoretical proof of this paper, mainly due to overly strong assumptions and several non-rigorous arguments. The overall stance of the reviewers is also leaning toward rejection.

1. The convergence analysis relies on a dual boundedness assumption. In my view, this property should be proved or guaranteed by the algorithm, rather than taken as an assumption. Establishing dual boundedness is often highly nontrivial.

2. The claims of primal feasibility and complementarity slackness require that $\lambda - \mu \to 0$. However, the authors’ argument for $\lambda - \mu \to 0$ is incorrect. At best, the current proof can only establish subsequence convergence to zero. Under the logic of the paper, this would then require assuming a global error bound condition, which also needs proper justification rather than being directly assumed.

3. Even a local error bound is not a mild condition. Proving local error bounds is frequently highly nontrivial and therefore must be carefully justified.

In summary, I believe the paper, in its current form, is not suitable for publication.

**Resubmission Of Major Revision:**

The authors may consider submitting a major revision at a later time.